# Influence of fake news in Twitter during the 2016 US presidential election

Alexandre Bovet [iD] [1,2,3] & Hernán A. Makse[1]

The dynamics and influence of fake news on Twitter during the 2016 US presidential election remains to be clarified. Here, we use a dataset of 171 million tweets in the five months preceding the election day to identify 30 million tweets, from 2.2 million users, which contain a link to news outlets. Based on a classification of news outlets curated by www.opensources. co, we find that 25% of these tweets spread either fake or extremely biased news. We characterize the networks of information flow to find the most influential spreaders of fake and traditional news and use causal modeling to uncover how fake news influenced the presidential election. We find that, while top influencers spreading traditional center and left leaning news largely influence the activity of Clinton supporters, this causality is reversed for the fake news: the activity of Trump supporters influences the dynamics of the top fake news spreaders.

[1] Levich Institute and Physics Department, City College of New York, New York, NY 10031, USA. [2] ICTEAM, Université Catholique de Louvain, Avenue George Lemaître 4, 1348 Louvain-la-Neuve, Belgium. [3] naXys and Department of Mathematics, Université de Namur, Rempart de la Vierge 8, 5000 Namur, Belgium. Correspondence and requests for materials should be addressed to H.A.M. (email: hmakse@ccny.cuny.edu)

Recent social and political events, such as the 2016 US presidential election[1], have been marked by a growing number of so-called "fake news", i.e. fabricated information that disseminate deceptive content, or grossly distort actual news reports, shared on social media platforms. While misinformation and propaganda have existed since ancient times[2], their importance and influence in the age of social media is still not clear. Indeed, massive digital misinformation has been designated as a major technological and geopolitical risk by the 2013 report of the World Economic Forum[3]. A substantial number of studies have recently investigated the phenomena of misinformation in online social networks such as Facebook[4–10], Twitter[10–13], YouTube[14], or Wikipedia[15]. These investigations, as well as theoretical modeling[16,17], suggest that confirmation bias[18] and social influence results in the emergence, in online social networks, of user communities that share similar beliefs about specific topics, i.e. echo chambers, where unsubstantiated claims or true information, aligned with these beliefs, are as likely to propagate virally[6,19]. A comprehensive investigation of the spread of true and false news in Twitter also showed that false news is characterized by a faster and broader diffusion than true news mainly due to the attraction of the novelty of false news[12]. A polarization in communities is also observed in the consumption of news in general[20,21] and corresponds with political alignment[22]. Recent works also revealed the role of bots, i.e. automated accounts, in the spread of misinformation[12,23–25]. In particular, Shao et al. found that, during the 2016 US presidential election on Twitter, bots were responsible for the early promotion of misinformation, that they targeted influential users through replies and mentions[26] and that the sharing of fact-checking articles nearly disappears in the core of the network, while social bots proliferate[13]. These results have raised the question of whether such misinformation campaigns could alter public opinion and endanger the integrity of the presidential election[24]. Here, we use a dataset of 171 million tweets sent by 11 million users covering almost the whole activity of users regarding the two main US presidential candidates, Hillary Clinton and Donald Trump, collected during the five months preceding election day and used to extract and analyze Twitter opinion trend in our previous work[27]. We compare the spread of news coming from websites that have been described as displaying fake news with the spread of news coming from traditional, fact-based, news outlets with different political orientations. We relied upon the opinion of communications scholars (see Methods for details) who have classified websites as containing fake news or extremely biased news. We investigate the diffusion in Twitter of each type of media to understand what is their relative importance, who are the top news spreaders, and how they drive the dynamics of Twitter opinion. We find that, among the 30.7 million tweets containing an URL directing to a news outlet website, 10% point toward websites containing fake news or conspiracy theory and 15% point toward websites with extremely biased news. When considering only tweets originating from non-official Twitter clients, we see a tweeting rate for users tweeting links to websites containing news classified as fake more than four times larger than for traditional media, suggesting a larger role of bots in the diffusion of fake news. We separate traditional news outlets from the least biased to the most biased and reconstruct the information flow networks by following retweets tree for each type of media. User diffusing fake news form more connected networks with less heterogeneous connectivity than users in traditional center and left leaning news diffusion networks. While top news spreaders of traditional news outlets are journalists and public figures with verified Twitter accounts, we find that a large number of top fake and extremely biased news spreaders are unknown users or users with deleted Twitter accounts. The presence of two

clusters of media sources and their relation with the supporters of each candidate is revealed by the analysis of the correlation of their activity. Finally, we explore the dynamics between the top news spreaders and the supporters' activity with a multivariate causal network reconstruction[28]. We find two different mechanisms for the dynamics of fake news and traditional news. The top spreaders of center and left leaning news outlets, who are mainly journalists, are the main drivers of Twitter's activity and in particular of Clinton supporters' activity, who represent the majority in Twitter[27]. For fake news, we find that it is the activity of Trump supporters that governs their dynamics and top spreaders of fake news are merely following it.

## Results

**News spreading in Twitter**. To characterize the spreading of news in Twitter we analyze all the tweets in our dataset that contained at least one URL (Uniform Resource Locator, i.e. web address) linking to a website outside of Twitter. We first separate URL in two main categories based on the websites they link to: websites containing misinformation and traditional, fact-based, news outlets. We use the term traditional in the sense that news outlets in this category follow the traditional rules of fact-based journalism and therefore also include recently created news outlets (e.g. vox.com).

Classifying news outlets as spreading misinformation or real information is a matter of individual judgment and opinion, and subject to imprecision and controversy. We include a finer classification of news outlets spreading misinformation in two sub-categories: fake news and extremely biased news. Fake news websites are websites that have been flagged as consistently spreading fabricated news or conspiracy theories by several fact-checking groups. Extremely biased websites include more controversial websites that not necessarily publish fabricated information but distort facts and may rely on propaganda, decontextualized information, or opinions distorted as facts. We base our classification of misinformation websites on a curated list of websites which, in the judgment of a media and communication research team headed by a researcher of Merrimack College, USA, are either fake, false, conspiratorial, or misleading (see Methods). They classify websites by analyzing several aspects, such as if they try to imitate existing reliable websites, if they were flagged by fact-checking groups (e.g. snopes. com, hoax-slayer.com, and factcheck.org), or by analyzing the sources cited in articles (the full explanation of their methods is available at www.opensources.co). We discard insignificant outlets accumulating less than 1% of the total number of tweets in their category. We classify the remaining websites in the extremely biased category according to their political orientation by manually checking the bias report of each websites on www. allsides.com and mediabiasfactcheck.com. Details about our classification of websites spreading misinformation is available in the Methods section.

We also use a finer classification for traditional news websites based on their political orientation. We identify the most important traditional news outlets by manually inspecting the list of top 250 URL's hostnames, representing 79% of all URLs, shared on Twitter. We classify news outlets as right, right leaning, center, left leaning, or left based on their reported bias on www. allsides.com and mediabiasfactcheck.com. The news outlets in the right leaning, center, and left leaning categories are more likely to follow the traditional rules of fact-based journalism. As we move toward more biased categories, websites are more likely to have mixed factual reporting. As for misinformation websites, we discard insignificant outlets by keeping only websites that accumulate more than 1% of the total number of tweets of their

respective category. Although we do not know how many news websites are contained in the list of less popular URLs, a threshold as small as 1% allows us to capture a relatively broad sample of the media in term of popularity. Assuming that the decay in popularity of the websites in each media category is similar, our measure of the proportion of tweets and users in each category should not be significantly changed if we extended our measure to the entire dataset of tweets with URLs. While the detail of our classification is subject to some subjectivity, we find that our analysis reveals patterns encompassing several media

categories that form a group with similar characteristics. Our results are therefore robust to changes of classification within these larger group of media.

We report the hostnames in each categories along with the number of tweets with a URL pointing toward them in Supplementary Table 1. Using this final separation in seven classes, we identify in our dataset (we give the top hostname as an example in parenthesis): 16 hostnames corresponding to fake news websites (e.g. thegatewaypundit.com), 17 hostnames for extremely biased (right) news websites (e.g. breitbart.com), 7 hostnames for extremely biased (left) news websites (e.g. dailynewsbin.com), 18 hostnames for left news websites (e.g. huffingtonpost.com), 19 hostnames for left leaning news websites (e.g. nytimes.com), 13 hostnames for center news websites (e.g. cnn.com), 7 hostnames for right leaning websites (e.g. wsj.com), and 20 hostnames for right websites (e.g. foxnews.com).

We identified 30.7 million tweets with an URL directing to a news outlet website, sent by 2.3 million users. An important point when comparing the absolute number of tweets and users contributing to the spread of different types of news is the bias introduced by the keywords selected during the data collection. Indeed, if we had used keywords targeting specific news outlets or hashtags concerning specific news event, it would be impossible to perfectly control the bias toward fake and reliable news or the representation of the political orientation of the tweet sample. Here, we used neutral keywords in term of media representation, the names of the two main candidates to the presidential election (see Methods), in order to collect a sample representative of the real coverage of the election on Twitter by all media sources.

We see a large number of tweets linking to fake news websites and extremely biased news websites (Fig. 1a and Table 1). However, the majority of tweets linking to news outlets points toward left leaning news websites closely followed by center news websites. Tweets directing to left and left leaning news websites represent together 38% of the total and tweets directing towards center news outlets represents 21%. Tweets directing to fake and extremely biased news websites represents a share of 25%. When considering the number of distinct users having sent the tweets instead of the number of tweets (Fig. 1b and Table 1), the share of left and left leaning websites increases to 43% and the share of center news to 29%, while the share going to fake news and extremely biased news is equal to 12% (the share of users differ slightly from Table 1 when grouping categories as users may belong to several categories). The number of tweets linking to websites producing fake and extremely biased news is comparable with the number for center, left and left leaning media outlets. However, users posting links to fake news or extreme bias (right) websites are, in average, more active than users posting links to other news websites (Table 1). In particular, they post around

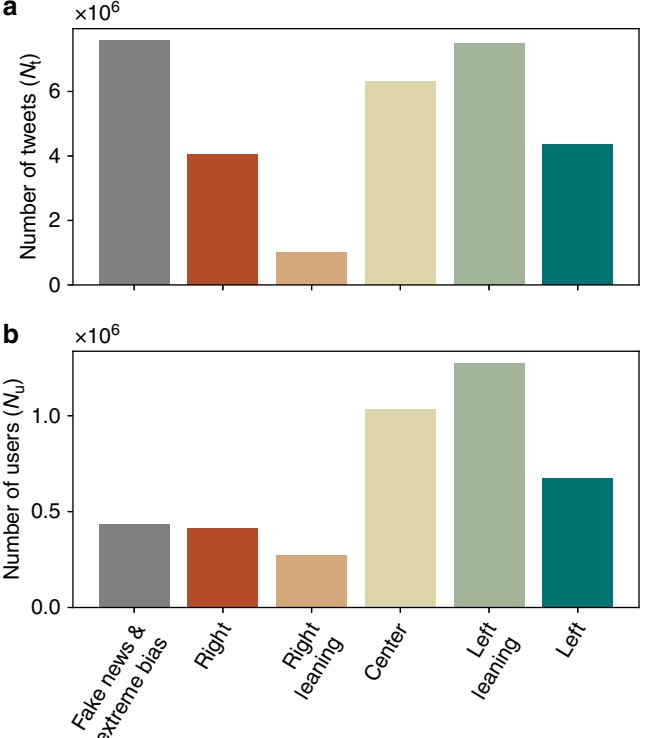

**Fig. 1** Importance of different types of news outlets in Twitter. Number of distinct tweets (**a**) and number of distinct users having sent tweets (**b**) with a URL pointing to a website belonging to one of following categories: fake or extremely biased, right, right leaning, center, left and left leaning news outlets. While the tweet volume of fake and extremely biased news is comparable to the tweet volumes of center and left volume (**a**), users posting fake and extremely biased news are around twice more active in average (see Table 1). Consequently, the share of users posting fake and extremely biased news (**b**) is smaller (12%) than the share of tweets directing toward fake and extremely biased news websites (25%)

**Table 1 Tweet and user volume corresponding to each media category in Twitter**

|  | $N_t$ | $p_t$ | $N_u$ | $p_u$ | $N_t/N_u$ | $p_{t,n/o}$ | $p_{u,n/o}$ | $N_{t,n/o}/N_{u,n/o}$ |
|---|---|---|---|---|---|---|---|---|
| Fake news | 2,991,073 | 0.10 | 204,899 | 0.05 | 14.60 | 0.19 | 0.03 | 80.35 |
| Extreme bias (right) | 3,969,639 | 0.13 | 294,175 | 0.07 | 13.49 | 0.09 | 0.03 | 36.52 |
| Right news | 4,032,284 | 0.13 | 416,510 | 0.10 | 9.68 | 0.11 | 0.04 | 24.80 |
| Right leaning news | 1,006,746 | 0.03 | 272,347 | 0.06 | 3.70 | 0.18 | 0.06 | 11.39 |
| Center news | 6,322,257 | 0.21 | 1,032,722 | 0.24 | 6.12 | 0.20 | 0.05 | 26.68 |
| Left leaning news | 7,491,344 | 0.24 | 1,272,672 | 0.30 | 5.89 | 0.14 | 0.04 | 18.64 |
| Left news | 4,353,999 | 0.14 | 674,744 | 0.16 | 6.45 | 0.14 | 0.05 | 16.64 |
| Extreme bias (left) | 609,503 | 0.02 | 99,743 | 0.02 | 6.11 | 0.06 | 0.03 | 11.46 |

Number, $N_t$, and proportion, $p_t$, of tweets with a URL pointing to a website belonging to one of the media categories. Number, $N_u$, and proportion, $p_u$, of users having sent the corresponding tweets, and average number of tweets per user, $N_t/N_u$, for each category. Proportion of tweets sent by non-official clients, $p_{t,n/o}$, proportion of users having sent at least one tweet from an non-official client, $p_{u,n/o}$, and average number of tweets per user sent from non-official clients, $N_{t,n/o}/N_{u,n/o}$

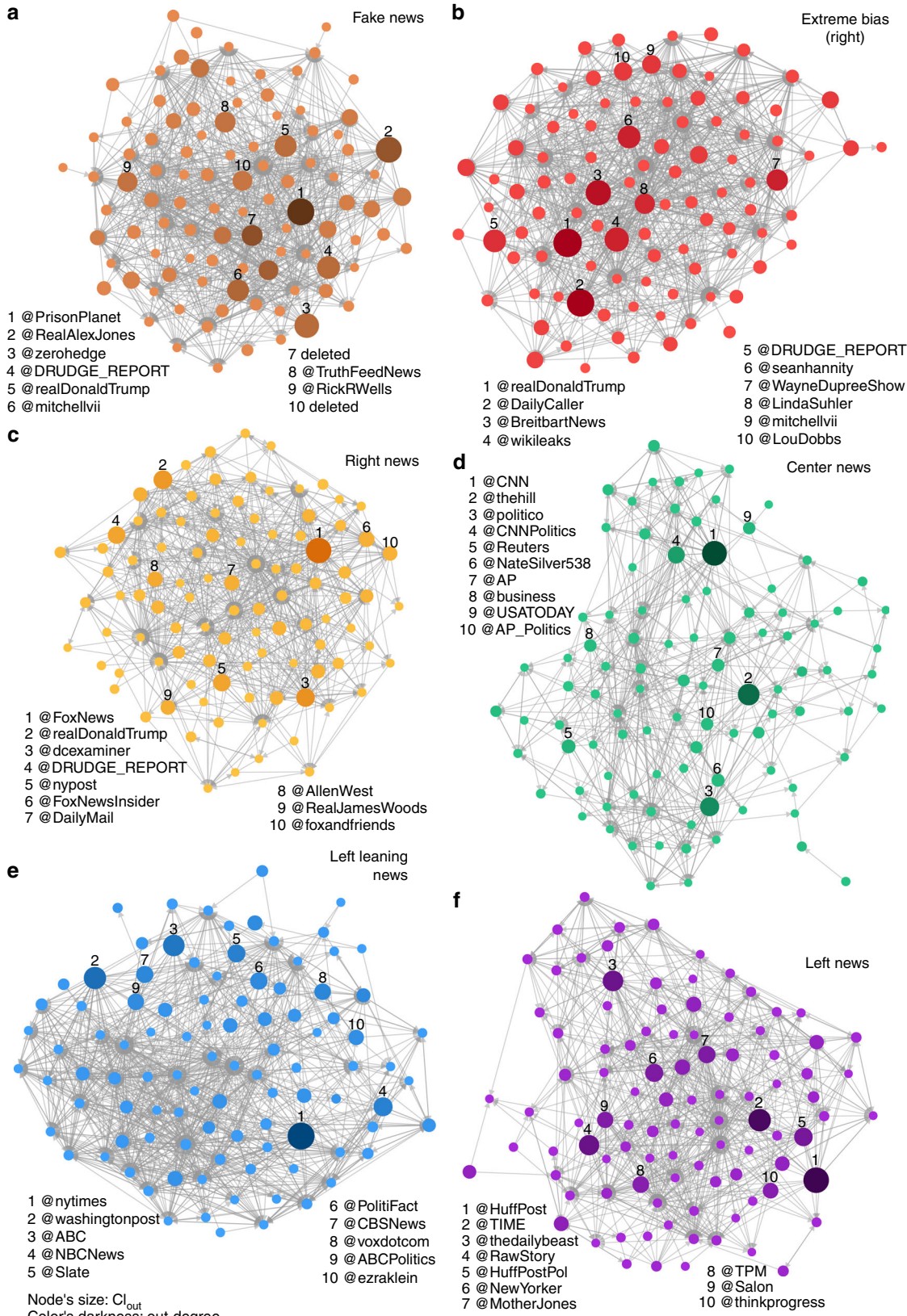

**Fig. 2** Retweet networks formed by the top 100 news spreaders of different media categories. Retweet networks for fake news (**a**), extreme bias (right) news (**b**), right news (**c**), center news (**d**), left leaning news (**e**), and left news (**f**) showing only the top 100 news spreaders ranked according to their collective influence. The direction of the links represents the flow of information between users. The size of the nodes is proportional to their Collective Influence score, $CI_{out}$, and the shade of the nodes' color represents their out-degree, i.e. the number of different users that have retweeted at least one of her/his tweets with a URL directing to a news outlet, from dark (high out-degree) to light (low out-degree). The network of fake (**a**) and extreme bias (right) (**b**) are characterized by a connectivity that is larger in average and less heterogeneous than for networks of center and left leaning news (Table 2)

**Table 2 Retweet networks characteristics for each news source categories**

|  | N nodes | N edges | $\langle k \rangle$ | $\sigma (k_{out})/\langle k \rangle$ | $\sigma (k_{in})/\langle k \rangle$ | max ($k_{out}$) | max ($k_{in}$) |
|---|---|---|---|---|---|---|---|
| Fake news | 175,605 | 1,143,083 | 6.51 | 32 ± 4 | 2.49 ± 0.06 | 42,468 | 1232 |
| Extreme bias (right) | 249,659 | 1,637,927 | 6.56 | 36 ± 6 | 2.73 ± 0.03 | 51,845 | 588 |
| Right | 345,644 | 1,797,023 | 5.20 | 44 ± 11 | 2.70 ± 0.04 | 86,454 | 490 |
| Right leaning | 216,026 | 495,307 | 2.29 | 45 ± 11 | 1.72 ± 0.02 | 32,653 | 129 |
| Center | 864,733 | 2,501,037 | 2.89 | 75 ± 39 | 2.69 ± 0.06 | 229,751 | 512 |
| Left leaning | 1,043,436 | 3,570,653 | 3.42 | 59 ± 19 | 3.38 ± 0.10 | 145,047 | 843 |
| Left | 536,903 | 1,801,658 | 3.36 | 47 ± 12 | 3.50 ± 0.08 | 58,901 | 733 |
| Extreme bias (left) | 78,911 | 277,483 | 3.52 | 33 ± 6 | 2.49 ± 0.08 | 23,168 | 648 |

We show the number of nodes and edges (links) of the networks, the average degree, $\langle k \rangle = \langle k_{in} \rangle = \langle k_{out} \rangle$ (the in-/out-degree of a node is the number of in-going/out-going links attached to it). In a directed network, the average in-degree and out-degree are always equal. The out-degree of a node, i.e. a user, is equal to the number of different users that have retweeted at least one of her/his tweets. Its in-degree represents the number of different users she/he retweeted. The ratio of the standard deviation and the average of the in- and out-degree distribution, $\sigma (k_{in})/\langle k \rangle$ and $\sigma (k_{out})/\langle k \rangle$, measures the heterogeneity of the connectivity of each networks. As the standard deviation of heavy-tailed degree distributions can depend on the network size, we computed the values of $\sigma (k_{in})/\langle k \rangle$ and $\sigma (k_{out})/\langle k \rangle$ by taking the average, and standard error, of 1000 independent samples, of 78,911 values each, drawn from the in- and out-degree distributions of each network

twice the number of tweets compared to users posting links towards center or left leaning news outlets.

The proportion of tweets sent by, and users using, non-official Twitter clients (Table 1) allows to evaluate the importance of automated posting in each category. Details about our classification of official Twitter clients are available in the Methods. We see that the two top categories are fake news and center news with around 20% of tweets being sent from non-official accounts. When considering the proportion of users sending tweets from non-official clients, the number is very similar for all categories, around 4%, showing that the automation of posting plays an important role across all media categories. Indeed, non-official clients includes a broad range of clients, from "social bots" to applications used to facilitate the management of professional Twitter accounts. A large discrepancy between sources arises when we consider the average number of tweets per users sent from non-official clients (Table 1). Users using non-official clients to send tweets with links directing to websites displaying fake news tweeted an average of 80 times during the collection period, which is more than twice the value for other types of news outlets. This high activity from non-official clients suggests an abnormal presence of bots. The role of bots in the diffusion of fake news has already been documented[13,26] as well as their presence in the Twitter discussions during 2016 US election[24].

We note that Breitbart News is the most dominant media outlet in term of number of tweets among the right end of the outlet categories with 1.8 million tweets (see Supplementary Table 1). We examine the relation between Breitbart and the rest of the media outlets in Supplementary Note 1, Supplementary Tables 2–6 as well as Supplementary Fig. 1. Our analysis shows that removing Breitbart from the extreme bias category does not change our results significantly.

**Networks of information flow**. To investigate the flow of information we build the retweet networks for each category of news websites, i.e. when a user $u$ retweets (a retweet allows a user to rebroadcast the tweet of an other user to his followers) the tweet of a user $v$ that contains a URL linking to a website belonging to one of the news media category, we add a link, or edge, going from node $v$ to node $u$ in the network. The direction of the links represents the direction of the information flow between Twitter users. We do not consider multiple links with the same direction between the same two users and neither consider self-links, i.e. when a user retweet her/his own tweet. The out-degree of a node is its number of outgoing links and is equal to the number of different users that have retweeted at least one of her/his tweets. Its in-degree is its number of in-going links and represents the number of different users she/he retweeted.

Figure 2 shows the networks formed by the top 100 news spreaders of the six most important retweet networks. The retweet networks for right leaning and extreme bias (left) news is shown in Supplementary Fig. 2. We explain in the section Top news spreaders and in the Methods how the news spreaders are identified. A clear difference is apparent between the networks representing the flow of fake and extremely biased (right) news and the networks for left leaning and center news (Table 2 and Supplementary Fig. 3). The left leaning and center news outlets correspond to larger networks in term of number of nodes and edges, revealing their larger reach and influence in Twitter. However, the retweet networks corresponding to fake and extremely biased (right) news outlets are the most dense with an average degree $\langle k \rangle \simeq 6.5$. The retweet network for right news has characteristics in between those two groups with a slightly larger size than the networks for fake and extremely biased (right) news and a larger average degree than center news. These results show that users spreading fake and extremely biased news, although in smaller numbers, are not only more active in average (Table 1), but also connected (through retweets) to more users in average than users in the traditional news networks. Table 2 also shows that the center and left leaning networks have the most heterogeneous out-degree distribution and the fake news retweet networks has the less heterogeneous out-degree distribution. We measure the heterogeneity of the distribution with a bootstrapping procedure (see Table 2) to ensure the independence of the measure on the networks' sizes. Our analysis indicates that the larger networks (center, left leaning) differ from the smaller ones not just by their size but also by their structure. The heterogeneity of the degree distribution plays an important role in spreading processes on networks, indicating a strong hierarchical diffusion cascade from hubs to intermediate degree, and finally to small degree classes[29,30]. The characteristics of the weighted retweet networks, taking into account multiple interactions between users, reveal the same patterns than the unweighted networks (Supplementary Table 7). Table 2 and Supplementary Fig. 3a reveals the existence of users with very large out-degree ($k_{out} > 5 \times 10^5$), in the center and left leaning networks, i.e. very important broadcasters of information, which are not present in other networks. This suggests that different mechanisms of information diffusion could be at play in the center and left leaning news networks, where high degree nodes may play a more important role, than in the fake and extremely biased news networks.

We note that a difference between the largest networks, i.e center and left leaning news, and the fake and extremely biased networks is that the former have typically access to more broadcasting technologies, which may be disruptive to understanding diffusion patterns based on network data[31]. The

**Table 3 Top 25 CI news spreaders of the retweet networks corresponding to each media category**

| Rank | Fake news (7 verified, 2 deleted, 16 unverified) | Extreme bias (right) news (15 verified, 1 deleted, 9 unverified) | Right news (23 verified, 0 deleted, 2 unverified) | Right leaning news (20 verified, 1 deleted 4 unverified) |
|---|---|---|---|---|
| 1 | @PrisonPlanet✓ | @realDonaldTrump✓ | @FoxNews✓ | @WSJ✓ |
| 2 | @RealAlexJones✓ | @DailyCaller✓ | @realDonaldTrump✓ | @WashTimes✓ |
| 3 | @zerohedge | @BreitbartNews✓ | @dcexaminer✓ | @RT_com✓ |
| 4 | @DRUDGE_REPORT | @wikileaks✓ | @DRUDGE_REPORT | @realDonaldTrump✓ |
| 5 | @realDonaldTrump✓ | @DRUDGE_REPORT | @nypost✓ | @RT_America✓ |
| 6 | @mitchellvii✓ | @seanhannity✓ | @FoxNewsInsider✓ | @WSJPolitics✓ |
| 7 | deleted | @WayneDupreeShow✓ | @DailyMail✓ | @DRUDGE_REPORT |
| 8 | @TruthFeedNews | @LindaSuhler✓ | @AllenWest✓ | @KellyannePolls✓ |
| 9 | @RickRWells | @mitchellvii✓ | @RealJamesWoods✓ | @TeamTrump✓ |
| 10 | deleted | @LouDobbs✓ | @foxandfriends✓ | @LouDobbs✓ |
| 11 | @gatewaypundit✓ | @PrisonPlanet✓ | @foxnation✓ | @rebeccaballhaus✓ |
| 12 | @infowars | @DonaldJTrumpJr✓ | @LouDobbs✓ | @WSJopinion✓ |
| 13 | @Lagartija_Nix | @gerfingerpoken | @KellyannePolls✓ | @reidepstein✓ |
| 14 | @DonaldJTrumpJr✓ | @FreeBeacon✓ | @JudicialWatch✓ | deleted |
| 15 | @ThePatriot143 | @gerfingerpoken2 | @PrisonPlanet✓ | @JasonMillerinDC✓ |
| 16 | @V_of_Europe | @TeamTrump✓ | @wikileaks✓ | @DanScavino✓ |
| 17 | @KitDaniels1776 | @Italians4Trump | @TeamTrump✓ | @PaulManafort✓ |
| 18 | @Italians4Trump | @benshapiro✓ | @IngrahamAngle✓ | @SopanDeb✓ |
| 19 | @_Makada_ | @KellyannePolls✓ | @marklevinshow✓ | @asamjulian |
| 20 | @BigStick2013 | @DanScavino✓ | @LifeZette✓ | @JudicialWatch✓ |
| 21 | @conserv_tribune✓ | deleted | @theblaze✓ | @_Makada_ |
| 22 | @Miami4Trump | @JohnFromCranber | @FoxBusiness✓ | @mtracey✓ |
| 23 | @MONAKatOILS | @true_pundit | @foxnewspolitics✓ | @Italians4Trump |
| 24 | @JayS2629 | @ThePatriot143 | @BIZPACReview | @Telegraph✓ |
| 25 | @ARnews1936 | @RealJack | @DonaldJTrumpJr✓ | @RealClearNews✓ |

| Rank | Center news (24 verified, 0 deleted, 1 unverified) | Left leaning news (25 verified, 0 deleted 0 unverified) | Left news (25 verified, 0 deleted, 0 unverified) | Extreme bias (left) news (7 verified, 1 deleted, 17 unverified) |
|---|---|---|---|---|
| 1 | @CNN✓ | @nytimes✓ | @HuffPost✓ | @Bipartisanism✓ |
| 2 | @thehill✓ | @washingtonpost✓ | @TIME✓ | @PalmerReport✓ |
| 3 | @politico✓ | @ABC✓ | @thedailybeast✓ | @peterdaou✓ |
| 4 | @CNNPolitics✓ | @NBCNews✓ | @RawStory✓ | @crooksandliars✓ |
| 5 | @Reuters✓ | @Slate✓ | @HuffPostPol✓ | @BoldBlueWave |
| 6 | @NateSilver538✓ | @PolitiFact✓ | @NewYorker✓ | @Shareblue✓ |
| 7 | @AP✓ | @CBSNews✓ | @MotherJones✓ | @Karoli |
| 8 | @business✓ | @voxdotcom✓ | @TPM✓ | @RealMuckmaker |
| 9 | @USATODAY✓ | @ABCPolitics✓ | @Salon✓ | @GinsburgJobs |
| 10 | @AP_Politics✓ | @ezraklein✓ | @thinkprogress✓ | @AdamsFlaFan |
| 11 | @FiveThirtyEight✓ | @nytpolitics✓ | @mmfa✓ | @mcspocky |
| 12 | @bpolitics✓ | @guardian✓ | @joshtpm✓ | @Shakestweetz✓ |
| 13 | @jaketapper✓ | @NYDailyNews✓ | @MSNBC✓ | deleted |
| 14 | @DRUDGE_REPORT | @latimes✓ | @NYMag✓ | @JSavoly |
| 15 | @cnnbrk✓ | @BuzzFeedNews✓ | @samstein✓ | @OccupyDemocrats |
| 16 | @businessinsider✓ | @Mediaite✓ | @JuddLegum✓ | @ZaibatsuNews |
| 17 | @AC360✓ | @HillaryClinton✓ | @mashable✓ | @wvjoe911 |
| 18 | @cnni✓ | @nytopinion✓ | @theintercept✓ | @DebraMessing✓ |
| 19 | @brianstelter✓ | @CillizzaCNN✓ | @DavidCornDC✓ | @SayNoToGOP |
| 20 | @KellyannePolls✓ | @MSNBC✓ | @dailykos✓ | @coton_luver |
| 21 | @wikileaks✓ | @KFILE✓ | @JoyAnnReid✓ | @EJLandwehr |
| 22 | @SopanDeb✓ | @TheAtlantic✓ | @nxthompson✓ | @mch7576 |
| 23 | @KFILE✓ | @SopanDeb✓ | @thenation✓ | @RVAwonk |
| 24 | @BBCWorld✓ | @Fahrenthold✓ | @justinjm1✓ | @_Carja |
| 25 | @NewDay✓ | @BuzzFeed✓ | @ariannahuff✓ | @Brasilmagic |

Verified users have a checkmark (✓) next to their user name. Verifying its accounts is a feature offered by Twitter that "lets people know that an account of public interest is authentic" (help.twitter.com/en/managing-your-account/about-twitter-verified-accounts). Unverified accounts do not have a checkmark and accounts marked as deleted have been deleted either by Twitter or by the users themselves

structural differences we observe may be explained by the fact that there is something different about the way that the people in these networks organize and share information but it may also be the case that there are subgroups of users in the center and left leaning news networks that form diffusion networks with a similar structure as the smaller fake and extremely biased news networks and then also have a large number of other individuals added to these subgroups due to the presence of important broadcast networks that feed their ideology or information needs.

While inspecting specific accounts is not the goal of this study, looking at the two accounts with the maximum $k_{out}$ and $k_{in}$ reveals an interesting contrast between users of both networks. The user with the largest out-degree of the center news network is the verified account of the Cable News Network, CNN, (@CNN),

which regularly posts links towards its own website using mainly the non-official professional client Sprinklr (www.sprinklr.com). The user with the largest in-degree of the fake news network is the user @Patriotic_Folks, which, at the moment of this writing, seems to belong to a deceiving user, whose profile description contains the hashtag #MAGA and refer to a website belonging to our fake news website list (thetruthdivision.com). The name of the account is "Annabelle Trump" and its profile picture is a young woman wearing cow-boy clothes (a reverse image search on the web reveals that this profile image is not authentic as it comes in fact from the catalog of a website selling western clothes). Most of its tweet are sent from the official Twitter Web Client, suggesting that a real person is managing the account, and contains URLs directing to the same fake news website. However, having a high in-degree does not indicate that this user has an important influence. Indeed, its out-degree is approximately 3.5 times smaller than its in-degree and, as we explain in the next section, influence is poorly measured by local network properties such as in- or out-degree.

**Top news spreaders**. In order to uncover the most influential users of each retweet network, we use the Collective Influence (CI) algorithm[32] which is based on the solution of the optimal network percolation. For a Twitter user to be highly ranked by the CI algorithm, she/he does not necessarily need to be directly retweeted by many users, but she/he needs to be surrounded by highly retweeted users (see Methods for more details).

We find that top news spreaders of left leaning and center news are almost uniquely verified accounts belonging to news outlets or journalists (Table 3). A very different situation for news spreaders of the fake news and extremely biased news websites is revealed, where, among verified accounts of news websites and journalists, we also find a large number of unknown, unverified, users that are not public figures but are important news spreaders in Twitter (Fig. 3 and Table 3). We also find deleted accounts, which could have been deleted either by Twitter for infringing their rules and policies or by the users themselves, mostly in the fake and extremely biased news spreaders. We find that, based on the timestamp of their last tweet in our dataset, 24 out of the 28 accounts had tweeted after election day (8 November 2016), indicating that they were deleted after the election. Deleted accounts were extremely active, with a median number of tweets of 2224 (minimum: 156, 1st quartile: 1400, 3rd quartile: 6711, and

maximum: 15,930). In comparison, the median number of tweets per users for our entire dataset is 2. We also find that 21 deleted accounts used an unofficial Twitter client (the most used one by deleted accounts is dlvr.it). The list of the right, right leaning, and left news top spreaders form a mix of verified and unverified accounts. Figure 2 shows the retweet networks formed by the top 100 spreaders of each category and Fig. 4 shows the combined retweet network formed by top 30 news spreaders of all media categories and reveals the separation of the top news spreaders in two main clusters as well as the relative importance of the top spreaders. The sets of top 100 fake news, extremely biased (right), right, and right leaning news spreaders have an important overlap, >30 (Fig. 4 and Supplementary Table 8). Fake and extremely biased news is mostly spread by unverified accounts which could be due to the fact that some accounts are trying to hide their real identity but also to the fact that audiences of the fake and extremely biased news are more likely to listen to "non-public" figures due to their distrust of the establishment.

We distinguish three types of unverified accounts: (1) unverified accounts that are not necessarily misleading or deceiving, for example, @zerohedge, @DRUDGE_REPORT or @TruthFeedNews make their affiliation to their respective news websites clear, although their identities or the ones of their websites administrators is not always clear; (2) unverified accounts that make their motif clear in their choice of screen-name, e.g. @Italians4Trump or @Miami4Trump, although the real identity of the persons behind such accounts is also usually undisclosed; (3) finally, unverified accounts that seem to be real persons with profile pictures and user names, e.g @Lagartija_Nix, @ThePatriot143, @BigStick2013, @LindaSuhler, @gerfingerpoken, or @AdamsFlaFan, but are not public figures. Whether such users are authentic, social bots or fake users operated by someone else is not clear. However, our results show that such users are not present in the top news spreaders of the center and left leaning news, while they have a high prevalence in the fake and extremely biased categories.

Another observation is the presence of members of the campaign staffs of each candidate in the top news spreaders (see Supplementary Note 2 and Supplementary Table 9). We see more users linked to the campaign staff of Donald Trump (13), and with higher ranks in term of influence, than to the campaign staff of Hillary Clinton (3), revealing the more important direct role of the Trump team in the diffusion of news in Twitter.

**News spreading dynamics**. To investigate the news spreading dynamics of the different media categories on Twitter, we analyze the correlations between the time series of tweeting rate measured for each category. The Twitter activity time series are constructed by counting the number of tweets with a URL directing toward a website belonging to each of the media category at a 15 min resolution. In addition to the activity related to each media group, we also consider the time series of the activity of the supporters of each presidential candidates. We classify supporters based on the content of their tweets using a supervised machine learning algorithm trained on a dataset obtained from the network of hashtag co-occurrences. The full detail of our method and the validation of its opinion trend with the national polling average of the New York Times is described in ref. [27]. We use our full dataset of tweets concerning the two candidates, namely 171 million tweets sent by 11 million distinct users during more than five months. After removing automated tweets (see Methods), we have a total of 157 million tweets. This represents an average of 1.1 million tweets per day (standard deviation of 0.6 million) sent by an average of about 375,000 distinct users per day (standard deviation of 190,000). A majority of users, 64%, is in favor of

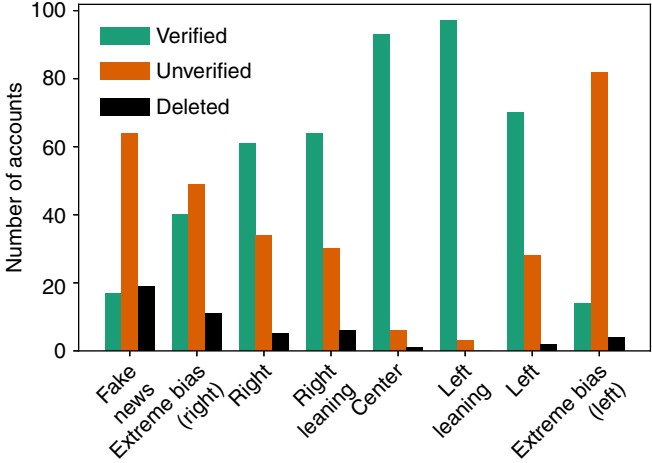

**Fig. 3** Types of top news spreaders accounts per media category. Proportion of verified (green), unverified (orange), and deleted (black) accounts among the top 100 news spreaders in each media category

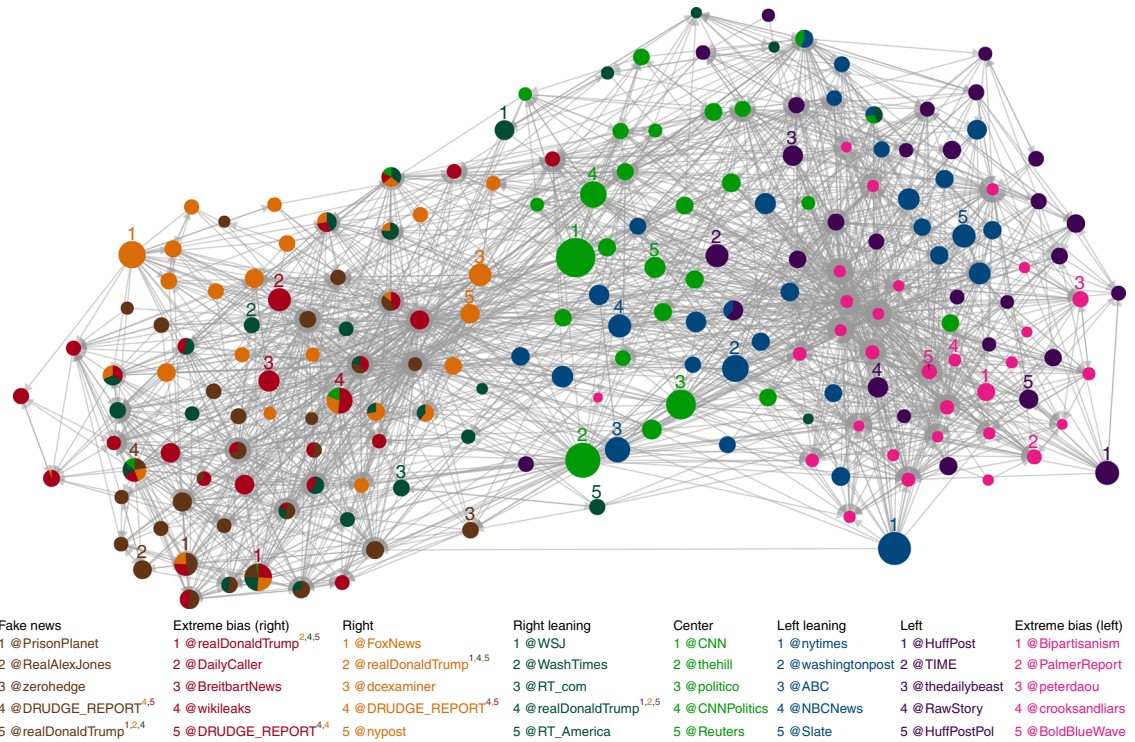

| Fake news | Extreme bias (right) | Right | Right leaning | Center | Left leaning | Left | Extreme bias (left) |
|---|---|---|---|---|---|---|---|
| 1 @PrisonPlanet | 1 @realDonaldTrump[2,4,5] | 1 @FoxNews | 1 @WSJ | 1 @CNN | 1 @nytimes | 1 @HuffPost | 1 @Bipartisanism |
| 2 @RealAlexJones | 2 @DailyCaller | 2 @realDonaldTrump[1,4,5] | 2 @WashTimes | 2 @thehill | 2 @washingtonpost | 2 @TIME | 2 @PalmerReport |
| 3 @zerohedge | 3 @BreitbartNews | 3 @dcexaminer | 3 @RT_com | 3 @politico | 3 @ABC | 3 @thedailybeast | 3 @peterdaou |
| 4 @DRUDGE_REPORT[4,5] | 4 @wikileaks | 4 @DRUDGE_REPORT[4,5] | 4 @realDonaldTrump[1,2,5] | 4 @CNNPolitics | 4 @NBCNews | 4 @RawStory | 4 @crooksandliars |
| 5 @realDonaldTrump[1,2,4] | 5 @DRUDGE_REPORT[4,4] | 5 @nypost | 5 @RT_America | 5 @Reuters | 5 @Slate | 5 @HuffPostPol | 5 @BoldBlueWave |

**Fig. 4** Retweet network formed by the top 30 influencers of each media category. The direction of the links represents the flow of information between users. The size of the nodes is proportional to their out-degree in the complete combined network, i.e. the number of different users that have retweeted at least one of her/his tweets with a URL directing to a news outlet, and the color of the nodes indicates to which news category they belong. Nodes that belong to several news categories are represented by pie charts where the size of each slice is proportional to their $CI_{out}$ ranking, taking into accounts only their rank among the top 30

Hillary Clinton while 28% is in favor of Donald Trump (8% are unclassified as they have the same number of tweets in each camp). However, we find that Trump supporters are, in average, 1.5 times more active than Clinton supporters[27]. The supporters therefore represent the general Twitter population commenting on the candidate of the election.

We removed the trend and circadian cycles present in the time series with the widely used STL (seasonal-trend decomposition procedure based on Loess) method[33], which is a robust iterative filtering method allowing to separate a time series in seasonal (in this case, daily), trend, and remainder components (see Methods).

The separation of the media sources in two correlated clusters is revealed when using a threshold of $r_0 = 0.49$, corresponding to the place of the largest gap between the sorted correlation values (Fig. 5). The value of each cross-correlation coefficient is reported in Supplementary Table 10. The first activity cluster (indicated by a red square in Fig. 5a) comprises the fake, extreme bias (right), and right news. The second activity cluster (indicated by a blue square) is made of the center, left, and left leaning news sources. The activities of right leaning and extremely biased (left) news are only poorly correlated with the other news categories or supporters (see Supplementary Table 10). We observe the following patterns between the media groups and the supporters dynamics: the activity of Clinton supporters has a higher correlation with the second cluster than with the first one while the activity of Trump supporters is equally correlated with the two clusters. This indicate that Trump supporters are likely to react to any type of news while Clinton supporters mostly react to center and news on the left and tend to ignore news coming from the right side.

These results indicate that the media included in the two clusters respond to two different news dynamics and show that

the polarization of news observed at the structural level in previous works[20–22] also corresponds to a separation in dynamics. This separation could be showing that Americans with different political loyalties prefer different news sources but could also be due to the fact that supporters prefer the news that their candidate prefers[34].

In order to investigate the causal relations between news media sources and Twitter dynamics, we use a multivariate causal network reconstruction of the links between the activity of top news spreaders and supporters of the presidential candidates based on a causal discovery algorithm[28,35,36]. The causal network reconstruction tests the independence of each pair of time series, for several time lags, conditioned on potential causal parents with a non-parametric conditional independence test[37,38] (see Methods). We use the causal algorithm as a variable selection and perform a regression of a linear model using only the true causal link discovered. We consider linear causal effects for their reliable estimation and interpretability. This permits us to compare the causal effect as first order approximations, estimate the uncertainties of the model, and reconstruct a causal directed weighted networks[28]. In this framework, the causal effect between a time series $X^i$ and $X^j$ at a time delay $\tau$, $I^{CE}_{i \to j}(\tau)$, is equal to the expected value of $X^j_t$ (in unit of standard deviation) if $X^i_{t-\tau}$ is perturbed by one standard deviation[28].

An assumption of causal discovery is causal sufficiency, i.e. the fact that every common cause of any two or more variables is in the system[35]. Here, causal sufficiency is not satisfied since Twitter's activity is only the observed part of a larger social system and the term "causal" must be understood to be meant relative to the system under study. As for the cross-correlation analysis, we use the residuals of the STL filtering of the 15 tweet volume time series (Fig. 6a, b).

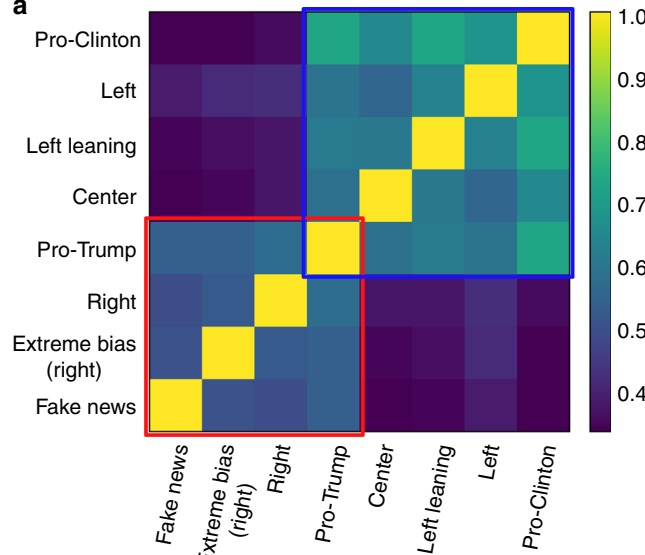

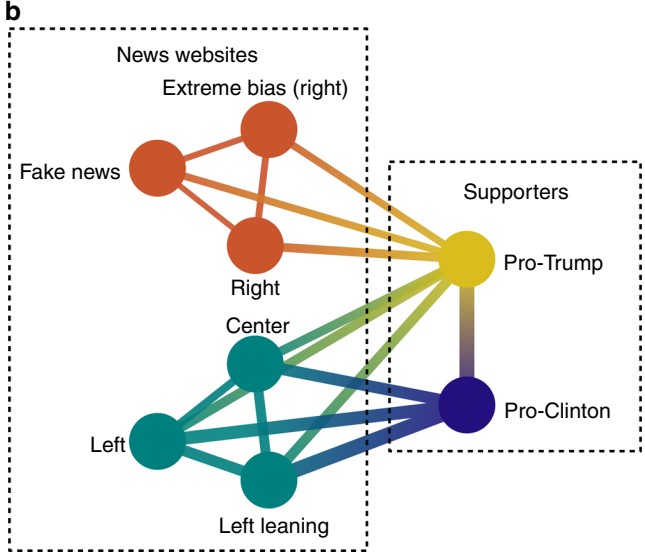

**Fig. 5** Activity correlation between news outlets and supporters. **a** Pearson cross-correlation coefficients between activity time series related to the different types of news outlets, Trump supporters and Clinton supporters. **b** Graph showing the correlation relations between the types of news websites and the supporters. The edges of the graph represent correlations larger than $r_0 = 0.49$. Fake news, extreme bias (right), and right websites form a first cluster, indicated by a red square in **a** and shown in orange in **b**, while center, left leaning, and left news websites form a second cluster, indicated by a blue square in **a** and shown in blue in **b**. The activity of Trump supporters is equally correlated with all news sources and the activity of Clinton supporters, which represents the largest activity, is mainly correlated with the second media cluster and only poorly with the first one

We consider only the activity of the top 100 news spreaders since, by definition of CI, they are the most important sources of information. Therefore, within the limitation of considering Twitter as a closed system, they are the most likely set of users to trigger the activity of the rest of the population. We test this hypothesis with Granger causal modeling.

Our causal analysis takes into account self-links, i.e. the auto-correlation of each time series, and reveals that they are the strongest causal effect for all time series. Since we are interested in the cross-links, we leave the self-links aside for the rest of the

discussion. The center and left leaning news spreaders have the strongest causation on the supporters activity, with a stronger effect on the Clinton supporters than on the Trump supporters (Table 4 and Fig. 6c). Since the Clinton supporters dominate Twitter activity, they also are the main drivers of the global activity. The other top news spreaders have only a small or negligible effect on the supporters activity. In particular, extreme bias (left), left, right leaning, and right news spreaders are more influenced by the activity of Clinton and Trump supporters than the opposite. We also observe that Trump supporters have a significant causal effect on the fake news spreaders' activity and Clinton supporters have a significant effect on extreme bias (left) spreaders' activity (Fig. 6c). This suggests that they are in fact following Twitter activity rather than driving it. Regarding the causal relations in-between news spreaders, center news spreaders are the most central driver as they are among the top three drivers of all news spreaders except for fake news (Table 4). Strong mutual causal effects are revealed between center and left leaning spreaders. Right leaning top spreaders are driving the activity of the right, extremely biased (right) and fake news spreaders. The two supporter groups have also strong mutual causal effects.

These results reveal two very different dynamics of news diffusion for traditional, center and left leaning, news and misinformation. Center and left leaning news spreaders are the most influential and are driving the supporters activity. On the other hand, the dynamics of fake news spreaders seems to be governed by the ensemble of Trump supporters.

The interpretation of the discovered causal effects must be understood within the limitation that we do not measure the diffusion of news outside of Twitter. Indeed, the reason why center and left leaning news spreaders have a causal effect on the Clinton supporters could be explained by the fact that they are the first to be "activated" by some news appearing, for example on television, while the supporters take more time to be "activated" by the same news. However, we have other indications that the news spreaders are directly causing at least part of the supporters' activity, namely that the top news spreaders are precisely the most important source of news retweets. Moreover, if the external driver is an other media outside of Twitter and that the center/left leaning news spreaders, who are almost all journalists, are the first to be activated, it is very likely that the media channel outside of Twitter is related to the journalists. In this case, even if the causation is indirect, we still identify the correct driver through the affiliation of the journalists. More importantly, while we observe a strong causal effect between center/left leaning news spreaders and the supporters, we do not observe a significant causal effect between other news spreaders and the supporters. This indicates that, even if the causal driver could be outside of Twitter, the diffusion mechanisms of traditional and fake news are very likely different.

We investigate the influence of the presence of staff members of the candidates' teams in Supplementary Note 2, Supplementary Fig. 4 and Supplementary Table 11. We observe no significant changes in the causal relations after having removed all users linked to the campaigns. We also repeated our analysis after having removed news aggregators from our dataset (see Supplementary Note 3, Supplementary Fig. 5, Supplementary Tables 12 and 13) and found that news aggregators are not responsible for the observed differences in dynamics.

## Discussion

Using a dataset of tweets collected during the 5 months preceding the 2016 presidential elections, we investigated the spread of content classified as fake news and compared its importance and influence with traditional, fact-based, media. We find that fake

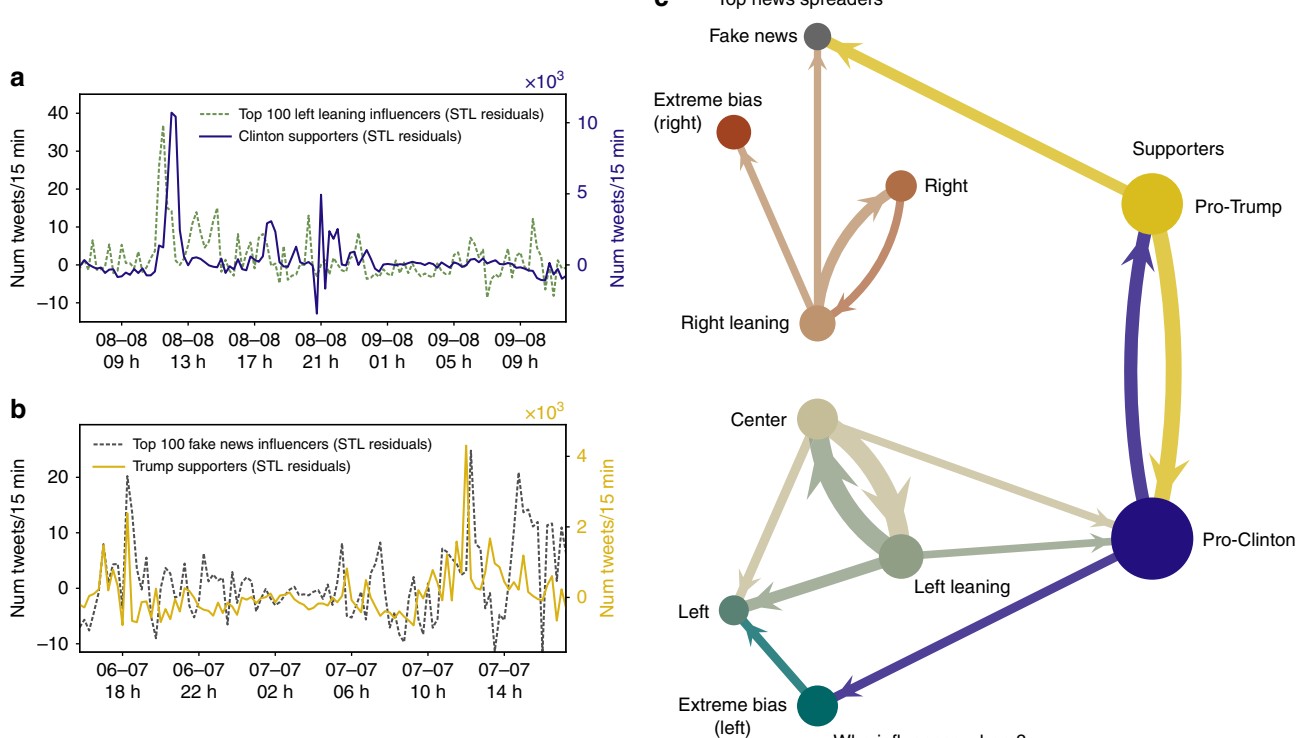

**Fig. 6** Granger causal network reconstruction between top news spreaders and supporters activity. **a** Activity time series corresponding to the top 100 left leaning news spreaders (dashed) and the Clinton supporters (continuous, right vertical axis). **b** Activity time series of the top 100 fake news spreaders (dashed) and the Trump supporters (continuous, right vertical axis). We show the residuals of the STL filtering after the removal of the seasonal (daily) and trend components. A causal effect seems apparent from the top 100 left leaning news spreaders to the Clinton supporters (**a**). Peaks in the left leaning news spreaders activity (yellow, dashed) tend to precede peaks in the activity of Clinton supporters (blue). A causal effect relation from the Trump supporters to the top 100 fake news news spreaders (**b**) seems also apparent. **c** Graph showing the maximal causal effects between the activity of the top 100 news spreaders of each media category (left) and the activity of the presidential candidate supporters (right) computed over the entire 5 months. Arrows indicate the direction of a the maximal causal effect (>0.05) between two activity time series. The width of each arrow is proportional to the strength of the causation and the size of each node is proportional to the auto-correlation of each time series. The center and left leaning top news spreaders are the news spreaders that show the strongest causal effect on the supporters activity. The values of the causal effects between each activity time series are shown in Table 4

news represents 10% and extremely biased news 15% of the tweets linking to a news outlet media. However, taking into account the difference in user activity decreases the share of fake and extremely biased news to 12%. Although we find approximately the same ratio of users using automated Twitter clients in each media category, we find that automated accounts diffusing fake news are much more active than the automated accounts diffusing other types of news. This results confirms the role of bots in the diffusion of fake news, which has been shown using a different method of bot detection[26], and shows that automated accounts also play a role, although smaller, in the diffusion of traditional news.

We analyzed the structure of the information diffusion network of each category of news and found that fake and extremely biased (right) news diffusion networks are more densely connected, i.e. users retweet more people and are more retweeted in average, and have less heterogeneous connectivity distributions than traditional, center, and left leaning, news diffusion networks. The heterogeneity of the degree distribution is known to play an important role in spreading processes on networks[29,30]. Spreading in networks with heterogeneous connectivity usually follows a hierarchical dynamics in which the information propagates from higher-degree to lower-degree classes[30].

We discovered the top news spreaders of each type of news by computing their Collective Influence[32] and found very different

profiles of fake and extremely biased news top spreaders compared to traditional news spreaders. While traditional news spreaders are mostly journalists with verified Twitter accounts, fake and extremely biased news top spreaders include unverified accounts with seemingly deceiving profiles and deleted accounts.

Analyzing the Twitter activity dynamics of the news diffusion corresponding to each media category, we reveal the existence of two main clusters of media in term of activity correlation which is consistent with the findings of previous works[4–9] that revealed the separation in polarized communities of online social media news consumers. We also show that right news media outlets are clustered together with fake news. Finally, a causality analysis between the top news spreaders activity and the activity of presidential candidate supporters revealed that the top news spreaders of center and left leaning news outlets are the ones driving Twitter activity while top news spreaders of fake news are in fact following Twitter activity, particularly Trump supporters activity.

Our analysis focuses on news concerning the candidate of the presidential election published from the most popular news outlets and therefore its results cannot be directly generalized to the entire Twitter population. Nevertheless, our investigation provides new insights into the dynamics of news diffusion in Twitter. Namely, our results suggests that fake and extremely biased news are governed by a different diffusion mechanisms than traditional center and left leaning news. Center and left

**Table 4 Causal effects between the top spreaders and the candidates supporters**

| ✎ | Pro-Clinton | Pro-Trump | Fake news | Extreme bias (right) | Right |
|---|---|---|---|---|---|
| Pro-Clinton | 0.65 ± 0.01 | **0.14 ± 0.01** | 0.029 ± 0.007 | 0.021 ± 0.006 | 0.002 ± 0.006 |
| Pro-Trump | **0.11 ± 0.02** | 0.46 ± 0.01 | 0.009 ± 0.006 | 0.003 ± 0.001 | 0.0014 ± 0.0009 |
| Fake news | 0.015 ± 0.003 | **0.10 ± 0.01** | 0.14 ± 0.01 | **0.05 ± 0.01** | 0.03 ± 0.01 |
| Extreme bias (right) | 0.02 ± 0.01 | 0.009 ± 0.002 | 0.03 ± 0.01 | 0.21 ± 0.01 | **0.04 ± 0.01** |
| Right | 0.009 ± 0.002 | 0.025 ± 0.008 | 0.03 ± 0.01 | 0.02 ± 0.01 | 0.18 ± 0.01 |
| Right leaning | 0.018 ± 0.008 | **0.038 ± 0.008** | 0.02 ± 0.01 | 0.01 ± 0.01 | **0.07 ± 0.01** |
| Center | **0.04 ± 0.01** | 0.023 ± 0.010 | 0.021 ± 0.007 | 0.0020 ± 0.0007 | 0.009 ± 0.008 |
| Left leaning | **0.04 ± 0.01** | 0.015 ± 0.006 | 0.003 ± 0.001 | 0.0010 ± 0.0005 | 0.009 ± 0.007 |
| Left | 0.03 ± 0.01 | 0.03 ± 0.01 | 0.010 ± 0.008 | 0.002 ± 0.001 | 0.01 ± 0.01 |
| Extreme bias (left) | **0.08 ± 0.01** | 0.03 ± 0.02 | 0.031 ± 0.009 | 0.03 ± 0.01 | 0.0025 ± 0.0008 |

| ✎ | Right leaning | Center | Left leaning | Left | Extreme bias (left) |
|---|---|---|---|---|---|
| Pro-Clinton | 0.003 ± 0.001 | **0.065 ± 0.008** | **0.062 ± 0.008** | 0.017 ± 0.009 | 0.006 ± 0.006 |
| Pro-Trump | 0.0020 ± 0.0009 | **0.038 ± 0.006** | **0.033 ± 0.008** | 0.020 ± 0.007 | 0.015 ± 0.006 |
| Fake news | **0.06 ± 0.01** | 0.037 ± 0.009 | 0.016 ± 0.002 | 0.014 ± 0.008 | 0.022 ± 0.009 |
| Extreme bias (right) | **0.06 ± 0.01** | **0.039 ± 0.009** | 0.018 ± 0.002 | 0.026 ± 0.009 | 0.027 ± 0.009 |
| Right | **0.09 ± 0.01** | **0.044 ± 0.009** | 0.016 ± 0.002 | 0.0026 ± 0.0009 | **0.033 ± 0.008** |
| Right leaning | 0.22 ± 0.01 | **0.042 ± 0.009** | 0.033 ± 0.009 | 0.0014 ± 0.0008 | 0.0027 ± 0.0008 |
| Center | 0.012 ± 0.010 | 0.266 ± 0.009 | **0.18 ± 0.01** | 0.019 ± 0.010 | 0.013 ± 0.008 |
| Left leaning | 0.005 ± 0.003 | **0.18 ± 0.01** | 0.299 ± 0.009 | 0.012 ± 0.008 | 0.003 ± 0.002 |
| Left | 0.015 ± 0.008 | **0.08 ± 0.01** | **0.10 ± 0.01** | 0.164 ± 0.010 | **0.07 ± 0.01** |
| Extreme bias (left) | 0.005 ± 0.009 | **0.034 ± 0.009** | **0.045 ± 0.009** | 0.03 ± 0.01 | 0.27 ± 0.01 |

We show the value of the maximal causal effect $I_{i \to j}^{CE,max} = max_{0 < \tau \leq \tau_{max}} \left| I_{i \to j}^{CE}(\tau) \right|$ between each pair $(i, j)$ of activity time series, where $\tau_{max} = 18 \times 15$ min = 4.5 h is the maximal time lag considered, with standard errors (s.d., see Methods). The arrows indicate the direction of the causal effect. For each activity time series, we indicate in bold the three most important drivers of activity (excluding themselves)

leaning news diffusion is driven by a small number of influential users, mainly journalists, and follow a diffusion cascade in a network with heterogeneous degree distribution which is typical of diffusion in social networks[30], while the diffusion of fake and extremely biased news seems to not be controlled by a small set of influencers but rather to take place in more connected clusters and to be the result of a collective behavior.

## Methods

**Twitter data collection and processing**. We collected tweets continuously using the Twitter Search API from 1 June 2016 to 8 November 2016. We gather a total of 171 million tweets in the English language, mentioning the two top candidates from the Republican Party (Donald J. Trump) and Democratic Party (Hillary Clinton) by using two different queries with the following keywords: hillary OR clinton OR hillaryclinton and trump OR realdonaldtrump OR donaldtrump.

We extracted the URLs from tweets by using the expanded_url field attached to each tweet containing at least one URL. A large number of URL were redirecting links using URL shortening services (e.g. bit.ly, dlvr.it, or ift.tt). News websites sometimes also uses shortened versions of their hostnames (e.g. cnn.it, nyti.ms, hill.cm, or politi.co). We programmatically resolved shortened URLs, using the Python Requests library, in order to find their final destination URL and extracted the hostname of each final URL in our dataset.

To identify tweets that may originate from bots, we extract the name of the Twitter client used to post each tweet from their source field and kept only tweets originating from an official twitter client. Third-party clients represents a variety of applications, form applications mainly used by professional for automating some tasks (e.g. sprinklr.com or dlvrit.com) to manually programmed bots, and are used to post ≤8% of the total number of tweets. When a programmatic access to Twitter is gained through its API to send tweets, the value of the *source* field of automated tweets corresponds to the name, which must be unique, given to the "App" during the creation of access tokens. Supplementary Table 14 shows the clients we consider as official and the corresponding number of tweets with URLs originating from each client. The number of tweets with a URL originating from official clients represents 82% of the total number of tweets with a URL. This simple method allows to identify tweets that have not been automated and scales very easily to large datasets contrary to more sophisticated methods[39]. Indeed, Botometer is not well suited for historical data as it requires several tweets per users (up to 200) and results of a Twitter search of tweets (up to 100) mentioning each users, which we cannot do retroactively. We compared our method with the results of Botometer (see Methods section of ref. [27]) and found that our method has a good accuracy but suffer from a relatively high number of false positive compared to Botometer. Advanced bots might not be detected by our method, but this is also a problem for more advanced methods that relies on a training set of known bots[39]. We remove

all tweets sent from non-official clients when computing the activity of supporters but we keep them when building the retweet networks, as we want to include automated accounts that play a role in the diffusion of news.

**News outlets classification**. Among the 55 million tweets with URLs linking outside of Twitter, we identified tweets directing to websites containing fake news by matching the URLs' hostname with a curated list of websites, which, in the judgment of a media and communication research team headed by Melissa Zimdars of Merrimack College, USA, are either fake, false, conspiratorial, or misleading. The list, freely available at www.opensources.co, classifies websites in several categories, such as "Fake News", "Satire", or "Junk Science". For our study, we construct two non-overlapping set of websites: fake news websites and extremely biased websites. The set of fake news website is constructed by joining the hostnames listed under the categories "Fake News" and "Conspiracy Theory" by www.opensources.co. The following definitions of these two categories are given at www.opensources.co

- "Fake News": sources that entirely fabricate information, disseminate deceptive content, or grossly distort actual news reports,
- "Conspiracy Theory": sources that are well-known promoters of kooky conspiracy theories.

The set of extremely biased websites contains hostnames appearing in the category "Extreme Bias" (defined as sources that come from a particular point of view and may rely on propaganda, decontextualized information, and opinions distorted as facts by www.opensources.co) but not in any of the categories used to construct the set of fake news. Hostnames in each categories along with the number of tweets with a URL pointing toward them are reported in Supplementary Table 1. We discard insignificant outlets accumulating less than 1% of the total number of tweets in their category.

Websites classified in the extremely biased (right) category, respectively extremely biased (left) category, have a ranking between right bias and extreme right bias, respectively left and extreme left, on mediabiasfactcheck.com. The bias ranking on www.allsides.com of these same websites is right, respectively left (corresponding to the most biased categories of www.allsides.com). The website mediabiasfactcheck.com also reports a level of factual reporting for each websites and we find that all the websites classified in the extremely bias category have a level of factual reporting which is mixed or worse. We also find that all the websites remaining in the fake news category have a bias between right and extreme right on mediabiasfactcheck.com. The website www.allsides.com rates media bias using a combination of several methods such as blind surveys, community feedback, and independent research (see www.allsides.com/media-bias/media-bias-rating-methods for a detailed explanation of the media bias rating methodology used by AllSides), and mediabiasfactcheck.com scores media bias by evaluating wording, sourcing, and story choices as well as political endorsement (see

mediabiasfactcheck.com/methodology for an explanation of Media Bias Fact Check methodology).

A potential issue with the methodology of OpenSources is the blurring of the assessment of "bias," which has to do with news content, with the assessment of "establishment", which has to do with news form. Specifically, their steps 4–6 indicate they count thinks like use of the Associated Press style guide and the production quality of the website. These criteria thus conflate adherence to establishment norms —which are likely to be correlated with things like budgets for professional design, fact-checking, editorial oversight etc.—with lack of bias. That is, if two media sites present the same news, but one does it in a less established format, it may be considered "extremely biased." For this reason, we manually reassessed the bias of each website in the extreme bias categories on mediabiasfactcheck.com and allsides.com to validate their bias, as these two websites do not list the rejection of the establishment as a criteria for their bias assessment. However, even if we do not use the criteria of adherence or rejection of the establishment in our classification, websites in the extreme bias (right) and extreme bias (left) categories are more likely to not adhere with the establishment as this variable seems to be highly correlated with political bias.

In order to validate our classification, we compare it to the domain-level ideological alignment scores of news outlets obtained by Bakshy et al.[22] which is based on the average self declared ideological alignment of Facebook users sharing URLs directing to news outlets. We find a $R^2 = 0.9$ for the linear regression between the ideological alignment found by Bakshy et al. and our classification where we mapped our categories between $-3$ and 3 (see Supplementary Fig. 6). Supplementary Data file SuppData_top_urls_per_category.csv contains the top 10 URLs of each media category along with notes about their classification on fact-checking websites (when available), links to the fact-checking websites, and additional information. We observe that the classification of the most popular URLs is well aligned with the label assigned to their domains.

We investigate the influence and importance of news at the domain level and not at the article level. Since a website classified as fake may contain factual articles and vice versa, domain-level classification implies a level of imprecision. However, it allows us to reveal the integrated effect of news outlets over more than 5 months and to measure the relative importance of each type of news by classifying all URLs directing to important news outlets. Moreover, classifying domains instead of URL (or article) allows to consider the extended effect of each type of news. Indeed, when a Twitter user follows a URL to a news article containing factual information on a website publishing mostly fake news, she/he will be exposed to the other articles containing fake news on the websites. Therefore, this particular fact-based news ultimately increases the potential influence of fake news.

**Collective influence algorithm in directed networks**. We use the CI algorithm[32] applied to directed networks to find the most influential nodes of the information retweet networks. The CI algorithm is based on the solution of the optimal percolation of random networks which consists of identifying the minimal set of nodes, the super-spreaders, whose removal would dismember the network in many disconnected and non-extensive components. The fragmentation of the network is measured by the size of the largest connected component, called the giant component of the network. The CI algorithm considers influence as an emergent collective property, not as a local property such as the node's degree, and has been shown to be able to identify super-spreaders of information in social networks[40,41]. Here, we consider a directed version of the algorithm where we target the super-sources of information.

The procedure is as follows[40]: we first compute the value of $CI_{\ell,out}(i)$ for all nodes $i = 1, \dots, N$ as

$$CI_{\ell,out}(i) = (k_{out}(i) - 1) \sum_{\substack{j \in \partial B_{out}(i,\ell) \\ k_{out}(j) > 0}} (k_{out}(j) - 1), \qquad (1)$$

where $\ell$ is the radius of the ball around each node we consider, here we use $\ell = 2$, $k_{out}(i)$ is the out-degree of node $i$, and $\partial B_{out}(i,\ell)$ is the set of nodes situated at a distance $\ell$ from node $i$ computed by following outgoing paths from $i$. The node with the largest $CI_{\ell,out}$ value is then removed from the network and the value of $CI_{\ell,out}$ of nodes whose value is changed by this removal is recomputed. This procedure is repeated until the size of the weakly connected largest component becomes negligible. The order of removal of the nodes corresponds to the final ranking of the network top news spreaders shown in Table 3.

A comparison of the ranking obtained by the CI algorithm with rankings obtained by considering out-degree (high degree centrality) and Katz centrality[42] (Supplementary Fig. 7) shows that high degree (HD) and Katz rankings of the top 100 CI spreaders fall mostly within the top 100 ranks of these two other measures with only a small number of top CI spreaders having a poor HD or Katz ranking. Note that the CI algorithm is especially good at identifying influential nodes that are locally weakly connected but are influent on a larger scale[32].

**Time series processing**. We find that a 15 min resolution offers a sufficiently detailed sampling of Twitter activity. Indeed, a representative time scale of Twitter activity is given by the characteristic retweet delay time, i.e. the typical time between an original tweet and its retweet. We find that the median time of the

retweet delay distribution in our dataset is 1 h 57 min and the distribution has a log-normal shape (first quartile at 20 min and third quartile at 9 h 11 min). We tested the consistency of our results using a resolution of 5 min and 1 h and did not see significant changes.

In order to perform the cross-correlation and causality analysis of the activity time series, we processed the time series to remove the trend and circadian activity cycles and to deal with missing data points. For each missing data points, we remove the entire day corresponding to the missing observation in order to keep the period of the circadian activity consistent over the entire time series. This is necessary to apply filtering technique to remove the periodic component of the time series. When removing an entire day, we consider that the day starts and ends at 4 a.m., corresponding to the time of the day with lowest Twitter activity. We removed a total of 24 days, representing 15% of our observation period. We then applied an STL (seasonal-trend decomposition procedure based on Loess)[33] procedure to extract the trend, seasonal and remainder components of each activity time series. We only consider the remainder components for the cross-correlation and causality analysis. We set the seasonal period of the STL filter equal to the number of observations per day, $n_p = 96$, and the seasonal smoothing period to $n_s = 95$, such that the seasonal component is smooth and the remainder component retains the higher frequency signal containing the activity of interest. Varying the value of the smoothing period to $n_s = 47$ does not change significantly the results.

**Causal analysis**. The STL procedure removes the trend and circadian pattern in the time series, resulting in stationary time series (the stationarity of each time series is confirmed by an augmented Dickey–Fuller test[43]). Before performing the causal analysis, we also standardized each time series in order to remove any influence of the difference in absolute values of time series. The causal analysis is performed using the entire time period (more than 5 months) and therefore reveals causal effects that are observed "in average" over the entire time period.

In order to infer the causal relations between the activity of the top news spreaders and the supporters, we use a multivariate causal discovery algorithm based on the PC algorithm[35] and further adapted for multivariate time series by Runge et al.[28,36,44]. Considering an ensemble of stochastic processes $\mathbf{X}$ the algorithm proceeds as follows. First, for every time series $Y \in \mathbf{X}$ the sets of preliminary parents is constructed by testing their independence at a range of time lags: $\mathcal{P}_{Y_t} = \{X_{t-\tau} | 0 < \tau \leq \tau_{max}, Y_t \not\perp X_{t-\tau}\}$. As this set also contains indirect links, they are then removed by testing if the dependence between $Y_t$ and each $X_\tau \in \mathcal{P}_{Y_t}$ vanishes when it is conditioned on an incrementally increased set of conditions $\mathcal{P}_{Y_t}^{n,i} \subseteq \mathcal{P}_{Y_t}$, where $n$ is the cardinality of $\mathcal{P}_{Y_t}^{n,i}$ and $i$ is the index iterating over the number of combinations of picking $n$ conditions from $\mathcal{P}_{Y_t}$. The combinations of parents having the strongest dependence in the previous step are selected first[28,44].

The main free parameters are the maximum time lag $\tau_{max}$ and the significance level of the independence test used during the first step to build the set of preliminary parents which we set to $\alpha_{PC} = 0.1$. We set the value of the maximum time lag to $\tau_{max} = 18$ time steps (i.e. 270 min) as it is the lag after which the lagged cross-correlations between each time series falls below 0.1 in absolute value (see Supplementary Figs. 8–11). We set the maximum number of tested combinations of the conditioning set to 3 and we do not limit the size of the conditioning set.

We test the conditional independence of time series with the non-parametric RCoT test[38]. This test uses random Fourier features to approximate the kernel-based conditional independence test KCIT[37] and is at least as accurate as KCIT while having a run time that scales linearly with sample size[38]. This point is crucial for our case given the size of our dataset (13,152 time points × 10 time series × 18 time lags). We set the number of Fourier features to $n_f = 400$.

We select the significant final causal links by applying a Benjamini–Hochberg False Discovery Rate (FDR) correction[45] to the $p$-values of the conditional independence tests with a threshold level of 0.05. FDR corrections allow to control the expected proportion of false positive. The final causal links, i.e. parents of each time series, are reported in Supplementary Table 15.

Following the procedure of refs. [28,46], We then regress a linear model:

$$\mathbf{X}_t = \sum_{\tau=1}^{\tau_{max}} \mathbf{\Phi}(\tau)\mathbf{X}_{t-\tau} + \varepsilon_t, \qquad (2)$$

where all time series are standardized and only coefficients corresponding to true causal links are estimated while all the other ones are kept equal to zero, i.e. $\mathbf{\Phi}_{ij}(\tau) \neq 0$ only for $X_{t-\tau}^i \to X_t^j$. The causal effect between a time series $X^i$ and $X^j$ at a time delay $\tau$ can be computed from the regressed coefficients as

$$I_{i \to j}^{CE}(\tau) = \mathbf{\Psi}_{ij}(\tau), \qquad (3)$$

where $\mathbf{\Psi}(\tau)$ is computed from the relation $\mathbf{\Psi}(\tau) = \sum_{s=1}^{\tau} \mathbf{\Phi}(s)\mathbf{\Psi}(\tau - s)$, with $\mathbf{\Phi}(0) = \mathbf{I}$. Here, $\mathbf{\Psi}_{ij}(\tau)$ gives the sum over the products of path coefficients along all causal paths up to a time lag $\tau$. The causal effect $I_{i \to j}^{CE}(\tau)$ represents the expected value of $X_t^j$ (in unit of standard deviation) if $X_{t-\tau}^i$ is perturbed by one standard deviation[28].

To reconstruct the causal network, we are interested in the aggregated effects and therefore use the lag with maximum effect:

$$I_{i \to j}^{CE,max} = \max_{0 < \tau \leq \tau_{max}} \left| I_{i \to j}^{CE}(\tau) \right|. \qquad (4)$$

We estimate the standard errors of each causal effects with a residual-based bootstrap procedure (similarly to ref. [28]). We employ 200 bootstrap surrogates time series generated by running model (2) with a joint random sample $\varepsilon_t^*$ (with replacement) of the original multivariate residual time series $\varepsilon_t$ and compute the standard deviation of the $I_{i \to j}^{CE,max}$ values.

**Code availability**. The analysis and plotting scripts allowing to reproduce the results of this paper are available at https://github.com/alexbovet/information_diffusion. The Python module used for the network analysis (graph-tool) is available at https://graph-tool.skewed.de. The causal discovery algorithm software (TIGRAMITE) is available at https://jakobrunge.github.io/tigramite. The code for the conditional independence test (RCIT and RCoT software) is available at https://github.com/ericstrobl/RCIT. The code for the LOESS processing is available at https://github.com/jcrotinger/pyloess.

## Data availability

The raw Twitter data cannot be directly shared as it would infringe the Twitter Developer Terms. However, we are sharing the tweet IDs of the data we collected which allows anyone to download the tweets used for this study directly from Twitter using Twitter's API. The datasets analyzed in this study are available under the limits of Twitter's Developer Terms at http://kcore-analytics.com. The classification of news as "fake" news or "extremely biased" news is a matter of opinion, rather than a statement of fact. This opinion originated in publically available datasets from fact-checking organizations (i.e. www.opensources.co). The conclusions contained in this article should not be interpreted as representing those of the authors.

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

## Acknowledgements

A.B. thanks the Swiss National Science Foundation (SNSF project P2ELP2_165158) and the Flagship European Research Area Network (FLAG-ERA) Joint Transnational

Call "FuturICT 2.0" for the financial support provided and R. Lambiotte for helpful comments.

## Author contributions

H.A.M. and A.B. conceived the project and wrote the manuscript. A.B. performed the analysis and prepared figures.

## Additional information

**Competing interests:** H.A.M. has shares in KCore Analytics, LLC. The remaining author declares no competing interests.

