## [Peer Review File · Nature Communications]

Reviewers' comments:

Reviewer #1 (Remarks to the Author):

I appreciate this work in a number of ways. The authors do a thorough, systematic analysis of an important phenomenon using reproducible methods. The dataset is thoughtfully collected, and so in some sense any systematic analysis of these data are helpful to the public interest because this topic is so important and the relevant data publicly available so sparse.

That said, I have some pretty substantial concerns about the interpretation of the results. As is often the case in big data analysis, the authors have refrained from making explicit claims about their findings, preferring to primarily "describe" patterns and indicate what they might suggest. But this strategy is not really adequate for explaining empirical findings about the social world. Basically, no matter the caveats or hedges the authors provide, their findings will be interpreted by other researchers, and the public and media, as discovering things about what happened in the 2016 election. This audience enthusiasm for over-interpreting the authors' work puts a burden on them to be explicit about what they found, and, in particular, to consider alternative explanations and to test for their operation. As I will show in detail below, the authors have not done this but in many cases could do so quite easily. This burden may seem unfair, but it is only the flip side of the positive attention and enthusiasm that the work generates because of its subject matter. In other words, it is interesting because of its immediate social significance, and the authors can focus on a single case and comparison (2016, U.S. Trump vs. Clinton), because of this significance, so they should in turn bear the burden of treading carefully about this phenomenon.

In keeping with this, the form of my primary concerns will be that the authors suggest or imply an interpretation of the results which, based on the analyses presented in the ms, is not justified because there is another, alternative, they have not considered. I present 5 of these below. Following these I also point out some smaller issues with the paper.

Concern 1

Authors imply: Granger-causal analyses reflect properties of political audiences (e.g. Trump supporters, Concern vs. Clinton supporters, Liberals)

Alternative explanation: Granger-causal analyses reflect Trump campaign's active and successful use of social media as a central tool of their campaign.

The analysis in section 1.4, in particular Figure 5 and 6, seem to tell the classic story of media polarization and filter bubbles, an interpretation the authors endorse in the discussion. However, this interpretation fails to account for the fact that a substantial portion of the "influencers" in each of the fake, extreme bias, and conservative news areas are Trump and members of his own campaign. This direct influence can be observed in Table 3. Trump himself is in the top 6 in fake news, extremely biased news, and right news. His surrogates (his son, Kellyanne Conway—top advisor and eventual campaign manager; Breitbart news--run by his eventual campaign chief executive, Steve Bannon) are also among the top influencers in both the fake/extremely biased news groups (Table 3).

This raises the question of what the "polarization" and specific preferences observed in Figures 5 and 6 is actually showing. First, the polarization might be showing that Americans with different political loyalties prefer different news sources, something which appears to be increasing over time. OR, it might be showing that Americans of different political loyalties prefer to listen to the candidates they support to get information, something which is not novel and is to be expected. In this second interpretation, the polarization in news sources comes from the choices that Trump and his campaign made. Trump chose different media, so his supporters retweeted different media. If Trump had chosen cat videos, then his supporters would have retweeted cat videos. The polarization is descriptively still real but any attempt to generalize it to new elections, or to

compare it with findings from other studies, would need to take this into context.

The same logic must be considered with respect to the kind of news they is shared. The authors imply that it is conservative audiences, or Trump supporters, who prefer “fake news” and “extremely biased” news. This is descriptively true, but it implies that this is feature of their political appetite or epistemology, such as the finding that conspiracy theory seeking seems to be a stable trait as in Bessi et al. (2015). But the explanation might simply be what we already expect, which is that they prefer the news that their candidate prefers, especially as an election approaches and they are engaged in actively try to promote their candidate so that they prevail in the competition (Margolin, Hannak, & Weber, 2018). They appear to like “fake” news, but really they like the news Trump shares, and he happens to like fake news, unsurprisingly, because it is favorable to him. This is campaigning as normal with the exception of Trump’s willingness to forego any concern about what is true or false.

To be clear, my argument is not that this is the explanation, only that it is equally plausible, and substantially different in implication, than what the authors have presented. Moreover, their analysis could consider it. They should, at the very least, remove campaign officials from the “influencer” lists and re-run the analyses, possibly even comparing them to see “how much” of the pattern comes from the campaign. In the absence of this analysis, any claim that attempts to attribute tweeting behavior to audiences preferences or habits, beyond the fact that they like what their candidate likes, is not justified.

Furthermore, the goal of the analysis is not to exonerate any particular individual for sharing fake news, but to use the available evidence to identify the root cause. On one level, it is relevant that Trump supporters shared X quantity of fake news. But the value of this kind of analysis is not in its ability to cast blame between sides but to understand how behavior come about to better understand future cases. In this context, it is crucial to know whether Trump supporters shared fake news because they prefer fake news or because they prefer Trump’s news and he preferred fake news.

Concern 2

Authors imply: Trump supporters/conservatives prefer “extreme bias” news more than Clinton supporters/liberals.

Alternative explanation: A single website, Breitbart News, accounts for the bulk of conservative over liberal preference for extreme bias. Moreover, This website is preferred because it is associated with the Trump campaign

According to Table S1, Breitbart News is by far the most dominant in the non-traditional or right wing space, with 1.8M shares – almost 2x or more than any nearest competitor in any other category here. They are also closely aligned with the Trump campaign, as their former leader (Steve Bannon) eventually joined Trump’s campaign as its chairman. Thus, throughout the campaign it would be fair to view Breitbart as promoting Trump, in particular through the highly contested GOP primary. They are almost a campaign surrogate in this period. We would thus expect that Trump supporters would be heavily engaged with this website.

This descriptive fact is important, and the authors provide evidence of just how influential Breitbart was, a potentially important contribution. But instead of the analysis focusing on this one unique case (highly popular, tied to a specific campaign), they put Breitbart in with all other “extreme bias” websites, allowing Breitbart’s very unique role to accrue to a general category with much farther implications. Put in another way, the distinguishing feature of this category is “extreme bias,” but this is not necessarily the functioning feature of Breitbart. Equally, if not more plausibly, the functioning feature is alignment with the ascendant candidate.

The authors should at least remove Breitbart from this category, possibly treat them as their own category, or otherwise link them with the campaign. Other websites that seem to work for the

campaign should also be identified and removed in this way.

Concern 3

Authors imply: Trump supporters/conservatives prefer “extreme bias” news more than Clinton supporters/liberals.

Alternative explanation: Upstart conservative websites, like Breitbart, are categorized as extremely biased rather than “right” because of deviance in form, not content.

The use of OpenSources raises another complication, namely, the blurring of the assessment of “bias,” which has to do with news content, with the assessment of “establishment”, which has to do with news form. The 2016 primaries showed a fissure on the American right. In that fissure, one side was the establishment (e.g. Jeb Bush) and the other anti-establishment. Some candidates (Trump) used foul language (anti-establishment form), others tried to play by the rules. Their positions were no more or less conservative. Trump was more conservative on immigration but less conservative on entitlements and infrastructure, for example. More broadly, norm following is not related to “bias” on a left-right spectrum. In 1992, Bill Clinton was the upstart Democrat in that he was more conservative (i.e., less extreme) than the party had preferred in recent years. Bias and establishment thus do not inherently tie together.

However, inspection of OpenSources coding methodology (from the web page shared in the paper) shows that they do, quite deliberately, conflate these (perhaps for valid reasons for their own research purposes). They use two broad criteria to assign their categories: 1) detection of bias; 2) detection of professionalism / lack of professionalism. Specifically, their steps 4-6 indicate they count things like use of the Associated Press style guide and the production quality of the website. These criteria thus conflate adherence to establishment norms—which are likely to be correlated with things like budgets for professional design, fact-checking, editorial oversight etc.—with lack of bias. That is, if two media sites present the same news, but one does it in a less established format, it will be considered “extremely biased.”

This assessment basically bakes in the cake the finding that Trump supporters/conservative appear more biased, because the anti-establishment rupture in their party is coded as one of bias shift rather than source shift. We also see this in the paucity of sites characterized as “right” or “right leaning.” Obviously, if most conservative sites are extremely biased, rather than just “right,” then conservative readers will appear to prefer more biased news. Perhaps if Bernie Sanders had been more successful on the left we would see both sides preferring “extremely biased” news, but in both cases this would be misleading, because what they prefer is non-traditional news, because they find that the traditional news is biased in its suppression of alternative points of view.

I don’t have a solution here other than to, a) find another site or way of categorizing these news sources that does not conflate form and content, as they are very disruptive to this paper’s analysis; or remove “extreme bias” altogether from the main analysis. It seems inherently biased. I would also move Breitbart over to “right” and see how much things changed.

Concern 4

Authors imply: Fake and extremely biased news sites tend to be unverified, suggesting they are non-compliant with Twitter rules, trolls, or other shady characters

Alternative explanation: Twitter’s criteria for verification overlap with OpenSources categorizations of establishment categories (not extreme bias).

This analysis is described near Table 3. It implies that there is something in the Twitter behavior of the fake news / extreme bias promulgators that makes them stick out as unverified. But, as described above, OpenSources uses a site’s establishment bonafides to determine its bias/content.

It turns out that this is also closely related to Twitter's criterion for verification, which is that they must be "public interest" (from twitter policy -- <https://help.twitter.com/en/managing-your-account/about-twitter-verified-accounts>). In simple terms, large media, such as sites associated with television stations or other large entities, such as rt.com (financed by the Russian government), are more likely to count as "right" or "right leaning" rather than "fake" or "extreme," and they are also more likely to be verifiable by Twitter's policy. Upstarts appear unverifiable by policy and "fake" or "extreme."

In this case I think the verification analysis should just be removed. It adds a biasing context to the rest of the paper but is not really essential to the larger work.

Concern 5

Authors imply: Fake and extreme news sharing communities are more connected due to higher average degree

Alternative explanation: Regular news sharing communities have artificially lower average degree due to expanded endpoints (people with zero in-degree).

First, in Table 2, it's not clear how the average degree is calculated—based on sum of indegree and outdegree?

More broadly, there is a problem with calculating central tendency in node-level statistics, only. The authors acknowledge that the distribution of degree is likely to be skewed, but the problem is deeper than that. This is a retweet network, a catalogue of individual events, not a "social network" where there are stable relationships or obligations between nodes (like in a follower network). The key difference is in how to interpret the "endpoints" – the nodes with low out-degree and 0 in-degree. In a social network, they are members of the community who are hard to reach, possibly expanding the diameter of the network and so forth. But in the context of a retweet network, they are essentially "free" from a structural point of view—they can meaningfully describe the network but they don't imply aspects of its functional structure. The more central nodes don't owe them anything or know they are there.

More formally, if two networks began as identical, and then in only 1 some subset of nodes were retweeted by a new set of nodes who, themselves, sent only 1 retweet (outdegree = 1) and received no retweets (indegree = 0), the average degree in this network would fall. This is because it just "cheaply" accumulated a bunch of low degree nodes. But it is, in all other ways, functionally the same as the first.

Next, consider that larger networks will, on average, have more opportunities to accumulate these meaningless endpoints, and so large networks may have lower average degree when calculated in this way. Most "social" networks do not face this problem, because it is unusual to accumulate huge numbers of very lower degree nodes – they would be very odd individuals. But in the case of retweets this seems quite likely.

The solution is to compare the distributions of degree and other structural features rather than just provide averages. For example, plots of the degree distribution and Kolmogorov Smirnov tests could indicate whether the differences in average degree are observed throughout the network or really just in the endpoints. Also, the network among the top 100 influencers could be analyzed. This is apples to apples as no community distorted by having a different size. It's actually not clear why the entirety of the data are used for these analysis whereas only the top influencers were used in the network diagrams.

Smaller Issues

Fake news and extreme bias categories hard to distinguish

It's not clear why Hate News, which does not appear to be judged by any epistemic criteria, but does relate to biased reasoning, would be placed in the fake news category rather than the extreme bias category.

Drew Margolin

References

Bessi, A., Coletto, M., Davidescu, G. A., Scala, A., Caldarelli, G., & Quattrociocchi, W. (2015). Science vs Conspiracy: Collective Narratives in the Age of Misinformation. *PLOS ONE*, 10(2), e0118093. <https://doi.org/10.1371/journal.pone.0118093>

Margolin, D. B., Hannak, A., & Weber, I. (2018). Political Fact-Checking on Twitter: When Do Corrections Have an Effect? *Political Communication*, 35(2), 196–219. <https://doi.org/10.1080/10584609.2017.1334018>

Reviewer #2 (Remarks to the Author):

This is an interesting and thorough paper with a number of important results, particularly about asymmetries between left and right. I think those results could be foregrounded more clearly in various ways, particularly how they relate to the results (not just methods used) in previous research on fake news and partisan asymmetry on Twitter. The limitations of this project also need more foregrounding and discussion. The most problematic section is the analysis of so-called Granger "causality," which would benefit from a number of additional tests, some strong caveats, and most importantly, an exploration of the full flow ("causation") network among the various subpopulations examined.

Deferring the temporal/causal analysis for a moment, the main caveat for most of these results that should be foregrounded more in the paper is that this analysis mainly examines a very small subset of major media and other "influencers" on Twitter. While it is certainly the case that this subset accounts for the majority of URL shares, it is not at all therefore the case that the asymmetries the authors find in their behavior in sections 1.1 and 1.2 is generalizable to the sharing behavior of the many millions of more infrequent Twitter users. That's not a fundamental problem, just a matter of making that caveat clearer: that this doesn't tell us much about the differing behaviors among the millions of members of the mass public, just the media elite.

Section 1.4 does go more explicitly examine the mass public, but here the authors provide insufficient detail about how the "supporters" are calculated. How many are they? How elite are they, or how representative are they of all Twitter users (let alone the public at large)? How disjoint are they from the influencers? This may be discussed in further detail in the supplemental information, but it needs more attention in the main paper because the keystone result -- the lead-lag asymmetry between Clinton and Trump supporters relative to the fake and non-fake influencers -- depends on us believing that these "supporters" reveal the "influence of fake news" as opposed to just the intra-elite dynamics of fake news.

Before turning to the temporal analysis, a few isolated comments and questions about the first section:

- The authors suggest on page 1 that "homogeneous beliefs" are equivalent to "echo chambers," but surely this is more the exception than the rule, since for the most part everyone shares homogenous beliefs about innumerable things (eg, that the world is round) which does not imply an "echo chamber". The propagation of false or unverified information is separate from

homogeneity, though the two may be related.

- The operationalization of "extremely biased" was not discussed in enough detail in the main text, especially considering that this is the most controversial of the seven categories.
- More generally, the categorization of left and right among media is a challenging and controversial problem, and I would have liked more robustness checks here to make sure that the categories basically adopted whole-cloth from allsides.com and mediabiasfactcheck.com are robust to alternative specifications.
- The bot detection approach here seems relatively crude compared to the methods at Botometer, etc. Is there a good reason not to use existing state-of-the-art methods?
- Why do the authors restrict themselves to unweighted retweet networks in Section 1.2? The results in Table 2 might differ significantly with weighted networks.
- For the measurement of influence, the authors employ the Collective Influence algorithm. How do the inferred results differ from a more traditional measure like (inflow) eigenvector centrality, or even just degree?

Turning now to the temporal analysis, as I'm sure the authors are well aware, using "causal" language invokes a world of criticism that may not be necessary here. There are of course many mechanisms apart from direct causation that might give rise to various groups leading or lagging each other, such as differing attention to television or other non-Twitter media, or just different activity levels or reading frequency on Twitter. While retweets are clearly causal in the most narrow sense, all the other things that lead one to decide to retweet a seen message operate by many different paths, and most importantly, the fact that group A is usually faster off the mark to share some hot piece of news than group B does not mean that A causes or even influences B. This is a hard problem to solve and I don't expect the authors to solve it, but they need to be much more careful in their use of causal language, particularly since "granger causality" is at this point viewed with some skepticism in the causal inference community, to put it mildly. Most importantly, if this isn't (directly) causal, how do the alternate mechanisms affect our interpretation of these results?

Turning to more specifics:

- Regardless of how one deploys causal language here, it is essential to study bidirectional effects among all the groups, as the authors do. But this seems much more suited to a Vector Autoregressive framework (or other methods suited to studying bidirectional temporal data) than a series of paired auto-regressive comparisons, particularly if we want to examine the influence of each node conditional on the others.
- Relatedly, I would very much like to see the lead-lag "causal" network among influencers and between the supporters, and not just between influencers and supporters. It may be that the flows between these groups complicate the supporter asymmetries in important ways, and the fact that they do not form perfectly disjoint sets is a relative fixable problem.

The above two tasks are non-trivial revisions, but worth tackling, since the temporal aspect is the most novel part of the paper.

More minor points about the temporal analysis follow:

- Perhaps the most interesting result of the first section is the asymmetry revealed in Table 2: The higher degree within the fake, biased and right media, vs the higher heterogeneity among the

center and left. This intra-group analysis seems in tension with the results on page 11 and Figure 5, where the mass public on the right (Trump supporters) shows the greater heterogeneity in their temporal sharing patterns than the more homogeneous left. Is this due to the difference between the elite and the masses, or due to differences between analyzing an a-temporal follower networks vs the temporal retweet-correlation network?

- As discussed above, the authors mention on page 14 that "we assume that [the influencers] are the ones triggering the activity of the rest of the population." This is a very strong assumption and needs to be foregrounded much more, and its robustness explored.

- A p threshold of 10^{-7} seems somewhat arbitrary, and I would like to see actual evidence that these patterns are robust to different thresholding, since such thresholding can often make a large difference in converting continuous relation data to networks. At the very least, a reasonable False Discovery Rate correction would make the threshold seem less arbitrary.

Reviewer #3 (Remarks to the Author):

This paper investigates the timely problem of news sources' influence on Twitter during the last US presidential election. The authors track and collect millions of tweets that contain URLs to news articles with different political alignments and to misinformation websites. Its an interesting and timely paper that is suited to Nat Comm. I would urge, however, the authors, to consider the following as they revise the ms:

1. Classification of news outlets based on a curated list seems like the most straight-forward approach, however, relying on domain name can be misleading. The authors should validate and support their claims that articles published in one of the fake news websites are totally wrong or fabricated. Because it might be the case that "fake news" and "extremely biased news" websites can also publish factual information to attract readers to their sites. Same is true for left-leaning news sources where partially factual or biased information can be shared.

For instance websites like beforeitsnews.com (one of the domain in fakenews category) doesn't create original content, it only aggregates news from different websites (it might be the case that the selection of sources is biased). I believe authors should make a more convincing case against how they label different news sources.

If annotators who build the curated lists rely on their observations on how news spread, authors results are self-fulfilling.

2. Since authors differentiated left and right-leaning news sources, is it possible to identify political views of "extremely biased news"?

3. Granger causality analysis for temporal activities of top 100 users and supporters are interesting. However, I wonder how robust is this analysis? There is 4 order of magnitude difference between the volumes of two timeseries. Also does the observations of one signal being granger-cause of the other consistent across all different dates?

4. In Fig4. date ranges for Clinton and Trump are different. How did you select those date ranges? It would be nice to support claims on one signal is the granger-cause of the other across wider time windows because there might be external events driving activities of top users and supporters behaviors differently.

5. My other observation pertains to the content of those different websites share. I suspect some of the "extremely biased" and "fake news" websites are news aggregators and copycats of other websites. This might be the reason of why the activity of supporters is granger-cause of influencers in news media. Those websites might be picking up the popular or controversial subject

to drive traffic their platforms for monetary or other reasons.

6. Can you also discuss why Granger causality is a good measure in this analysis? Selection of appropriate lag for granger analysis needs better explanation and details for alternative measures in supporting material. One can also employ transfer entropy which is a non-parametric measure amount of directed information transfer between two random processes.

5. It would be an interesting contribution to analyze deleted accounts. Can you infer deletion time of the accounts based on the last tweet observed? I suspect some of these accounts removed from the platform after the elections. If that's the case what are the shared properties of these accounts?

5. Authors might find the references below relevant to their work

- Shao, Chengcheng, et al. "Anatomy of an online misinformation network." arXiv preprint arXiv:1801.06122 (2018).

- Ferrara, Emilio, et al. "The rise of social bots." Communications of the ACM 59.7 (2016): 96-104.

6. Minor points:

- Figure3 can be better visualized by using bar charts or stack plots instead of line charts.

Thank you very much for obtaining three Referee reports on our manuscript “Influence of fake news in Twitter during the 2016 US presidential election” (NCOMMS-18-08034). We are grateful that you have given us the opportunity to address the criticisms of the Referees. The very relevant remarks of the Referees lead us to substantially revise and complement our manuscript.

In what follows, we reply to each separate critique in detail. We have greatly revised the manuscript to comply with all the Reviewer’s critiques. We have marked the changes to the revised manuscript in red for easier identification and explain them in detail below. In the following we have colored the Reviewers comments in blue and the text pasted from the revised manuscript in red to facilitate reading. We hope that you will find these changes adequate to reconsider the manuscript for publication in Nature Communications.

“ Reviewer #1 (Remarks to authors):

I appreciate this work in a number of ways. The authors do a thorough, systematic analysis of an important phenomenon using reproducible methods. The dataset is thoughtfully collected, and so in some sense any systematic analysis of these data are helpful to the public interest because this topic is so important and the relevant data publicly available so sparse.

That said, I have some pretty substantial concerns about the interpretation of the results. As is often the case in big data analysis, the authors have refrained from making explicit claims about their findings, preferring to primarily “describe” patterns and indicate what they might suggest. But this strategy is not really adequate for explaining empirical findings about the social world. Basically, no matter the caveats or hedges the authors provide, their findings will be interpreted by other researchers, and the public and media, as discovering things about what happened in the 2016 election. This audience enthusiasm for over-interpreting the authors’ work puts a burden on them to be explicit about what they found, and, in particular, to consider alternative explanations and to test for their operation. As I will show in detail below, the authors have not done this but in many case could do so quite easily. This burden may seem unfair, but it is only the flip side of the positive attention and enthusiasm that the work generates because of its subject matter. In other words, it is interesting because of its immediate social significance, and the authors can focus on a single case and comparison (2016, U.S. Trump vs. Clinton), because of this significance, so they should in turn bear the burden of treading carefully about this phenomenon.

In keeping with this, the form of my primary concerns will be that the authors suggest or imply an interpretation of the results which, based on the analyses presented in the ms, is not justified because there is another, alternative, they have not considered. I present 5 of these below. Following these I also point out some smaller issues with the paper. ”

We thank the Referee for carefully reading and commenting on our manuscript. In the following we address his concerns and, in particular, consider explicitly the different possible interpretations of our findings.

“ Concern 1

Authors imply: Granger-causal analyses reflect properties of political audiences (e.g. Trump supporters, Concern vs. Clinton supporters, Liberals)

Alternative explanation: Granger-causal analyses reflect Trump campaign’s active and successful use of social media as a central tool of their campaign.

The analysis in section 1.4, in particular Figure 5 and 6, seem to tell the classic story of media polarization and filter bubbles, an interpretation the authors endorse in the discussion. However, this interpretation fails to account for the fact that a substantial portion of the “influencers” in each of the fake, extreme bias, and conservative news areas are Trump and members of his own campaign. This direct influence can be observed in Table 3. Trump himself is in the top 6 in fake news, extremely biased news, and right news. His surrogates (his son, Kellyanne Conway—top advisor and eventual campaign manager; Breitbart news—run by his eventual campaign chief executive, Steve Bannon) are also among the top influencers in both the fake/extremely biased news groups (Table 3).

This raises the question of what the “polarization” and specific preferences observed in Figures 5 and 6 is actually showing. First, the polarization might be showing that Americans with different political loyalties prefer different news sources, something which appears to be increasing over time. OR, it might be showing that Americans of different political loyalties prefer to listen to the candidates they support to get information, something which is not novel and is to be expected. In this second interpretation, the polarization in news sources comes from the choices that Trump and his campaign made. Trump chose different media, so his supporters retweeted different media. If Trump had chosen cat videos, then his supporters would have retweeted cat videos. The polarization is descriptively still real but any attempt to generalize it to new elections, or to compare it with findings from other studies, would need to take this into context.

The same logic must be considered with respect to the kind of news they is shared. The authors imply that it is conservative audiences, or Trump supporters, who prefer “fake news” and “extremely biased” news. This is descriptively true, but it implies that this is feature of their political appetite or epistemology, such as the finding that conspiracy theory seeking seems to be a stable trait as in Bessi et al. (2015). But the explanation might simply be what we already expect, which is that they prefer the news that their candidate prefers, especially as an election approaches and they are engaged in actively try to promote their candidate so that they prevail in the competition (Margolin, Hannak, & Weber, 2018). They appear to like “fake” news, but really they like the news Trump shares, and he happens to like fake news, unsurprisingly, because it is favorable to him. This is campaigning as normal with the exception of Trump’s willingness to forego any concern about what is true or false.

To be clear, my argument is not that this is the explanation, only that it is equally plausible, and substantially different in implication, than what the authors have presented. Moreover, their analysis could consider it. They should, at the very least, remove campaign officials from the “influencer” lists and re-run the analyses, possibly even comparing them to see “how much” of the pattern comes from the campaign. In the absence of this analysis, any claim that attempts to attribute tweeting behavior to audiences preferences or habits, beyond the fact that they like what their candidate likes, is not justified.

Furthermore, the goal of the analysis is not to exonerate any particular individual for sharing fake news, but to use the available evidence to identify the root cause. On one level, it is relevant that Trump supporters shared X quantity of fake news. But the value of this kind of analysis is not in its ability to cast blame between sides but to understand how behavior come about to better understand future cases. In this context, it is crucial to know whether Trump supporters shared fake news because they prefer fake news or because they prefer Trump’s news and he preferred fake news. ”

We thank the Referee for pointing out this important element. We realize indeed that our explanations were insufficient and left room for some misinterpretation. In order to clearly understand the role of the campaign staff in the spread of news, we manually checked each influencer to identify campaign staffers (see Tab. S9) and repeated our causal analysis after having removed the member of each campaign staff (see Fig. S4 and Tab. S11). We observe no significant changes in the causal effects after having removed the campaign staffers.

Concerning our comments on the polarization of the media in two clusters in Figure 6, in this part of our study, we look at the temporal correlation between the activity (tweeting rate) corresponding to each media category and also corresponding to the supporters of each candidate. Here, we do not look at the activity of the influencers, but at the total activity corresponding to each media category. We discover two clear clusters in the media landscape. We related this results to the several previous studies (e.g. [1–3]) that showed a polarization of the news consumers in online social media. What is novel in our work, is that we reveal this polarization only by looking at the temporal correlation of activities and not by tracking which users follow or retweet which media. This shows that news consumers are not only structurally separated but also dynamically separated.

Concerning the interpretation of this polarization, at this point both explanations are indeed equally plausible. Namely, as the Referee says: “the polarization might be showing that Americans with different political loyalties prefer different news sources, something which appears to be increasing over time. OR, it might be showing that Americans of different political loyalties prefer to listen to the candidates they support to get information, something which is not novel and is to be expected.” However, our analysis of the most important spreaders of news and our causal analysis can in fact shed some light on this question and help us have a better understanding of the dynamics between news sources and supporters.

Indeed, by investigating the “super-spreaders” of news (identified using the Collective Influence algorithm) we see more users linked to the campaign staff of Donald Trump (13) than to the campaign staff of Hillary Clinton (3). We also see that Trump staffers have higher ranks in term of influence and cover a broader spectrum of media categories than Clinton staffers. These results are reported in a new table (Tab. S9()) in the Supplementary Information. This reveals that the Trump team played an important role in the diffusion of news in Twitter and in particular they seem to have been much more effective than the Clinton team in this regard. This suggests that the reason why Trump supporters follow news from the center, right and fake news is in part due to the involvement of the Trump team to spread these news and the same cannot be said for the Clinton team.

Our causal analysis (Fig. 7) investigates the role of the top 100 spreaders of each media type and measures how much their activity “causes” (here, the causation must be understood within the limit of the statistical model we use) the activity of the supporters and vice-versa. We find that the only influencers that really drive the supporters activity are the center and left leaning influencers, who are mostly journalists. These results show that the users who are the most important sources of fake news and right news (who include some Trump staffers) are not driving the activity of the Trump supporters. Indeed, the groups that have the largest causal effect on the Trump supporters activity (Tab. 4) are the Clinton supporters, the left leaning news influencers and the center news influencers. Our analysis therefore reveals that although Trump campaign staff play an import role in the diffusion of fake and conservative news, center and left leaning news outlets and journalists are the ones driving Twitter activity, including Trump supporters activity. As suggested by the referee, we rerun the analysis after having removed all spreaders linked to the campaign staff of each candidate. We report the causal graph obtained after removing the staffers of each candidate staff in Fig. S4. We observe no significant changes compared to the causal relations when the staffers are considered (Fig. 7).

To clarify this point we added the following paragraph about the polarization on page 15 of the revised manuscript:

“ These results indicate that the media included in the two clusters respond to two different news dynamics and show that the polarization of news observed at the structural level in previous works [1–3] also correspond to a separation in dynamics of their activity. This separation could be showing that Americans with different political loyalties prefer different news sources but could also be due to the fact that supporters prefer the news that their candidate prefers, especially as an election approaches and they are engaged in actively promoting their candidate [4]. In order to understand the influence of the most important news spreaders on the activity of the supporters, we next investigate the causal relations between the activity of news spreaders and supporters. ”

We also added the following paragraph about the role of campaign staffers on page 13:

“ Another observation is the presence of several member of the campaign staffs of each candidate in the top news spreaders. We report the ranking in each news categories of campaign staffers among the top 100 news spreaders in Tab. 9 of the Supplementary Information. We see more users linked to the campaign staff of Donald Trump (13) than to the campaign staff of Hillary Clinton (3). We also see that Trump staffers have higher ranks in term of influence and cover a broader spectrum of media categories (fake news (3), extreme bias (right) (9), right (9), right leaning (8), center (8) and left leaning (1)) than Clinton staffers (center (1), left leaning (2), left (1) and extreme bias (left) (1)). This reveals that the Trump team played an important direct role in the diffusion of news in Twitter. ”

Finally, to address how the causal analysis can help us understand the influence of campaign staffers, we added the following paragraph on page 19:

“ Although members of the Trump team are prevalent in the top spreaders of fake, extremely biased (right), right and right leaning news, this causal analysis reveals that they are not driving the activity of Trump and Clinton supporters which is more importantly influenced by the top center and left leaning spreaders, consisting mainly of journalists. To verify the importance of users linked to the candidates’ teams, we repeated this causal analysis after having removed all users linked to the campaigns. We report these results in Fig. S4 and Tab. S11 of the Supplementary Information. We observe no significant changes in the causal relations between the different groups as the relations are still dominated by center and left leaning top spreaders. ”

[1] Bakshy, E., Messing, S. & Adamic, L. A. Exposure to ideologically diverse news and opinion on Facebook. *Science* **348**, 1130–1132 (2015). URL <http://www.sciencemag.org/cgi/doi/10.1126/science.aaa1160>.

[2] Schmidt, A. L. *et al.* Anatomy of news consumption on Facebook. *Proceedings of the National Academy of Sciences* **114**, 3035–3039 (2017). URL <http://www.pnas.org/lookup/doi/10.1073/pnas.1617052114>.

[3] Del Vicario, M., Zollo, F., Caldarelli, G., Scala, A. & Quattrociocchi, W. Mapping social dynamics on Facebook: The Brexit debate. *Social Networks* **50**, 6–16 (2017). URL <http://dx.doi.org/10.1016/j.socnet.2017.02.002><http://linkinghub.elsevier.com/retrieve/pii/S0378873316304166>.

[4] Margolin, D. B., Hannak, A. & Weber, I. Political Fact-Checking on Twitter: When Do Corrections Have an Effect? *Political Communication* **35**, 196–219 (2018). URL <https://doi.org/10.1080/10584609.2017.1334018><https://www.tandfonline.com/doi/full/10.1080/10584609.2017.1334018>

“ Concern 2

Authors imply: Trump supporters/conservatives prefer “extreme bias” news more than Clinton supporters/liberals.

Alternative explanation: A single website, Breitbart News, accounts for the bulk of conservative over liberal preference for extreme bias. Moreover, This website is preferred because it is associated with the Trump campaign

According to Table S1, Breitbart News is by far the most dominant in the non-traditional or right wing space, with 1.8M shares – almost 2x or more than any nearest competitor in any other category here. They are also closely aligned with the Trump campaign, as their former leader (Steve Bannon) eventually joined Trump’s campaign as its chairman. Thus, throughout the campaign it would be fair to view Breitbart as promoting Trump, in particular through the highly contested GOP primary. They are almost a campaign surrogate in this period. We would thus expect that Trump supporters would be heavily engaged with this website.

This descriptive fact is important, and the authors provide evidence of just how influential Breitbart was, a potentially important contribution. But instead of the analysis focusing on this one unique

case (highly popular, tied to a specific campaign), they put Breitbart in with all other “extreme bias” websites, allowing Breitbart’s very unique role to accrue to a general category with much farther implications. Put in another way, the distinguishing feature of this category is “extreme bias,” but this is not necessarily the functioning feature of Breitbart. Equally, if not more plausibly, the functioning feature is alignment with the ascendant candidate. The authors should at least remove Breitbart from this category, possibly treat them as their own category, or otherwise link them with the campaign. Other websites that seem to work for the campaign should also be identified and removed in this way. ”

We thank the Referee for this remark. Indeed Breitbart is a rather special news outlets in our dataset as it represents a very large share of the “extreme bias” tweet volume and is linked to the Trump campaign through Steve Bannon.

To understand its role and how it relates to the extreme bias category, we have rerun all the analysis by splitting the extreme bias (right) category in two: 1) Breitbart and 2) all the other outlets in the extreme bias (right). Note, that we have now separated the extreme bias category in extreme bias (right) and extreme bias (left). We have also identified other news outlets that were directly linked to a candidate campaign, i.e. shareblue.com and bluenationreview.com owned by David Brock, a political operative of the Hillary Clinton campaign (https://en.wikipedia.org/wiki/David_Brock). Similarly, we have rerun our analysis by splitting the extreme bias (left) category in two: 1) Shareblue and Bluenationreview and 2) all the other outlets in the extreme bias (left) category. We report the number of tweets and users in these sub-categories in Tab. S1 and the characteristics of their corresponding retweet networks in Tab. S3. We find that the average number of tweets per users and the proportion of tweets sent from unofficial clients are very similar for each sub-categories. The average degree and the heterogeneity of the degree distributions are also similar for each sub-categories. We report the Jaccard indices (i.e. the amount of overlap) of the set of users having tweeted URLs to extreme bias (right) and extreme bias (left) news outlets and their sub-categories in Tab. S2. We find that among all the users that tweeted links toward extreme bias (right) news outlets, 56% tweeted at least one URL directing toward breitbart and 81% tweeted links pointing to extreme bias (right) websites other than breitbart. The percentage of users tweeting links toward breitbart and other extreme bias (right) websites is 37%. This means that users that tweeted only links pointing to breitbart but not to any other extreme bias (right) websites represent 19% of all the users sharing extreme bias (right) news and that among users sharing URLs directing toward breitbart, 66% also share tweets directing to other extreme bias (right) websites. Concerning extreme bias (left) websites, we find that Shareblue and Bluenationreview (SB+BNR) represent only 29% of the users tweeting links directing toward extreme bias (left) websites and other extreme bias (left) websites represent 91%. The intersection between SB+BNR and other extreme bias (left) websites represents 20% of the users sharing extreme bias (left) news. Table S6 shows the Pearson correlation coefficient between the different media categories where we isolated breitbart and shareblue + bluenationreview (SB+BNR) in their own categories and removed them from the extreme bias (right) and extreme bias (left) categories, respectively. We observed that the correlation profile of breitbart and extreme bias (left) minus breitbart are very similar. Far-left minus breitbart has a slightly higher correlation with the right new (0.49 vs 0.35 for breitbart alone) and with the pro-Trump supporters (0.47 vs 0.40 for breitbart alone). On the other hand, SB+BNR has a relatively different correlation profile than extreme bias (left) minus SB+BNR, as it is poorly correlated with all of other categories. Table S8 shows the overlap between the top influencers of each group. We see that the breitbart and extreme bias (right) top 100 influencers share 73 common influencers and the breitbart and extreme bias (right) minus breitbart share 46 common influencers. The extreme bias (left) and (SB+BNR) share 42 influencers. The (SB+BNR) and extreme bias (left) minus (SB+BNR) share 26 top 100 influencers.

We also repeated our causal analysis when considering breitbart and (SB+BNR) as separated from extreme bias (right) and extreme bias (left), respectively. We report the results in Fig. S3 and Tab. S10. We observe that the causal relations of extreme bias (left), (SB+BNR) and extreme bias (left) minus (SB+BNR) are very similar. The causal relations of extreme bias (right), breitbart and extreme bias (right) minus breitbart are also very similar, with the only significant changes being that breitbart top spreaders are more influenced by the activity of Trump supporters than the top spreaders of extreme bias (right) minus breitbart.

Our analysis reveals that, although breitbart represents the largest tweet share of the extreme bias (right) category, the majority (66%) of users sharing links directing toward breitbart also share links toward other websites of the extreme bias (right) category. We also find similar characteristics in term of average activity, retweet network structure, activity correlation and causal relations between breitbart and the rest of the extreme bias (right) category. Considering shareblue and bluenationreview, we find that they form a minority group of the extreme bias (left) category with a strong overlap (69%) and similar characteristics with the rest of the extreme bias (left) category.

The dynamics analysis reveals that the importance and influence of shareblue and bluenationreview is insignificant compared to the other media outlets.

To address this points, we added Tables S1, S2, S3, S6 and S10 as well as Figure S3 to the Supplementary Information and the following paragraph on page 6 of the main revised manuscript:

“ We note that Breitbart News (extreme bias (right)) is the most dominant media outlet in term of number of tweets among the right end of the outlet categories with 1.8 million tweets (see Tab. S13). Breitbart is closely aligned with the Trump campaign as Steve Bannon, who co-founded Breitbart, eventually joined Trump’s campaign as its chief executive. We examine the relation between Breitbart and the rest of the extreme bias (right) outlets in the Supplementary Information (Tables S1, S2, S3, S6 and S10 as well as Figure S3). Our analysis reveals that, although breitbart represents the largest tweet share of the extreme bias (right) category, the majority (66%) of users sharing links directing toward breitbart also share links toward other websites of the extreme bias (right) category. We also find similar characteristics in term of average activity, retweet network structure, activity correlation and causal relations between breitbart and the rest of the extreme bias (right) category. ”

“ Concern 3

Authors imply: Trump supporters/conservatives prefer “extreme bias” news more than Clinton supporters/liberals.

Alternative explanation: Upstart conservative websites, like Breitbart, are categorized as extremely biased rather than “right” because of deviance in form, not content.

The use of OpenSources raises another complication, namely, the blurring of the assessment of “bias,” which has to do with news content, with the assessment of “establishment”, which has to do with news form. The 2016 primaries showed a fissure on the American right. In that fissure, one side was the establishment (e.g. Jeb Bush) and the other anti-establishment. Some candidates (Trump) used foul language (anti-establishment form), others tried to play by the rules. Their positions were no more or less conservative. Trump was more conservative on immigration but less conservative on entitlements and infrastructure, for example. More broadly, norm following is not related to “bias” on a left-right spectrum. In 1992, Bill Clinton was the upstart Democrat in that he was more conservative (i.e., less extreme) than the party had preferred in recent years. Bias and establishment thus do not inherently tie together.

However, inspection of OpenSources coding methodology (from the web page shared in the paper) shows that they do, quite deliberately, conflate these (perhaps for valid reasons for their own research purposes). They use two broad criteria to assign their categories: 1) detection of bias; 2) detection of professionalism / lack of professionalism. Specifically, their steps 4-6 indicate they count thinks like use of the Associated Press style guide and the production quality of the website. These criteria thus conflate adherence to establishment norms—which are likely to be correlated with things like budgets for professional design, fact-checking, editorial oversight etc.—with lack of bias. That is, if two media sites present the same news, but one does it in a less established format, it will be considered “extremely biased.”

This assessment basically bakes in the cake the finding that Trump supporters/conservative appear more biased, because the anti-establishment rupture in their party is coded as one of bias shift rather than source shift. We also see this in the paucity of sites characterized as “right” or “right leaning.” Obviously, if most conservative sites are extremely biased, rather than just “right,” then conservative readers will appear to prefer more biased news. Perhaps if Bernie Sanders had been more successful on the left we would see both sides preferring “extremely biased” news, but in both cases this would be misleading, because what they prefer is non-traditional news, because they find that the traditional news is biased in its suppression of alternative points of view.

I don’t have a solution here other than to, a) find another site or way of categorizing these news sources that does not conflate form and content, as they are very disruptive to this paper’s analysis;

or remove "extreme bias" altogether from the main analysis. It seems inherently biased. I would also move Breitbart over to "right" and see how much things changed. "

We thank the referee for bringing this important point to our attention. We realized that the category "extreme bias" was not well defined and could indeed conflate adherence to establishment norms and political bias.

We have also revised our classification of news outlets to keep only the most important news outlets of each categories and created the sub-categories extreme bias (right) and extreme bias (left) that do not take into account adherence or rejection of the establishment. As we removed websites with an insignificant volume of tweet (we keep only websites that represent at least 1% of the volume of tweet of their corresponding category), we have a smaller number of hostnames and we are able to manually check the political orientation of each website based on [AllSides.com](http://www.allsides.com) and mediabiasfactcheck.com, who do not consider adherence to establishment as a feature of their classification. We then validated our classification by comparing it with the results obtained by Bakshy *et. al* [1] (see Fig. S6) which are based on the average self declared ideological alignment of Facebook users sharing URLs directing to news outlets. We find a $R^2 = 0.9$ for the linear regression between the ideological alignment found by Bakshy *et. al* and our classification where we mapped our categories between -3 and 3.

We agree that it is difficult to completely separate bias due to political view and bias due to rejection of the establishment. Even if we do not use the criteria of adherence or rejection of the establishment in our new classification, websites in the extreme bias (right) and extreme bias (left) categories are probably more likely to not adhere with the establishment as this variable seem to be highly correlated with political bias. We keep Breitbart in the extreme bias (right) category as mediabiasfactcheck.com report it as more biased than Fox News. Breitbart is also classified as more strongly biased than Fox News by Bakshy *et al.* [1] and by Budak *et al.* [5] (we also asked the authors of Budak *et al.* [5] for their bias data in order to compare it with out, but did not receive any answer). Note that now that we added a extreme bias (left) category, we see that highly biased news is also present on the left side of the political spectrum. We find that clear patterns encompassing several media categories emerge from our analysis. Fake news, extreme bias (right) and right as well as center, left-leaning and left from two correlated group in term of activity time series and finally center and left leaning influencers are the only two group that have a significant "causal" effect on the supporter activity. Therefore, our results are robust to any changes of classification within these larger group of media.

We rewrote the explanations of our revised outlet classification on page 3 to make these issues clearer. We report below these modified paragraphs:

" We discarded insignificant outlets accumulating less then one percent of the total number of tweets in their category. We classified the remaining websites in the extremely biased category according to their political orientation by manually checking the bias report of each websites on www.allsides.com and mediabiasfactcheck.com. We find that each website in the extremely biased (right), respectively extremely biased (left), categories has a ranking between right bias and extreme right bias, respectively left and extreme left, with at least a mixed level of factual reporting on mediabiasfactcheck.com and a ranking of right, respectively left, on www.allsides.com (corresponding to the most biased categories of www.allsides.com). We also find that all the websites remaining in the fake news category have a bias between right and extreme right on mediabiasfactcheck.com. The website www.allsides.com rates media bias using a combination of several methods such as blind surveys, community feedback and independent research (see www.allsides.com/media-bias/media-bias-rating-methods for a detailed explanation of the media bias rating methodology used by AllSides), and mediabiasfactcheck.com scores media bias by evaluating wording, sourcing and story choices as well as political endorsement (see mediabiasfactcheck.com/methodology for an explanation of Media Bias Fact Check methodology). "

" We classify news outlets as right, right leaning, center, left leaning and left based on their reported bias on www.allsides.com and mediabiasfactcheck.com. The news outlets in the right leaning, center and left leaning categories are more likely to follow the traditional rules of fact-based journalism. As we move toward more biased categories, websites are more likely to have mixed factual reporting. As for misinformation websites, we discard insignificant outlets by keeping only websites that accumulate more than one percent of the total number of tweets of their respective category. Although we do not know how many news websites are contained in the list of less popular URLs, a threshold

as small as 1% allows us to capture a relatively broad sample of the media in term of popularity. Assuming that the decay in popularity of the websites in each media category is similar, our measure of the proportion of tweets and users in each category should not be significantly changed if we extended our measure to the entire dataset of tweets with URLs.

In order to validate our classification, we compare it to the domain-level ideological alignment scores of news outlets obtained by Bakshy *et al.* [1] which is based on the average self declared ideological alignment of Facebook users sharing URLs directing to news outlets. We find a $R^2 = 0.9$ for the linear regression between the ideological alignment found by Bakshy *et al.* and our classification where we mapped our categories between -3 and 3 (see S6 of the Supplementary Information). While the detail of our classification is subject to some subjectivity, we find that our analysis reveals patterns encompassing several media categories that form group with similar characteristic. Therefore, our results are robust to changes of classification within these larger group of media. ”

We also added the following paragraph to address the issue of the conflation of bias and rejection of the establishment on page 21:

“ A potential issue with the methodology of OpenSources is the blurring of the assessment of “bias,” which has to do with news content, with the assessment of “establishment”, which has to do with news form. Specifically, their steps 4-6 indicate they count thinks like use of the Associated Press style guide and the production quality of the website. These criteria thus conflate adherence to establishment norms – which are likely to be correlated with things like budgets for professional design, fact-checking, editorial oversight etc. – with lack of bias. That is, if two media sites present the same news, but one does it in a less established format, it will be considered “extremely biased.” For this reason, we manually checked the bias of each website in the extreme bias categories on mediabiasfactcheck.com and allsides.com to validate their bias, as these two websites do not list the rejection of the establishment as a criteria for their bias assessment. However, even if we do not use the criteria of adherence or rejection of the establishment in our classification, websites in the extreme bias (right) and extreme bias (left) categories are probably more likely to not adhere with the establishment as this variable seem to be highly correlated with political bias. ”

However, a more detailed study of the interplay between political bias and rejection of the establishment is out of scope for our study. Our main interest is to study the difference in the spreading of news between fake and more traditional news. In this regard, we find important differences between the least biased news and fake news. We also find that the characteristics of the news spreaders and the news diffusion networks change gradually from the least biased to the more biased, indicating that, for example, extreme bias (right) news diffusion shares many characteristics with right news diffusion.

[1] Bakshy, E., Messing, S. & Adamic, L. A. Exposure to ideologically diverse news and opinion on Facebook. *Science* **348**, 1130–1132 (2015). URL <http://www.sciencemag.org/cgi/doi/10.1126/science.aaa1160>.

[5] Budak, C., Goel, S. & Rao, J. M. Fair and Balanced? Quantifying Media Bias through Crowdsourced Content Analysis. *Public Opinion Quarterly* **80**, 250–271 (2016). URL <https://academic.oup.com/poq/article-lookup/doi/10.1093/poq/nfw007>

“ Concern 4

Authors imply: Fake and extremely biased news sites tend to be unverified, suggesting they are non-compliant with Twitter rules, trolls, or other shady characters

Alternative explanation: Twitter’s criteria for verification overlap with OpenSources categorizations of establishment categories (not extreme bias).

This analysis is described near Table 3. It implies that there is something in the Twitter behavior of the fake news / extreme bias promulgators that makes them stick out as unverified. But, as described above, OpenSources uses a site’s establishment bonafides to determine its bias/content. It turns out that this is also closely related to Twitter’s criterion for verification, which is that

they must be “public interest” (from twitter policy – <https://help.twitter.com/en/managing-your-account/about-twitter-verified-account> s). In simple terms, large media, such as sites associated with television stations or other large entities, such as rt.com (financed by the Russian government), are more likely to count as “right” or “right leaning” rather than “fake” or “extreme,” and they are also more likely to be verifiable by Twitter’s policy. Upstarts appear unverifiable by policy and “fake” or “extreme.”

In this case I think the verification analysis should just be removed. It adds a biasing context to the rest of the paper but is not really essential to the larger work. ”

We thanks the Referee for asking this question. Indeed we did not clearly explain the mechanism of verification used by Twitter and did not wanted to imply that unverified accounts are “non-compliant with Twitter rules, trolls, or other shady characters”.

As explained by Twitter: “An account may be verified if it is determined to be an account of public interest. Typically this includes accounts maintained by users in music, acting, fashion, government, politics, religion, journalism, media, sports, business, and other key interest areas. A verified badge does not imply an endorsement by Twitter”. (see <https://help.twitter.com/en/managing-your-account/about-twitter-verified-accounts>)

We also want to stress that being verified by Twitter is not linked to the behavior of the user but requires that the user send a proof of its identify to Twitter. Verified accounts are public figures and they may be anti-establishment (for example: @PrisonPlanet or @RealAlexJones).

Being unverified does not mean that an account does not follow Twitter rules (in this case the account is likely to be deleted). In fact, the vast majority of Twitter accounts are not verified. If an account is unverified it means that either the account did not ask to have its identity verified or that it asked for a verification and Twitter refused.

We therefore find that reporting this measure for the top news spreaders is a valuable information. However, we agree that its interpretation has to be cautious. Fake and extremely biased news is mostly spread by non-public figures. This could be due to the fact that some accounts are trying to hide their real identity but also to the fact that audiences of the extremely biased news are more likely to listen to “non-public” figures due to their distrust of the establishment. We think that both mechanisms are probably at play. The fact that we also observe more deleted accounts on the far end of the spectrum also indicate that accounts not respecting Twitter rules were probably playing a role.

To clarify this point, we added the following remark on page 13:

“ Fake and extremely biased news is mostly spread by unverified accounts which could be due to the fact that some accounts are trying to hide their real identity but also to the fact that audiences of the fake and extremely biased news are more likely to listen to “non-public” figures due to their distrust of the establishment. ”

“ Concern 5

Authors imply: Fake and extreme news sharing communities are more connected due to higher average degree

Alternative explanation: Regular news sharing communities have artificially lower average degree due to expanded endpoints (people with zero in-degree).

First, in Table 2, it’s not clear how the average degree is calculated—based on sum of indegree and outdegree?

More broadly, there is a problem with calculating central tendency in node-level statistics, only. The authors acknowledge that the distribution of degree is likely to be skewed, but the problem is deeper than that. This is a retweet network, a catalogue of individual events, not a “social network” where there are stable relationships or obligations between nodes (like in a follower network). The key difference is in how to interpret the “endpoints” – the nodes with low out-degree and 0 in-degree. In a social network, they are members of the community who are hard to reach, possibly expanding the

diameter of the network and so forth. But in the context of a retweet network, they are essentially “free” from a structural point of view—they can meaningfully describe the network but they don’t imply aspects of its functional structure. The more central nodes don’t owe them anything or know they are there.

More formally, if two networks began as identical, and then in only 1 some subset of nodes were retweeted by a new set of nodes who, themselves, sent only 1 retweet (outdegree = 1) and received no retweets (indegree = 0), the average degree in this network would fall. This is because it just “cheaply” accumulated a bunch of low degree nodes. But it is, in all other ways, functionally the same as the first.

Next, consider that larger networks will, on average, have more opportunities to accumulate these meaningless endpoints, and so large networks may have lower average degree when calculated in this way. Most “social” networks do not face this problem, because it is unusual to accumulate huge numbers of very lower degree nodes – they would be very odd individuals. But in the case of retweets this seems quite likely.

The solution is to compare the distributions of degree and other structural features rather than just provide averages. For example, plots of the degree distribution and Kolmogorov Smirnov tests could indicate whether the differences in average degree are observed throughout the network or really just in the endpoints. Also, the network among the top 100 influencers could be analyzed. This applies to apples as no community distorted by having a different size. It’s actually not clear why the entirety of the data are used for these analysis whereas only the top influencers were used in the network diagrams. ”

We thank the Referee for this interesting remark. We realize that our explanations were not clear. In fact, in a directed graph, the average in-degree and out-degree are always equal. This can be understood by observing that when a directed edge is added to the graph, the number of out-going edges as well as the number of in-going edges both increase by one. In the end the two numbers are equal. The sum of the degrees of each node is equal to the number of edges by the Handshaking lemma (for directed graph). The average degree being the number of edges divided by the number of nodes is therefore the same whether we count in-degree or out-degree. To make this point clearer, we added this information in the caption of Tab. 2.

Let us also remind our definition of in and out-degree. In our networks, the direction of an edge indicates the direction of the information flow. The out-degree of a node, i.e. a user, is equal to the number of different users that have retweeted at least one of her/his tweets. Its in-degree represents the number of different users she/he retweeted. Therefore the “end-points” of information will have a out-degree equal to zero and a in-degree equal to at least one.

Concerning the interpretation of the results, we did not want to imply that “Fake and extreme news sharing communities are more connected due to higher average degree”. In fact, we meant that “users sharing fake and extremely biased news are in average more connected, in terms of information flow network, than users sharing center/left leaning news”. This is shown by the higher average degree of fake and extreme bias (right) networks. However, we do not think that this claim contradicts the second interpretation, namely that center/left leaning information flow network have a larger number of “end-points” with low out-degree. We also agree with the fact that only comparing the average degree is not sufficient to understand the differences between the two types of distributions and this is precisely why we added the measure of degree heterogeneity $\sigma(k_{\text{out}})/\langle k \rangle$ and $\sigma(k_{\text{in}})/\langle k \rangle$. The heterogeneity measures the spread between low-degree and high-degree nodes and reveal that this gap is more important for the center/left leaning networks. Since this measure can depend on the size of the network we used the same number of nodes across all networks when computing the heterogeneity and report the average \pm standard deviation from a bootstrap procedure. Concerning the average degree, the same bootstrap procedure gives the same results than considering the full networks, up to the second decimal. This is due to the fact the the first moment of a heavy tailed distribution converges faster than higher moments (such as the standard deviation). The first moment is generally convergent unless the tail is extremely heavy, with a power-law decay $\sim k^{-\alpha}$, where $\alpha \leq 2$.

To give a complete picture of the degree distributions, we added a figure showing the degree distributions of each network as suggested by the Referee (Fig. 3 on page 8). We hope that the Referee will find these explanations and

added material clarifying. We also performed Kolmogorov-Smirnov tests between each pair of distributions and find that the null hypothesis is rejected for all comparisons except for the out-degree distributions for the right leaning and left news, with two-sided p-values $p < 0.05/n$, where $n = 7$ (Bonferroni correction).

Concerning the interpretation that center/left-leaning networks have and “artificially” lower average degree and that end-points do not imply aspects of their functional structure, we showed that the difference we observe in term of average degree and degree heterogeneity are independent of the sizes of the networks. Therefore we do not think that the explanation of the two networks starting similarly with only one of them accumulating “artificial” low degree nodes stands. We think that the two networks structures we observe indicate two different mechanisms of information spreading. Moreover, we also observe that two different mechanisms of diffusion are also revealed by our study of the diffusion dynamics.

It is also very usual for social networks to have a very large number of low degree nodes and a small number of very high degree nodes. This is revealed by the “power-law” shape of the degree distribution (a straight line with negative slope on a log-log plot) observed in many social networks [6–8]. We also think that the “end-points” of the network play a “functional role” in the diffusion of information. Indeed, even if they are not retweeted by other users, they will broadcast the tweets they retweets to their followers and therefore have a “function” in the diffusion of the information.

Finally, we thank the Referee for suggesting to add plots of the distributions because the main point that we want to make is clearly apparent in Fig. 3. An important feature of the center and left leaning networks is the existence of users with very large out-degree ($k_{\text{out}} > 5 \times 10^4$), i.e. very important broadcasters of information, which are not present in other networks and responsible for the high measured heterogeneity. This supports our claim that the center and left leaning diffusion networks are “controlled” by a small number of influencers. Something that we also show with our causality analysis. We also see that fake, extreme bias (right), extreme bias (left) and right networks are characterized with a higher probability of having lower degree nodes ($k_{\text{out}} \lesssim 100$), indicating that the audience of these news is on average more connected, i.e. retweet more people, than the audience of more traditional news.

In order to make this point clearer we added the following remark in page 9:

“ Figure 3 reveals the existence of users with very large out-degree ($k_{\text{out}} > 5 \times 10^4$), in the center and left leaning networks, i.e. very important broadcasters of information, which are not present in other networks and responsible for the high heterogeneity. We also see that fake, extremely biased (right and left) and right networks are characterized by a higher probability of having lower degree nodes ($k_{\text{out}} \lesssim 100$), indicating that the audience of these news is on average more connected, i.e. retweet more people, than the audience of more traditional news. ”

[6] Watts, D. J. & Strogatz, S. H. Collective dynamics of “small-world” networks. *Nature* **393**, 440–442 (1998)

[7] Barabasi, A.-L. & Albert, R. Emergence of scaling in random networks. *Science* **286**, 11 (1999). URL <http://arxiv.org/abs/cond-mat/9910332>.

[8] Vespignani, A. Modelling dynamical processes in complex socio-technical systems. *Nature Physics* **8**, 32–39 (2011). URL <http://www.nature.com/doi/10.1038/nphys2160>

“ Smaller Issues

Fake news and extreme bias categories hard to distinguish.

It’s not clear why Hate News, which does not appear to be judged by any epistemic criteria, but does relate to biased reasoning, would be placed in the fake news category rather than the extreme bias category. ”

We realize this problem and have now revised our outlet classification (see above).

We have also removed the websites from “Hate News” category from our analysis. We do not find any significant changes in our results after removing Hate News websites.

“ Reviewer #2 (Remarks to the Author):

This is an interesting and thorough paper with a number of important results, particularly about asymmetries between left and right. I think those results could be foregrounded more clearly in various ways, particularly how they relate to the results (not just methods used) in previous research on fake news and partisan asymmetry on Twitter. The limitations of this project also need more foregrounding and discussion. The most problematic section is the analysis of so-called Granger “causality,” which would benefit from a number of additional tests, some strong caveats, and most importantly, an exploration of the full flow (“causation”) network among the various subpopulations examined. ”

We thank the Referee for reading our paper, pointing out its unclear elements and its limitations and suggesting solutions for solving these issues.

In the following, we answer to each remarks in detail.

“ Deferring the temporal/causal analysis for a moment, the main caveat for most of these results that should be foregrounded more in the paper is that this analysis mainly examines a very small subset of major media and other “influencers” on Twitter. While it is certainly the case that this subset accounts for the majority of URL shares, it is not at all therefore the case that the asymmetries the authors find in their behavior in sections 1.1 and 1.2 is generalizable to the sharing behavior of the many millions of more infrequent Twitter users. That’s not a fundamental problem, just a matter of making that caveat clearer: that this doesn’t tell us much about the differing behaviors among the millions of members of the mass public, just the media elite. ”

We thank the Referee for asking us to be more precise on this point. Indeed, our analysis does not cover all the URLs shared on Twitter during our period of observation and therefore our results cannot be generalized to the entire Twitter population without making assumptions on the rest of the population.

Our original dataset contains 171 million tweets sent by 11 million users and concerns only tweets mentioning one of the two main candidates of the elections. We then kept only tweets that contained at least one URL, which represent 55 million tweets sent by 4.0 million users.

In order to have a fair representation of the importance of each media category, we slightly changed our method to select outlets. This new method does not modify any of our results significantly. We rank outlets by the number of tweets linking to them. We then add outlets in each category starting from the top ones until an additional outlet would change the total number of tweets in its category by less than 1%. For this, we manually checked the top 250 websites representing 79% of the total number of tweets with URLs. We find a total of 30.7 million tweets sent by 2.3 million distinct users. Although we do not know how many news websites are contained in the list of less popular URLs, we know that each one of them would represent less than 1% of our selection. We think that using a threshold as small as 1% allows us to capture a relatively broad sample of the media in term of popularity and probably more than just the “elite” media. Although, we agree that by definition of our method, we clearly only focus on the popular media and cannot measure the role of the less popular media. Assuming that the decay in popularity of the websites in each media category is similar, our measure of the proportion of tweets and users in each category should not be significantly changed if we extended our measure to the entire dataset of tweets with URLs.

To make this point clearer we added the following paragraph on page 3:

“ Although we do not know how many news websites are contained in the list of less popular URLs, a threshold as small as 1% allows us to capture a relatively broad sample of the media in term of popularity. Assuming that the decay in popularity of the websites in each media category is similar, our measure of the proportion of tweets and users in each category should not be significantly changed if we extended our measure to the entire dataset of tweets with URLs. ”

We also added the following remark in the discussion on page 20:

“ While our analysis focuses on news concerning the candidate of the presidential election published from the most popular news outlets and therefore its results cannot be generalized to the entire

Twitter population, our investigation provides new insights into the dynamics of news diffusion in Twitter. ”

“ Section 1.4 does go more explicitly examine the mass public, but here the authors provide insufficient detail about how the ”supporters” are calculated. How many are they? How elite are they, or how representative are they of all Twitter users (let alone the public at large)? How disjoint are they from the influencers? This may be discussed in further detail in the supplemental information, but it needs more attention in the main paper because the keystone result – the lead-lag asymmetry between Clinton and Trump supporters relative to the fake and non-fake influencers – depends on us believing that these ”supporters” reveal the ”influence of fake news” as opposed to just the intra-elite dynamics of fake news. ”

We realize that our explanation of the supporter classification were not sufficient. We also realize that without clear information about the number of supporters, one could think that we only focused on the intra-elite dynamics of news diffusion. We now provide more information about the supporters in the main manuscript and hope that this will clarify this question for the Referee.

We classify supporters based on the content of their tweets using a supervised machine learning algorithm trained on a dataset of 1 million tweets obtained from the network of hashtag co-occurrences. The full detail and validation of our method is described in reference [9]. We validated the opinion trend obtained from our method with the national polling average aggregated by the New York Times.

For the classification of the supporters, we used our full dataset of tweets concerning the two candidates, namely 171 million tweets sent by 11 million distinct users during more than five months. After removing automated tweets (sent from unofficial clients) from the dataset we have a total of 157 million tweets. This represents an average of 1.1 million tweets per day (standard deviation of 0.6 million) sent by an average of about 375,000 distinct users per day (standard deviation of 190,000). The supporters therefore represent the general Twitter population commenting on the candidate of the election.

Since the supporters include all users from our initial datasets, the influencers are included in the supporters, however they represent an insignificant fraction (100 compared to 11 million) of the total number of supporters.

To clarify this issue, we added the following paragraph on page 13:

“ We classify supporters based on the content of their tweets using a supervised machine learning algorithm trained on a dataset obtained from the network of hashtag co-occurrences. The full detail of our method and the validation of its opinion trend with the national polling average of the New York Times is described in Ref. [9].

For the classification of the supporters, we used our full dataset of tweets concerning the two candidates, namely 171 million tweets sent by 11 million distinct users during more than five months. After removing automated tweets (see Methods), we have a total of 157 million tweets. This represents an average of 1.1 million tweets per day (standard deviation of 0.6 million) sent by an average of about 375,000 distinct users per day (standard deviation of 190,000). A majority of users, 64%, is in favor of Hillary Clinton while 28% are in favor of Donald Trump (8% are unclassified as they have the same number of tweets in each camp). However, we find that Trump supporters are, in average, 1.5 times more active than Clinton supporters [9]. The supporters therefore represent the general Twitter population commenting on the candidate of the election. ”

“ Before turning to the temporal analysis, a few isolated comments and questions about the first section:

- The authors suggest on page 1 that ”homogeneous beliefs” are equivalent to ”echo chambers,” but surely this is more the exception than the rule, since for the most part everyone shares homogenous beliefs about innumerable things (eg, that the world is round) which does not imply an ”echo chamber”. The propagation of false or unverified information is separate from homogeneity, though the two may be related. ”

We thank the Referee for noticing this mistake. Indeed, we were not sufficiently precise in the introduction and we did not want to mean that ”homogeneous beliefs” are equivalent to ”echo chambers”. What we meant was that

an “echo chamber” is characterized by homogeneous belief about a certain topic, meaning that inside an “echo chamber” people share the same point of view about a given topic, although they may disagree on other topics. Claims that are aligned with this belief, whether false or true, will generally rapidly propagate inside the echo chamber.

We have clarified this point in the Introduction, on page 1, with the following modification:

“ These investigations, as well as theoretical modeling [10, 11], suggest that confirmation bias [12] and social influence results in the emergence, in online social networks, of user communities that share similar beliefs about specific topics, i.e. echo chambers, where unsubstantiated claims or true information, aligned with these beliefs, are as likely to propagate virally [13, 14]. ”

“ - The operationalization of ”extremely biased” was not discussed in enough detail in the main text, especially considering that this is the most controversial of the seven categories.

- More generally, the categorization of left and right among media is a challenging and controversial problem, and I would have liked more robustness checks here to make sure that the categories basically adopted whole-cloth from allsides.com and mediabiasfactcheck.com are robust to alternative specifications. ”

We realize that our classification of extreme bias needed more robustness checks and explanations. We have therefore revised our classification of news outlets to keep only the most important news outlets of each categories and created the sub-categories extreme bias (right) and extreme bias (left) and verified the classification of these websites with mediabiasfactcheck.com in addition to opensources.co. We have also checked the robustness of the classification of mediabiasfactcheck.com and allsides.com with an alternative method.

As we now only keep the most important outlets in each category. Since we now have a smaller number of outlets (which does not change significantly any of our results), we have manually checked each outlets in our categories by looking at their report, on mediabiasfactcheck.com, AllSides.com, wikipedia.org and Pew Research Center (<http://www.journalism.org/interactives/media-polarization>), when available. We also verified the reports of all the outlets in the fake and extreme bias (right/left) category on mediabiasfactcheck.com. We find that outlets that are in the extreme bias category are outlet that are in the most extreme political category on mediabiasfactcheck.com (right to extreme-right or left to extreme-left) and AllSides.com (right or left). They also have a mixed or worse level of factual reporting on mediabiasfactcheck.com.

Finally to test the robustness of our classification with an alternative method, we compared it with the classification obtained by Bakshy *et al.* [1] which is based on the average self declared ideological alignment of Facebook users sharing URLs directing to news outlets. We report this comparison in Fig. S6. We find a $R^2 = 0.9$ for the linear regression between the ideological alignment found by Bakshy *et. al* and our classification where we mapped our categories between -3 and 3.

We are fully aware of the fact that any categorization of media outlets is controversial. However, we find that clear patterns encompassing several media categories emerge from our analysis. In particular, fake news and extreme bias (right) as well as center and left leaning share similar characteristics in term of user average activity and network structure. Fake news, extreme bias (right) and right as well as center, left-leaning and left from two correlated group in term of activity time series and finally center and left leaning influencers are the only two group that have a significant “causal” effect on the supporter activity. Therefore, our results are robust to any changes of classification within these larger group of media.

To make our classification procedure clearer we modified our manuscript with the following explanations on page 3:

“ We discarded insignificant outlets accumulating less then one percent of the total number of tweets in their category. We classified the remaining websites in the extremely biased category according to their political orientation by manually checking the bias report of each websites on www.allsides.com and mediabiasfactcheck.com. We find that each website in the extremely biased (right), respectively extremely biased (left), categories has a ranking between right bias and extreme right bias, respectively left and extreme left, with at least a mixed level of factual reporting

on mediabiasfactcheck.com and a ranking of right, respectively left, on www.allsides.com (corresponding to the most biased categories of www.allsides.com). We also find that all the websites remaining in the fake news category have a bias between right and extreme right on mediabiasfactcheck.com. The website www.allsides.com rates media bias using a combination of several methods such as blind surveys, community feedback and independent research (see www.allsides.com/media-bias/media-bias-rating-methods for a detailed explanation of the media bias rating methodology used by AllSides), and mediabiasfactcheck.com scores media bias by evaluating wording, sourcing and story choices as well as political endorsement (see mediabiasfactcheck.com/methodology for an explanation of Media Bias Fact Check methodology).

We classify news outlets as right, right leaning, center, left leaning and left based on their reported bias on www.allsides.com and mediabiasfactcheck.com. The news outlets in the right leaning, center and left leaning categories are more likely to follow the traditional rules of fact-based journalism. As we move toward more biased categories, websites are more likely to have mixed factual reporting. As for misinformation websites, we discard insignificant outlets by keeping only websites that accumulate more than one percent of the total number of tweets of their respective category. Although we do not know how many news websites are contained in the list of less popular URLs, a threshold as small as 1% allows us to capture a relatively broad sample of the media in term of popularity. Assuming that the decay in popularity of the websites in each media category is similar, our measure of the proportion of tweets and users in each category should not be significantly changed if we extended our measure to the entire dataset of tweets with URLs.

In order to validate our classification, we compare it to the domain-level ideological alignment scores of news outlets obtained by Bakshy *et al.* [1] which is based on the average self declared ideological alignment of Facebook users sharing URLs directing to news outlets. We find a $R^2 = 0.9$ for the linear regression between the ideological alignment found by Bakshy *et al.* and our classification where we mapped our categories between -3 and 3 (see S6 of the Supplementary Information). While the detail of our classification is subject to some subjectivity, we find that our analysis reveals patterns encompassing several media categories that form group with similar characteristic. Therefore, our results are robust to changes of classification within these larger group of media. ”

[1] Bakshy, E., Messing, S. & Adamic, L. A. Exposure to ideologically diverse news and opinion on Facebook. *Science* **348**, 1130–1132 (2015). URL <http://www.sciencemag.org/cgi/doi/10.1126/science.aaa1160>.

“ - The bot detection approach here seems relatively crude compared to the methods at Botometer, etc. Is there a good reason not to use existing state-of-the-art methods? ”

The reason why we used our simple method instead of Botometer is that it is well suited for our dataset as it discriminates at the tweet level, can be applied to historical data and scales very easily to large datasets contrary to more sophisticated methods such as Botometer [15]. Indeed, Botometer is not suited for historical data as it requires several tweets per users (up to 200) and results of a Twitter search of tweets (up to 100) mentioning each users. We compared our method with the results of Botometer (see Methods section of Ref. [9]) and found that our method has a good accuracy but suffer from a relatively high number of false positive compared to Botometer.

We added the following remark on page 21 to clarify this issue:

“ This simple method allows to identify tweets that have not been automated and scales very easily to large datasets contrary to more sophisticated methods [15]. Indeed, Botometer is not well suited for historical data as it requires several tweets per users (up to 200) and results of a Twitter search of tweets (up to 100) mentioning each users, which we cannot do retroactively. We compared our method with the results of Botometer (see Methods section of Ref. [9]) and found that our method has a good accuracy but suffer from a relatively high number of false positive compared to Botometer. ”

[9] Bovet, A., Morone, F. & Makse, H. A. Validation of Twitter opinion trends with national polling aggregates: Hillary Clinton vs Donald Trump. *Scientific Reports* **8**, 8673 (2018). URL <http://www.nature.com/articles/>

s41598-018-26951-y.

[15] Varol, O., Ferrara, E., Davis, C. A., Menczer, F. & Flammini, A. Online human-bot interactions: detection, estimation, and characterization. In *Proc. 11th Int. AAAI Conf. Weblogs Soc. Media*, 280–289 (2017)

“ - Why do the authors restrict themselves to unweighted retweet networks in Section 1.2? The results in Table 2 might differ significantly with weighted networks. ”

The reason why we restricted our analysis to unweighted network is that we wanted to separate the analysis if the activity of users, reported in Tab. 1, and the structure of the networks, reported in Tab. 2. We have repeated our analysis on the weighted networks and report the results in Tab. S4. Our observations of the unweighted networks are also valid for the weighted ones. Namely, the fake news and extreme bias (right) networks have a larger average degree while the center and left leaning networks have a more heterogeneous degree distribution.

We added the following remark on page 9 to address this question:

“ The characteristics of the weighted retweet networks, taking into account multiple interactions between users, are reported in Tab. S4 of the Supplementary Information and reveal the same pattern than the unweighted networks. ”

“ - For the measurement of influence, the authors employ the Collective Influence algorithm. How do the inferred results differ from a more traditional measure like (inflow) eigenvector centrality, or even just degree? ”

We show the comparison of ranking obtained by comparing the ranking obtained by the Collective Influence for the top 100 influencers with their ranking obtained from High degree (HD) and Katz centrality (a version of the eigenvector centrality better suited for directed networks) in Fig. S1. We see that the HD and Katz ranking of the top 100 influencers fall mostly within the top 100 ranks of these two other measure with only a small number of CI influencers having a poor HD or Katz ranking, This shows that the influencer set we find we CI is very similar to the influencer sets we find with HD or Katz centrality. Note that the CI algorithm is especially good at identifying influential nodes that are locally weakly connected but have a large influence on a larger scale (strength of weak ties) [16].

We added Figure S1 in the Supplementary Information and the following remark on page 10:

“ A comparison of the ranking obtained by the CI algorithm with rankings obtained by considering out-degree (high degree) and Katz centrality [17] (see Fig. S1 in the Supplementary Information) shows that high degree and Katz ranking of the top 100 influencers fall mostly within the top 100 ranks of these two other measures with only a small number of CI influencers having a poor HD or Katz ranking. Note that the CI algorithm is especially good at identifying influential nodes that are locally weakly connected but have a large influence on a larger scale [16]. ”

[16] Morone, F. & Makse, H. A. Influence maximization in complex networks through optimal percolation. *Nature* (2015)

[17] Katz, L. A new status index derived from sociometric analysis. *Psychometrika* **18**, 39–43 (1953). URL <http://link.springer.com/10.1007/BF02289026>

“ Turning now to the temporal analysis, as I’m sure the authors are well aware, using ”causal” language invokes a world of criticism that may not be necessary here. There are of course many mechanisms apart from direct causation that might give rise to various groups leading or lagging each other, such as differing attention to television or other non-Twitter media, or just different activity levels or reading frequency on Twitter. While retweets are clearly causal in the most narrow sense, all the other things that lead one to decide to retweet a seen message operate by many different paths, and most importantly, the fact that group A is usually faster off the mark to share some hot piece of news than group B does not mean that A causes or even influences B. This is a hard problem to solve and I don’t expect the authors to solve it, but they need to be much more careful in their use of causal language, particularly since ”granger causality” is at this point viewed with

some skepticism in the causal inference community, to put it mildly. Most importantly, if this isn't (directly) causal, how do the alternate mechanisms affect our interpretation of these results? ”

We thank the Referee for raising this important point. We have revised our causal analysis to use state-of-the-art techniques that overcome most of the limitations of Granger-causality. We now use a non-parametric conditional independence test [18] in a framework developed to infer causal graphs in multiple time series [19].

However, we agree that any causal analysis focusing only on Twitter is necessarily limited since, here, Twitter is only the observed part of a larger system. This means that any measured causal effect could be explained by an unmeasured common driver outside of the measured Twitter activity. Our causal analysis can still help understand the physical process at play and help identify potential driver. In particular, the absence of causal effect between two groups indicate that a physical mechanism between them is not likely. The presence of a causal effect should be interpreted with caution.

In our case, as mentioned by the Referee “the fact that group A is usually faster off the mark to share some hot piece of news than group B does not mean that A causes or even influences B”. Indeed, the fact that center and left leaning news influencers have a causal effect on the Clinton supporters could be explained by the fact that they simply are the first to be “activated” by some news appearing, for example on television, while the supporters take more time to be “activated” by the same news. However, we have other indications that the influencers are directly causing at least part of the supporter activity, namely that the influencers are precisely the most important source of news retweets. Moreover, if the external driver is an other media outside of Twitter and that the center/left leaning influencers, who are almost all journalists, are the first to be activated, it is very likely that the outside media is related to the journalists. In this case, even if the causation is indirect, we still identify the correct driver through the affiliation of the journalists. An other interesting fact is that while we observe a strong causal effect between center/left leaning influencers and the supporters, we do not observe a significant causal effect between other influencers and the supporters. This indicates that, even if the causal driver could be outside of Twitter, the diffusion mechanisms of traditional and fake news are very likely different.

In order to make these import limitations clearer, we added the following paragraph on page 16:

“ An important assumption of causal discovery is causal sufficiency, i.e. the fact that every common cause of any two or more variables in the system is in the system [20]. Here, causal sufficiency is not satisfied since Twitter's activity is only the observed part of a larger social system and the term “causal” must be understood to be meant relative to the system under study. ”

As well as the following paragraph on page 18:

“ The interpretation of the discovered causal effects must be understood within the limitation that we do not measure the diffusion of news outside of Twitter. Indeed, the fact that center and left leaning news influencers have a causal effect on the Clinton supporters could be explained by the fact that they simply are the first to be “activated” by some news appearing, for example on television, while the supporters take more time to be “activated” by the same news. However, we have other indications that the influencers are directly causing at least part of the supporters' activity, namely that the influencers are precisely the most important source of news retweets. Moreover, if the external driver is an other media outside of Twitter and that the center/left leaning influencers, who are almost all journalists, are the first to be activated, it is very likely that the media channel outside of Twitter is related to the journalists. In this case, even if the causation is indirect, we still identify the correct driver through the affiliation of the journalists. More importantly, while we observe a strong causal effect between center/left leaning influencers and the supporters, we do not observe a significant causal effect between other influencers and the supporters. This indicates that, even if the causal driver could be outside of Twitter, the diffusion mechanisms of traditional and fake news are very likely different. ”

[18] Strobl, E. V., Zhang, K. & Visweswaran, S. Approximate Kernel-based Conditional Independence Tests for Fast Non-Parametric Causal Discovery 1–25 (2017). URL <http://arxiv.org/abs/1702.03877>.

[19] Runge, J. *et al.* Identifying causal gateways and mediators in complex spatio-temporal systems. *Nature Communications* **6**, 8502 (2015). URL <http://www.nature.com/articles/ncomms9502>.

[20] Spirtes, P., Glymour, C. & Scheines, R. *Causation, Prediction, and Search* (MIT Press, 2000)

“ Turning to more specifics:

- Regardless of how one deploys causal language here, it is essential to study bidirectional effects among all the groups, as the authors do. But this seems much more suited to a Vector Autoregressive framework (or other methods suited to studying bidirectional temporal data) than a series of paired auto-regressive comparisons, particularly if we want to examine the influence of each node conditional on the others.

- Relatedly, I would very much like to see the lead-lag “causal” network among influencers and between the supporters, and not just between influencers and supporters. It may be that the flows between these groups complicate the supporter asymmetries in important ways, and the fact that they do not form perfectly disjoint sets is a relative fixable problem.

The above two tasks are non-trivial revisions, but worth tackling, since the temporal aspect is the most novel part of the paper. ”

We are grateful to the Referee for pointing out these limitations of our causal analysis. Indeed, simple pairwise Granger-causality tests are susceptible of revealing many spurious links. In our revised manuscript we used a state-of-the art causal discovery algorithm which eliminates spurious links [19,21] by first identifying all significant dependence and then iteratively removing spurious links by conditioning the independence test on all combination of candidate links. As we use a non-parametric conditional dependence test [18], the value of the test statistic is difficult to interpret as a strength of causation. For this reason, once we have discovered all the true parent, we fit a linear model to the data forcing all the non-causal links to zero (similarly to the procedure followed in [19]). This permits us to have a first order measure of causal effects that is easy to interpret and to compute uncertainties on the strength of the links by using a bootstrapping procedure. We also include the links in-between influencer groups and supporter groups in our analysis. The results of the analysis are reported in Fig. 7 and Tab. 4. We also report the causal effects among influencers and between the supporters, and not just between influencers and supporters. We observe that the main results of our analysis are similar to the results we obtained previously. However, they are now much more robust.

We explain our new procedure on page 16:

“ In order to investigate the causal relations between news media sources and Twitter dynamics, we use a multivariate causal network reconstruction of the links between the activity of news influencers and supporters of the presidential candidates based on a causal discovery algorithm [19–21]. The causal network reconstruction tests the independence of each pair of time-series, for several time lags, conditioned on potential causal parents with a non-parametric kernel-based conditional independence test [18,22] (see Methods). To estimate the causal effects between the different processes, we use the causal algorithm as a variable selection and perform a linear regression using only the true causal link discovered. We consider linear causal effects for their reliable estimation and interpretability. This permits us to compare the causal effect as first order approximations, estimate the uncertainties of the model and reconstruct a causal directed weighted networks [19]. In this framework, the causal effect between a time series X^i and X^j at a time delay τ , $I_{i \rightarrow j}^{CE}(\tau)$, is equal to the expected value of X_t^j (in unit of standard deviation) if $X_{t-\tau}^i$ is perturbed by one standard deviation [19]. ”

Moreover, we give all the details of the causal analysis in a new section of the Methods called “Causal analysis” on page 22.

We also added the following comment about the causal links in-between sub-groups on page 18:

“ Regarding the causal relations in-between news spreaders, center news spreaders are the most central driver as they are among the top three driver of all news spreaders. Strong mutual causal effects are revealed between center and left leaning spreaders. Right leaning top spreaders are driving

the activity of the right, extremely biased (right) and fake news spreaders. The two supporter groups have also strong mutual causal effects. ”

[18] Strobl, E. V., Zhang, K. & Visweswaran, S. Approximate Kernel-based Conditional Independence Tests for Fast Non-Parametric Causal Discovery 1–25 (2017). URL <http://arxiv.org/abs/1702.03877>.

[19] Runge, J. *et al.* Identifying causal gateways and mediators in complex spatio-temporal systems. *Nature Communications* **6**, 8502 (2015). URL <http://www.nature.com/articles/ncomms9502>.

[20] Spirtes, P., Glymour, C. & Scheines, R. *Causation, Prediction, and Search* (MIT Press, 2000)

[21] Runge, J., Heitzig, J., Petoukhov, V. & Kurths, J. Escaping the Curse of Dimensionality in Estimating Multivariate Transfer Entropy. *Physical Review Letters* **108**, 258701 (2012). URL <https://link.aps.org/doi/10.1103/PhysRevLett.108.258701>

[22] Zhang, K., Peters, J., Janzing, D. & Schoelkopf, B. Kernel-based Conditional Independence Test and Application in Causal Discovery. In *UAI 2011, Proceedings of the Twenty-Seventh Conference on Uncertainty in Artificial Intelligence*, 804–813 (Barcelona, 2011). URL <http://arxiv.org/abs/1202.3775>.

“ More minor points about the temporal analysis follow:

- Perhaps the most interesting result of the first section is the asymmetry revealed in Table 2: The higher degree within the fake, biased and right media, vs the higher heterogeneity among the center and left. This intra-group analysis seems in tension with the results on page 11 and Figure 5, where the mass public on the right (Trump supporters) shows the greater heterogeneity in their temporal sharing patterns than the more homogeneous left. Is this due to the difference between the elite and the masses, or due to differences between analyzing an a-temporal follower networks vs the temporal retweet-correlation network? ”

This is an interesting question. We do not think that the heterogeneity in the diffusion network connectivity is linked to the heterogeneity in the sharing patterns of the Trump supporters.

Indeed, the heterogeneity in the diffusion network connectivity measures the difference in term of number of connections (retweets) within users sharing a certain type of news. Note that the networks are not only formed by the elite but by all users sharing URLs directing to the news outlets (the sizes of all networks are reported in Tab. 2). On the other hand, the fact that Trump supporters’ activity is correlated with the activity of all media categories means that they tend to be activated at the same time. This indicate that Trump supporters are more likely to react to any type of news while Clinton supporters mostly react to center and left news and tend to ignore right news.

We added the following paragraph on page 15 to make this point clearer.

“ We observe the following patterns between the media groups and the supporters dynamics: the activity of Clinton supporters has a higher correlation with the second cluster than with the first one while the activity of Trump supporters is equally correlated with the two clusters. This indicate that Trump supporters are likely to react to any type of news while Clinton supporters mostly react to center and news on the left and tend to ignore news coming from the right side. ”

“ - As discussed above, the authors mention on page 14 that ”we assume that [the influencers] are the ones triggering the activity of the rest of the population.” This is a very strong assumption and needs to be foregrounded much more, and its robustness explored. ”

We agree that this assumption is very strong and regret to not have given a clearer explanation of our motivation. In fact, what we meant is that, within the limitation of considering Twitter as a closed system, the influencers being the most important sources of retweets, are the more likely to be the ones triggering the activity. We have now also shown that the ranking obtained by CI is similar to the rankings obtained by other more common centrality measures (see Fig. S1) We therefore use them to test this hypothesis. As a matter of fact, we find that this is likely to be the case only for the center and left leaning influencers.

We have reformulated and better explained this point in the manuscript on page 16 and report this modification below:

“ We consider only the activity of the top influencers since, by the definition of CI, they are the most important sources of information. Therefore, within limitation of considering Twitter as a closed system, they are the more likely set of users to trigger the activity of the rest of the population. We test this hypothesis with our causal analysis. ”

“ - A p threshold of 10^{-7} seems somewhat arbitrary, and I would like to see actual evidence that these patterns are robust to different thresholding, since such thresholding can often make a large difference in converting continuous relation data to networks. At the very least, a reasonable False Discovery Rate correction would make the threshold seem less arbitrary. ”

We thank the Referee for suggesting to use False Discovery Rate (FDR) correction instead of an arbitrary p-value threshold. In the revised manuscript, we used a FDR correction of the p-values [23] and used a threshold level of 0.05 to select significant causal links.

We specify this point on page 23:

“ We select the significant final causal links by applying a Benjamini-Hochberg False Discovery Rate correction [23] to the p -values of the conditional independence tests with a threshold level of 0.05. FDR corrections allow to control the expected proportion of false positive. The final causal links, i.e. parents of each time series, are reported in Tab. S15 of the Supplementary Information. ”

[23] Benjamini, Y. & Hochberg, Y. Controlling the False Discovery Rate : A Practical and Powerful Approach to Multiple Testing. *Journal of the Royal Statistical Society Series B* **57**, 289–300 (1995)

“ Reviewer #3 (Remarks to the Author):

This paper investigates the timely problem of news sources’ influence on Twitter during the last US presidential election. The authors track and collect millions of tweets that contain URLs to news articles with different political alignments and to misinformation websites. Its an interesting and timely paper that is suited to Nat Comm. I would urge, however, the authors, to consider the following as they revise the ms:

1. Classification of news outlets based on a curated list seems like the most straight-forward approach, however, relying on domain name can be misleading. The authors should validate and support their claims that articles published in one of the fake news websites are totally wrong or fabricated. Because it might be the case that ”fake news” and ”extremely biased news” websites can also publish factual information to attract readers to their sites. Same is true for left-leaning news sources where partially factual or biased information can be shared. ”

We thank the Referee for having taken the time to read and comment on our manuscript. We thank the Referee for this important remark about domain-level classification. We agree that classifying URLs based on their domain name is an imperfect method. For example, some news articles containing true information but emanating from websites classified as “fake news” will be classified as fake while some false information published by “traditional” media will not be considered as fake.

Our decision to classify news at the domain level must therefore be clearly motivated. In the following, we try to give a better explanation of our motivations. Firstly, we are not interested in the punctual effect of a real or fake news but in understanding the integrated effect of fake and traditional news over the entire 5 months of observation. For this reason we want to consider all the URLs linking to a news outlets. While, ideally classifying each URL as fake or true would be interesting, the fraction of fact-checked news compared to the total amount of news is in fact extremely small. For example, Vosoughi *et al.* [24] found only 126,000 fact-checked rumors in the entire Twitter dataset since its creation. However, we think that classifying domains instead of URL (or article) is relevant. Indeed, when a Twitter user follows a URL to a news article containing factual information on a website publishing mostly fake news, she/he will be exposed to the other articles containing fake news on the websites. Therefore, this particular fact-based news ultimately increases the potential influence of fake news.

To make this point clearer and to stress that our results do not directly measure the influence of fake or traditional news but rather the influence of news outlets which publish mostly fake or mostly fact-based news, we added the following paragraph on page 4:

“ We investigate the influence and importance of news at the domain level and not at the article level. Since a website classified as fake may contain factual articles and vice versa, domain-level classification implies a level of imprecision. However, it allows us to reveal the integrated effect of news outlets over more than five months and to measure the relative importance of each type of news by classifying all URLs directing to important news outlets. Moreover, classifying domains instead of URL (or article) allows to consider the extended effect of each type of news. Indeed, when a Twitter user follows a URL to a news article containing factual information on a website publishing mostly fake news, she/he will be exposed to the other articles containing fake news on the websites. Therefore, this particular fact-based news ultimately increases the potential influence of fake news. ”

[24] Vosoughi, S., Roy, D. & Aral, S. The spread of true and false news online. *Science* **359**, 1146–1151 (2018). URL <http://www.sciencemag.org/lookup/doi/10.1126/science.aap9559>

“ For instance websites like beforeitsnews.com (one of the domain in fakenews category) doesn’t create original content, it only aggregates news from different websites (it might be the case that the selection of sources is biased). I believe authors should make a more convincing case against how they label different news sources. ”

We are grateful to the Referee for pointing out the issue of news aggregators which indeed may play a different role than news producers. We verified all the news outlets in our revised manuscript to check if they were classified as news aggregators by wikipedia.org, politifact.com and by manually checking the websites. We find the following

news aggregators: zerothedge.com (fake news), wnd.com (extreme bias (right)), realclearpolitics.com (right leaning) and truepundit.com (extreme bias (right)). In our revised manuscript we consider less hostnames than in our initial manuscript (we discard websites that accumulate less than one percent of the total number of tweets of their respective category) and find similar results. For this reason, beforeitsnews.com is no longer part of our list of hostnames. To understand the role of news aggregators, we repeated dynamics analysis without news aggregators (see Tab. S7, Tab. S8 and . We answer more in detail to this point below (question 5).

We also revised our classification of outlets to make it clearer. We discarded insignificant outlets and added the categories extreme bias (right) and extreme bias (left), and manually checked the bias of each websites. We then validated our classification by comparing it with the results obtained by Bakshy *et. al* [1] (see Fig. S6) which are based on the average self declared ideological alignment of Facebook users sharing URLs directing to news outlets. We find a $R^2 = 0.9$ for the linear regression between the ideological alignment found by Bakshy *et. al* and our classification where we mapped our categories between -3 and 3.

We rewrote the explanations of our revised outlet classification on page 3 to make these issues clearer. We report below these modified paragraphs:

“ We discarded insignificant outlets accumulating less then one percent of the total number of tweets in their category. We classified the remaining websites in the extremely biased category according to their political orientation by manually checking the bias report of each websites on www.allsides.com and mediabiasfactcheck.com. We find that each website in the extremely biased (right), respectively extremely biased (left), categories has a ranking between right bias and extreme right bias, respectively left and extreme left, with at least a mixed level of factual reporting on mediabiasfactcheck.com and a ranking of right, respectively left, on www.allsides.com (corresponding to the most biased categories of www.allsides.com). We also find that all the websites remaining in the fake news category have a bias between right and extreme right on mediabiasfactcheck.com. The website www.allsides.com rates media bias using a combination of several methods such as blind surveys, community feedback and independent research (see www.allsides.com/media-bias/media-bias-rating-methods for a detailed explanation of the media bias rating methodology used by AllSides), and mediabiasfactcheck.com scores media bias by evaluating wording, sourcing and story choices as well as political endorsement (see mediabiasfactcheck.com/methodology for an explanation of Media Bias Fact Check methodology). ”

“ We classify news outlets as right, right leaning, center, left leaning and left categories based on their reported bias on www.allsides.com and mediabiasfactcheck.com. The news outlets in the right leaning, center and left leaning categories are more likely to follow the traditional rules of fact-based journalism. As we move toward more biased categories, websites are more likely to have mixed factual reporting. As for misinformation websites, we discard insignificant outlets by keeping only websites that accumulate more than one percent of the total number of tweets of their respective category. Although we do not know how many news websites are contained in the list of less popular URLs, a threshold as small as 1% allows us to capture a relatively broad sample of the media in term of popularity. Assuming that the decay in popularity of the websites in each media category is similar, our measure of the proportion of tweets and users in each category should not be significantly changed if we extended our measure to the entire dataset of tweets with URLs.

In order to validate our classification, we compare it to the domain-level ideological alignment scores of news outlets obtained by Bakshy *et al.* [1] which is based on the average self declared ideological alignment of Facebook users sharing URLs directing to news outlets. We find a $R^2 = 0.9$ for the linear regression between the ideological alignment found by Bakshy *et. al* and our classification where we mapped our categories between -3 and 3 (see S6 of the Supplementary Information). While the detail of our classification is subject to some subjectivity, we find that our analysis reveals patterns encompassing several media categories that form group with similar characteristic. Therefore, our results are robust to changes of classification within these larger group of media. ”

“ If annotators who build the curated lists rely on their observations on how news spread, authors results are self-fulfilling. ”

We thank the Referee for this pertinent remark. In our revised manuscript, we use OpenSources.com for the list of fake news websites who do not rely on how news spread for constructing their list. Moreover, we manually checked each fake news websites on fact checking websites (snopes, factcheck.org,...) which verify the factual accuracy and do not rely on how news spread. For all the other categories, we used AllSides.com and mediabiasfactcheck.com, which estimate the bias of news outlets without taking into account how news spread.

“ 2. Since authors differentiated left and right-leaning news sources, is it possible to identify political views of ”extremely biased news”? ”

We agree that the “extreme bias” category was not well defined since it was potentially mixing outlets with left and right political orientations As we explained above, in our revised manuscript, we created the categories extreme bias (right) extreme bias (left) which separates the most extreme sources according to their political orientation. We find that the extreme bias (right) category contains 17 outlets and the extreme bias (left) contains 7 outlets.

“ 3. Granger causality analysis for temporal activities of top 100 users and supporters are interesting. However, I wonder how robust is this analysis? There is 4 order of magnitude difference between the volumes of two timeseries. Also does the observations of one signal being granger-cause of the other consistent across all different dates? ”

In the revised manuscript, we completely revised the causal analysis to use a robust framework of causal network discovery. All the time series are standardized before performing the causal dependence tests, therefore removing any influence of the difference in absolute values of time series.

Concerning the robustness of the causal analysis at different dates, the results of the causal analysis we show are in fact performed using the entire time period (more than 5 months) and therefore show a causal effects that are observed “in average” over the entire time period. We apologize for not making these points clearer and have added a remark on page 16 of the revised manuscript clarifying these issues:

“ Before performing the causal analysis, we also standardized each time series in order to remove any influence of the difference in absolute values of time series. The causal analysis is performed using the entire time period (more than 5 months) and therefore reveals causal effects that are observed “in average” over the entire time period. ”

“ 4. In Fig4. date ranges for Clinton and Trump are different. How did you select those date ranges? It would be nice to support claims on one signal is the granger-cause of the other across wider time windows because there might be external events driving activities of top users and supporters behaviors differently. ”

Again, we apologize for this unclear point. The causal relations between the different time series are found using the full time span of our dataset. Figure 5 shows only an illustrative example of the causal relations in order to facilitate the understanding of what we mean by “causal effect”. But all the results of the causal analysis are performed on the full time span, taking into account the largest possible amount of information we have.

In addition to the remark about the used time span above, we have also revised the caption of 5 in order to clarify this point:

“ In Fig. 7 and Tab. 4 we report the causal effects computed over the entire time span, from June 2016 until election day. ”

“ 5. My other observation pertains to the content of those different websites share. I suspect some of the ”extremely biased” and ”fake news” websites are news aggregators and copycats of other websites. This might be the reason of why the activity of supporters is granger-cause of influencers in news media. Those websites might be picking up the popular or controversial subject to drive traffic their platforms for monetary or other reasons. ”

We thank the Referee for this very pertinent point. Indeed, after manually checking each websites we found that several in the fake news, extreme bias (right) and lean-right category were, at least partly, news aggregators. To understand if this fact plays a role in the news dynamics we observe, we repeated our analysis of the dynamics after

having removed the news aggregators from our dataset. We show the results of the activity correlation analysis in Tab. S7. We observe no significant changes in the correlation coefficient between the analysis with and without news aggregators. The maximum difference in correlation (0.02) is between the right leaning and extreme bias (right). We also observe in Tab. S8 that the set of top 100 influencers does not change greatly when removing the news aggregators. The sets of top 100 fake news and fakes news without aggregators influencers have 96 influencers in common. Their are also 96 influencers in common in the top 100 sets of extreme bias (right) and extreme bias (right) without aggregators. The right leaning and right leaning without aggregators top 100 influencers see the largest change, but still have 82 influencers in common.

We report the results of the causal analysis after having removed the news aggregators in Fig. S5 and Tab. S12. We see that our conclusions stay valid even without the news aggregators, namely the domination of center and left leaning influencers in term of causal effects. We observe a small decrease in the intensity of the causal effect of center influencers toward Clinton supporters (0.065 to 0.046), but the effect is still the second most important after the left leaning influencers. We also observe a small increase of the causal effect of Clinton supporters on the fake news top spreaders. Without the news aggregators, the top fake news, extreme bias (right) and right leaning spreaders have a smaller causal effect on the other groups.

To clarify this issue, we added Tables S7, S8 & S12 and Fig. S5 in the Supplementary Information as well as the following paragraph on page 19 of the revised manuscript:

“ A possible distinction between the diffusion mechanisms of different news outlets could be due to the fact that some websites aggregates news from other websites instead of producing news. We find four websites that, at least partly, aggregates news: zerothedge.com (fake news), wnd.com (extreme bias (right)), realearpolitics.com (right leaning) and truepundit.com (extreme bias (right)). To understand if the presence of news aggregators in categories other than the center and left leaning could explain the difference in dynamics that we observe, we repeated our analysis of the dynamics after having removed the news aggregators from our dataset. We report the results in the Supplementary Information (Tabs. S7, S8 & S12 and Fig. S5). We observe no significant changes in the activity correlations and and that without the news aggregators, the top fake news, extreme bias (right) and right leaning spreaders have a smaller causal effect on the other groups, while the left leaning and center influencers stay the dominant ones. This shows that news aggregators are not responsible for the differences on dynamics that we observe. ”

“ 6. Can you also discuss why Granger causality is a good measure in this analysis? Selection of appropriate lag for granger analysis needs better explanation and details for alternative measures in supporting material. One can also employ transfer entropy which is a non-parametric measure amount of directed information transfer between two random processes. ”

We agree with the referee that Granger causality could be too rudimentary for our analysis and have therefore completely revised our framework for causal analysis. We now use a kernel based non-parametric conditional independence test [18, 22] between time series to identify causal links within a range of time delays. Using transfer entropy in our case is unfeasible due to the size of our problem (13 152 time points \times 10 time series \times 18 time lags). We also use a causal discovery framework that infer links by taking into account potential spurious dependences due to common drivers [19, 21, 25], especially when strong autocorrelations are present, instead of the simple pairwise tests we used in the first version of our manuscript. However, the results of this state-of-the-art causal discovery framework are not significantly different than the results we obtained with the Granger-causality method which indicates that our observations are robust.

We explain our new procedure on page 16:

“ In order to investigate the causal relations between news media sources and Twitter dynamics, we use a multivariate causal network reconstruction of the links between the activity of news influencers and supporters of the presidential candidates based on a causal discovery algorithm [19–21]. The causal network reconstruction tests the independence of each pair of time-series, for several time lags, conditioned on potential causal parents with a non-parametric kernel-based conditional independence test [18, 22] (see Methods). To estimate the causal effects between the different processes, we use the causal algorithm as a variable selection and perform a linear regression using only the true causal link discovered. We consider linear causal effects for their reliable estimation and interpretability.

This permits us to compare the causal effect as first order approximations, estimate the uncertainties of the model and reconstruct a causal directed weighted networks [19]. In this framework, the causal effect between a time series X^i and X^j at a time delay τ , $I_{i \rightarrow j}^{\text{CE}}(\tau)$, is equal to the expected value of X_t^j (in unit of standard deviation) if $X_{t-\tau}^i$ is perturbed by one standard deviation [19]. ”

Moreover, we give all the details our new framework in the Methods section on page 22 and report below this section. we also show the correlations as a function of the time lag in between time series in Figures S7, S8, S9 and S10.

“ In order to infer the causal relations between the activity of the influencers and the supporters, we use a multivariate causal discovery algorithm based on the PC algorithm [20] and further adapted for multivariate time series by Runge *et al.* [19, 21, 25, 26]. Considering an ensemble of stochastic processes X described by their standardized time series, the algorithm proceeds as follow. First, for every time series $Y \in X$ the sets of preliminary parents is constructed by testing the their independence at a range of time lags: $\mathcal{P}_{Y_t} = \{X_{t-\tau} | 0 < \tau \leq \tau_{\text{max}}, Y_t \not\perp\!\!\!\perp X_{t-\tau}\}$. As this set also contains indirect links, they are then removed by testing if the dependence between Y_t and each $X_\tau \in \mathcal{P}_{Y_t}$ vanishes when it is conditioned on an incrementally increased set of conditions $\mathcal{P}_{Y_t}^{n,i} \subseteq \mathcal{P}_{Y_t}$, where n is the cardinality of $\mathcal{P}_{Y_t}^{n,i}$ and i is the index iterating over the number of combinations of picking n conditions from \mathcal{P}_{Y_t} . The combinations of parents having the strongest dependence in the previous step are selected first [19, 25].

The main free parameters are the maximum time lag τ_{max} and the significance level of the independence test used during the first step to build the set of preliminary parents which we set to $\alpha_{\text{PC}} = 0.1$. We set the value of the maximum time lag to $\tau_{\text{max}} = 18$ time steps (i.e. 270 min) as it is the lag after which the lagged cross-correlations between each time series falls below 0.1 in absolute value (see Figures S7, S8, S9 and S10 in the Supplementary Information). We set the maximum number of tested combinations of the conditioning set to 3 and we do not limit the size of the conditioning set.

We test the conditional independence of time series with the non-parametric RCoT test [18, 27]. This test uses random Fourier features to approximate the kernel-based conditional independence test KCIT [22] and is at least as accurate as KCIT while having a run time that scales linearly with sample size [18]. This point is crucial for our case given the size of our dataset (13 152 time points \times 10 time series \times 18 time lags).

We select the significant final causal links by applying a Benjamini-Hochberg False Discovery Rate correction [23] to the p -values of the conditional independence tests with a threshold level of 0.05. FDR corrections allow to control the expected proportion of false positive. The final causal links, i.e. parents of each time series, are reported in Tab. S15 of the Supplementary Information.

Following the procedure of Refs. [19, 28], we then regress a linear model:

$$X_t = \sum_{\tau=1}^{\tau_{\text{max}}} \Phi(\tau) X_{t-\tau} + \varepsilon_t, \quad (1)$$

where all time series are standardized and only coefficient corresponding to causal links are estimated while all the other ones are kept equal to zero, i.e. $\Phi_{ij}(\tau) \neq 0$ only for $X_{t-\tau}^i \rightarrow X_t^j$. The causal effect between a time series X^i and X^j at a time delay τ can be computed from the regressed coefficients as:

$$I_{i \rightarrow j}^{\text{CE}}(\tau) = \Psi_{ij}(\tau), \quad (2)$$

where $\Psi(\tau)$ is computed from the relation $\Psi(\tau) = \sum_{s=1}^{\tau} \Phi(s) \Psi(\tau - s)$, with $\Phi(0) = I$. Here, $\Psi_{ij}(\tau)$ gives the sum over the products of path coefficients along all causal paths up to a time lag τ . The causal effect $I_{i \rightarrow j}^{\text{CE}}(\tau)$ represents the expected value of X_t^j (in unit of standard deviation) if $X_{t-\tau}^i$ is perturbed by one standard deviation [19].

To reconstruct the causal network, we are interested in the aggregated effects and therefore use the lag with maximum effect:

$$I_{i \rightarrow j}^{\text{CE,max}} = \max_{0 < \tau \leq \tau_{\text{max}}} |I_{i \rightarrow j}^{\text{CE}}(\tau)|. \quad (3)$$

We estimate the standard errors of each causal effects with a residual-based bootstrap procedure (similarly to Ref. [19]). We employ 200 bootstrap surrogates time series generated by running model (1) with a joint random sample ε_t^* (with replacement) of the original multivariate residual time series ε_t . ”

[18] Strobl, E. V., Zhang, K. & Visweswaran, S. Approximate Kernel-based Conditional Independence Tests for Fast Non-Parametric Causal Discovery 1–25 (2017). URL <http://arxiv.org/abs/1702.03877>.

[19] Runge, J. *et al.* Identifying causal gateways and mediators in complex spatio-temporal systems. *Nature Communications* **6**, 8502 (2015). URL <http://www.nature.com/articles/ncomms9502>.

[21] Runge, J., Heitzig, J., Petoukhov, V. & Kurths, J. Escaping the Curse of Dimensionality in Estimating Multivariate Transfer Entropy. *Physical Review Letters* **108**, 258701 (2012). URL <https://link.aps.org/doi/10.1103/PhysRevLett.108.258701>

[22] Zhang, K., Peters, J., Janzing, D. & Schoelkopf, B. Kernel-based Conditional Independence Test and Application in Causal Discovery. In *UAI 2011, Proceedings of the Twenty-Seventh Conference on Uncertainty in Artificial Intelligence*, 804–813 (Barcelona, 2011). URL <http://arxiv.org/abs/1202.3775>.

[25] Runge, J., Sejdinovic, D. & Flaxman, S. Detecting causal associations in large nonlinear time series datasets (2017). URL <http://arxiv.org/abs/1702.07007>.

[26] Runge, J. TIGRAMITE software. URL <https://jakobrunge.github.io/tigramite>

[27] Strobl, E. V. RCIT and RCoT software. URL <https://github.com/ericstrobl/RCIT>

[28] Eichler, M. & Didelez, V. On Granger causality and the effect of interventions in time series. *Lifetime Data Analysis* **16**, 3–32 (2010)

“ 5. It would be an interesting contribution to analyze deleted accounts. Can you infer deletion time of the accounts based on the last tweet observed? I suspect some of these accounts removed from the platform after the elections. If that’s the case what are the shared properties of these accounts? ”

We thank the Referee for this interesting suggestion. We have investigated the 28 deleted accounts in the top 100 influencers and we find that, based on the timestamp of their last tweet in our dataset, 24 out of the 28 accounts had tweeted after election day (November 8th, 2016) indicating that they were deleted after the election. We find that they tend to be very active, with a median number of tweets of 2224 (minimum: 156, 1st quartile: 1400, 3rd quartile 6711 and max: 15930). In comparison, the median number of tweet for our entire dataset is 2. We also find that 21 accounts used at least once an unofficial Twitter client (the most used unofficial client is dlvr.it) to send tweets.

We report these findings on page 12:

“ We find that, based on the timestamp of their last tweet in our dataset, 24 out of the 28 accounts had tweeted after election day (November 8th, 2016) indicating that they were deleted after the election. Deleted accounts were extremely active, with a median number of tweets of 2224 (minimum: 156, 1st quartile: 1400, 3rd quartile 6711 and max: 15930). In comparison, the median number of tweets per users for our entire dataset is 2. We also find that 21 accounts used at least once an unofficial Twitter client (the most used unofficial client by deleted accounts is dlvr.it) to send tweets. ”

“ 5. Authors might find the references below relevant to their work

- Shao, Chengcheng, et al. ”Anatomy of an online misinformation network.” arXiv preprint arXiv:1801.06122 (2018).

- Ferrara, Emilio, et al. ”The rise of social bots.” *Communications of the ACM* **59.7** (2016): 96-104.

”

Thank you very much for these interesting and relevant references, we have added them to the introduction on page 2:

“ Recent works also revealed the role of bots, i.e. automated accounts, in the spread of misinformation [24, 29–31]. In particular, Shao *et al.* found that, during the 2016 US presidential election on Twitter, bots were responsible for the early promotion of misinformation, that they targeted influential users through replies and mentions [32] and that the sharing of fact-checking articles nearly disappears in the core of the network, while social bots proliferate [33]. ”

[31] Ferrara, E., Varol, O., Davis, C., Menczer, F. & Flammini, A. The rise of social bots. *Communications of the ACM* **59**, 96–104 (2016). URL <http://arxiv.org/abs/1407.5225><http://dl.acm.org/citation.cfm?doid=2963119.2818717>.

[33] Shao, C. *et al.* Anatomy of an online misinformation network. *arXiv* (2018). URL <http://arxiv.org/abs/1801.06122>.

“ 6. Minor points:

- Figure3 can be better visualized by using bar charts or stack plots instead of line charts. ”

We have changed the Fig. 4 to a bar plot, which provides, indeed, a better visualization.

We thank again the Reviewers for their constructive comments and hope that they will find these substantial revisions adequate to reconsider the manuscript for publication in Nature Communications.

Yours sincerely,

Alexandre Bovet & Hernán A. Makse

References

- [1] Bakshy, E., Messing, S. & Adamic, L. A. Exposure to ideologically diverse news and opinion on Facebook. *Science* **348**, 1130–1132 (2015). URL <http://www.sciencemag.org/cgi/doi/10.1126/science.aaa1160>. .
- [2] Schmidt, A. L. *et al.* Anatomy of news consumption on Facebook. *Proceedings of the National Academy of Sciences* **114**, 3035–3039 (2017). URL <http://www.pnas.org/lookup/doi/10.1073/pnas.1617052114>. .
- [3] Del Vicario, M., Zollo, F., Caldarelli, G., Scala, A. & Quattrociocchi, W. Mapping social dynamics on Facebook: The Brexit debate. *Social Networks* **50**, 6–16 (2017). URL <http://dx.doi.org/10.1016/j.socnet.2017.02.002><http://linkinghub.elsevier.com/retrieve/pii/S0378873316304166>. .
- [4] Margolin, D. B., Hannak, A. & Weber, I. Political Fact-Checking on Twitter: When Do Corrections Have an Effect? *Political Communication* **35**, 196–219 (2018). URL <https://doi.org/10.1080/10584609.2017.1334018><https://www.tandfonline.com/doi/full/10.1080/10584609.2017.1334018>.
- [5] Budak, C., Goel, S. & Rao, J. M. Fair and Balanced? Quantifying Media Bias through Crowdsourced Content Analysis. *Public Opinion Quarterly* **80**, 250–271 (2016). URL <https://academic.oup.com/poq/article-lookup/doi/10.1093/poq/nfw007>.
- [6] Watts, D. J. & Strogatz, S. H. Collective dynamics of “small-world” networks. *Nature* **393**, 440–442 (1998).
- [7] Barabasi, A.-L. & Albert, R. Emergence of scaling in random networks. *Science* **286**, 11 (1999). URL <http://arxiv.org/abs/cond-mat/9910332>. .
- [8] Vespignani, A. Modelling dynamical processes in complex socio-technical systems. *Nature Physics* **8**, 32–39 (2011). URL <http://www.nature.com/doi/10.1038/nphys2160>.
- [9] Bovet, A., Morone, F. & Makse, H. A. Validation of Twitter opinion trends with national polling aggregates: Hillary Clinton vs Donald Trump. *Scientific Reports* **8**, 8673 (2018). URL <http://www.nature.com/articles/s41598-018-26951-y>. .
- [10] Del Vicario, M., Scala, A., Caldarelli, G., Stanley, H. E. & Quattrociocchi, W. Modeling confirmation bias and polarization. *Scientific Reports* **7**, 40391 (2017). URL <http://dx.doi.org/10.1038/srep40391><http://www.nature.com/articles/srep40391>. .
- [11] Askitas, N. Explaining opinion polarisation with opinion copulas. *PLOS ONE* **12**, e0183277 (2017). URL <http://dx.plos.org/10.1371/journal.pone.0183277>.
- [12] Klayman, J. & Ha, Y.-W. Confirmation, disconfirmation, and information in hypothesis testing. *Psychological Review* **94**, 211–228 (1987). URL <http://doi.apa.org/getdoi.cfm?doi=10.1037/0033-295X.94.2.211>.
- [13] Mocanu, D., Rossi, L., Zhang, Q., Karsai, M. & Quattrociocchi, W. Collective attention in the age of (mis)information. *Computers in Human Behavior* **51**, 1198–1204 (2015). URL <http://linkinghub.elsevier.com/retrieve/pii/S0747563215000382>. .
- [14] Qiu, X., F. M. Oliveira, D., Sahami Shirazi, A., Flammini, A. & Menczer, F. Limited individual attention and online virality of low-quality information. *Nature Human Behaviour* **1**, 0132 (2017). URL <http://www.nature.com/articles/s41562-017-0132>. .
- [15] Varol, O., Ferrara, E., Davis, C. A., Menczer, F. & Flammini, A. Online human-bot interactions: detection, estimation, and characterization. In *Proc. 11th Int. AAAI Conf. Weblogs Soc. Media*, 280–289 (2017).
- [16] Morone, F. & Makse, H. A. Influence maximization in complex networks through optimal percolation. *Nature* (2015).
- [17] Katz, L. A new status index derived from sociometric analysis. *Psychometrika* **18**, 39–43 (1953). URL <http://link.springer.com/10.1007/BF02289026>.
- [18] Strobl, E. V., Zhang, K. & Visweswaran, S. Approximate Kernel-based Conditional Independence Tests for Fast Non-Parametric Causal Discovery 1–25 (2017). URL <http://arxiv.org/abs/1702.03877>. .
- [19] Runge, J. *et al.* Identifying causal gateways and mediators in complex spatio-temporal systems. *Nature Communications* **6**, 8502 (2015). URL <http://www.nature.com/articles/ncomms9502>. .

- [20] Spirtes, P., Glymour, C. & Scheines, R. *Causation, Prediction, and Search* (MIT Press, 2000).
- [21] Runge, J., Heitzig, J., Petoukhov, V. & Kurths, J. Escaping the Curse of Dimensionality in Estimating Multivariate Transfer Entropy. *Physical Review Letters* **108**, 258701 (2012). URL <https://link.aps.org/doi/10.1103/PhysRevLett.108.258701>.
- [22] Zhang, K., Peters, J., Janzing, D. & Schoelkopf, B. Kernel-based Conditional Independence Test and Application in Causal Discovery. In *UAI 2011, Proceedings of the Twenty-Seventh Conference on Uncertainty in Artificial Intelligence*, 804–813 (Barcelona, 2011). URL <http://arxiv.org/abs/1202.3775>.
- [23] Benjamini, Y. & Hochberg, Y. Controlling the False Discovery Rate : A Practical and Powerful Approach to Multiple Testing. *Journal of the Royal Statistical Society Series B* **57**, 289–300 (1995).
- [24] Vosoughi, S., Roy, D. & Aral, S. The spread of true and false news online. *Science* **359**, 1146–1151 (2018). URL <http://www.sciencemag.org/lookup/doi/10.1126/science.aap9559>.
- [25] Runge, J., Sejdinovic, D. & Flaxman, S. Detecting causal associations in large nonlinear time series datasets (2017). URL <http://arxiv.org/abs/1702.07007>.
- [26] Runge, J. TIGRAMITE software. URL <https://jakobrunge.github.io/tigramite>.
- [27] Strobl, E. V. RCIT and RCoT software. URL <https://github.com/ericstrobl/RCIT>.
- [28] Eichler, M. & Didelez, V. On Granger causality and the effect of interventions in time series. *Lifetime Data Analysis* **16**, 3–32 (2010).
- [29] Lee, K., Eoff, B. D. & Caverlee, J. Seven Months with the Devils: A Long-Term Study of Content Polluters on Twitter. In *Proceedings of the Fifth International AAAI Conference on Weblogs and Social Media*, 185–192 (2006).
- [30] Bessi, A. & Ferrara, E. Social bots distort the 2016 U.S. Presidential election online discussion. *First Monday* **21** (2016).
- [31] Ferrara, E., Varol, O., Davis, C., Menczer, F. & Flammini, A. The rise of social bots. *Communications of the ACM* **59**, 96–104 (2016). URL <http://arxiv.org/abs/1407.5225><http://dl.acm.org/citation.cfm?doi=2963119.2818717>.
- [32] Shao, C., Ciampaglia, G. L., Varol, O., Flammini, A. & Menczer, F. The spread of misinformation by social bots. *arXiv* (2017). URL <http://arxiv.org/abs/1707.07592>.
- [33] Shao, C. *et al.* Anatomy of an online misinformation network. *arXiv* (2018). URL <http://arxiv.org/abs/1801.06122>.

Reviewers' comments:

Reviewer #1 (Remarks to the Author):

I appreciate the authors' efforts to address my concerns. I am satisfied with regard to my first 4 points. On the 5th point, while I agree with the authors' explanation, I still feel the ms requires some clarification so that the reader is not misled.

First, the authors correctly pointed out, I had confused in-degree and out-degree in my review. In re-reading the ms, however, I find myself doing the same again. The problem is that, in my mind and I expect in many readers' minds, the act of retweeting is associated with sending information out. So one naturally assumes that high "out-degree" is something performed by the user. This is also typical in social network analysis of small networks. Out-degree is the friends I nominate, in-degree is the friends who nominate me. My point isn't that this is the "right" way to think about it, only that it is, for myself and probably many others, the natural way and so when digesting complex information, particularly with other complementary poles (left vs. right), it is easy to get confused or interpret everything in a reversed manner.

I don't think the authors should change their terms—this manner is consistent with other retweet studies. But it would reduce reader confusion if there could be more reminders of which is which so that the substance of the claims is more apparent.

This leads to the larger, more substantive point I pointed to in my initial review. Admittedly my articulation was not so clear, but in reading the authors' response, and the revised ms, I have a better fix on what concerns me.

The problem stems from the meaning of the term "connectivity" and its interpretation within regions of a network of self-selected nodes. Connectivity can mean, technically, the quantity of connections within the network (measured by density or average degree etc.). And it can mean, substantively, the social cohesiveness of the network--its ability to spread information and so forth.

Unfortunately, in this context, the two are not as closely tied together as they normally would be because of the nature of retweet networks as "self-selected." Specifically, because they exclude isolates -- individuals who consume and share this information but do not retweet -- these networks are specifically biased in the way they represent connectivity between networks of different sizes.

A network with a small number of active individuals and a large number of isolates is not well connected in a substantive way. And if isolates are measured, such a network will show low connectivity via average degree because of all of the 0 degree scores. Furthermore, in such networks, a community that converts isolates to nodes of degree 1 raises average degree and raises connectivity and cohesiveness. But in a retweet network, where isolates are not included, such conversions penalize the connectivity as scored by average degree. Suddenly there are more 1 or 2 degree nodes, reducing average degree, but they do not replace 0 degree nodes, as they were censored out. So at the margins, retweet networks that inspire participation, generating low degree nodes, appear less "connected" than networks that don't.

Another way of saying this is that, in a retweet network where isolates are excluded, size itself is an indicator of connectivity. That is, larger networks are better connected in some ways than smaller ones, because they have penetrated a larger portion of the underlying population. This argument does not counter the author's technical findings, but it means they should be careful with the words used to describe those findings.

Based on this I have a few suggestions. First, instead of using the broad term "connectivity,"

which is technically correct but substantively ambiguous in this context, the authors should describe what each measure means for substantive understanding.

Second, the average degree analysis should not be presented in a way that suggests any conclusions can be drawn from it on its own, without consulting the distributions. The authors do draw such conclusions on p. 9 when they state "However the retweet networks... tighter community structures." It would be better to hold off on this until after the distributions have been described and compared.

Thirdly, the big question is whether the larger networks (center, left-leaning) are elaborations of the smaller ones (fake news, extreme right), or whether they entirely lack the structure that the smaller ones have achieved. For example, is there a "fake news-like" subgraph nested within the center left? Is the center-left a fake-news like core plus some very high out degree nodes and a large number of excess, low in degree nodes? If this is the case, the underlying distributions would be different, but the substantive meaning would overlap.

Looking at figure 3, and heeding the author's analysis of the probability of low out degree nodes (p. 9), this doesn't seem likely. But if this is more directly answerable it would add substantially to the understanding and limit misinterpretations.

Also, the following wording on page 9 could be clarified "We also see that fake, extremely biased (right and left) and right networks are characterized by a higher probability of having lower degree nodes ($k_{out} < 100$), indicating that the audience of these news is on average more connected, i.e. retweet more people, than the audience of more traditional news." I had to read this several times to get it right in my head. What is meant, I think, is that in these extreme networks, a greater portion of the degree (retweets) appear in the lower degree nodes, thus indicating a diversity of attention. At any rate, it is confusing to think about a higher probability of lower degree nodes implying more connectivity without referring to the contrast (that there are fewer high degree nodes).

Signed,
Drew Margolin

Reviewer #2 (Remarks to the Author):

The authors have done a thorough job responding to my two main issues: the rating of bias and ideology among urls, and the causal analysis. The main remaining issues are mainly ones of presentation: the article continues to do a poor job of foregrounding and streamlining its main results. I discuss a few such matters below, but enjoin the editors to spend particular care in polishing and organizing the exposition, if possible.

Regarding the revisions, the distinction into extreme-bias-left through extreme-bias-right plus fake news is much clearer. While one can always quibble, reducing the total set to a smaller number that can be more directly manually verified is substantial improvement.

Regarding the causal analysis, while one can always quibble with an approach, the revised methods are much superior to the previous granger causality, even if the overall results remain the same. And the joint analyses and plots in Figures 6 and 7 are a distinct improvement as well. One remaining puzzling aspect for me is how the right "influencers" are nevertheless caused by, and not causal of, the Trump or other groups. This apparent contrast between the network and temporal analyses seems under-explored in the discussion. In particular, the paragraph added ("The interpretation of...") and comment made ("In our case...") on page 17 of the author

response are well put, but there still seems to be a tension between the belief that the center/left influencers are truly causing the Clinton supporters because they are journalists, and the implicit suggestion that nevertheless the fake news and right are not actually causing the Trump supporters despite being "influencers". Perhaps this is just a confusion due to the term "influencers," which emphasizes the temporal activity at least as much as the network structures; this confusion is especially compounded in the final discussion where the (equally confusing) term "spreaders" seems to appear instead.

I also appreciate the authors responses to the other reviewers, particularly trying to parse out the unique effects of Trump staff and Breitbart, although the claim that removing Breitbart from the analysis doesn't change the results could be made more explicitly in the main text.

A few other minor points:

- I find that neither figure 2 nor 3 is particularly informative. It seems like Figure 2 would be better with all the groups/networks combined in a single network, rather than a series of nearly-identical artificially separated sub-networks.
- Figure 5 is potentially a bit misleading, suggesting that the fake news influencers in the bottom panel are the analog to the left-leaning influencers in the top panel.
- P 14: why a threshold of 0.49?
- Page 3: The newly added material beginning with "We find that each website..." is confusingly written.

Reviewer #3 (Remarks to the Author):

First of all, I would like to thank the authors for their effort to improve the manuscript and addressing most of the reviewers' comments and providing additional evidence.

In general, I am satisfied with the revised version. Yet, I would urge the authors to formulate some claims more carefully to prevent confusion and reduce the risk of misinterpretation.

In the abstract authors state "We find that 25% of these tweets disseminate fake or extremely biased news". Although I raised earlier my concern about classifying single URL based on the domain, this sentence also states a conclusion regarding the content of an article. I think a more strict statement may require the authors to investigate the fraction of fake or extremely biased articles in each domain and their true popularity. Do extremely biased news from Breitbart shared more than some of the factual news from the same site?

Since the authors collected individual tweets, it would be a useful exercise to evaluate most shared articles in each category to show the categorization of the articles aligns with the label assigned to the domain.

Yet, these are ideas to improve the paper-- up to the authors to consider or not. The paper is fir for publication, either way.

Response to Referees Letter

Dear Editor and Reviewers:

We have received your reports on our manuscript “Influence of fake news in Twitter during the 2016 US presidential election”(NCOMMS-18-08034). We wish to thank you for your constructive comments and for the time spent on the review process. We hope that you will find that these new revisions fully clarify our manuscript and make it adequate to be reconsidered for publication in Nature Communications.

In the following we have colored the Reviewers comments in blue and the text pasted from the revised manuscript in red to facilitate reading.

“ Reviewer #1 (Remarks to authors):

I appreciate the authors’ efforts to address my concerns. I am satisfied with regard to my first 4 points. On the 5th point, while I agree with the authors’ explanation, I still feel the ms requires some clarification so that the reader is not misled. ”

We are glad that the Referee is satisfied with regards to his first 4 points and hope that he will find his last point clarified in this second revision.

“ First, the authors correctly pointed out, I had confused in-degree and out-degree in my review. In re-reading the ms, however, I find myself doing the same again. The problem is that, in my mind and I expect in many readers’ minds, the act of retweeting is associated with sending information out. So one naturally assumes that high “out-degree” is something performed by the user. This is also typical in social network analysis of small networks. Out-degree is the friends I nominate, in-degree is the friends who nominate me. My point isn’t that this is the “right” way to think about it, only that it is, for myself and probably many others, the natural way and so when digesting complex information, particularly with other complementary poles (left vs. right), it is easy to get confused or interpret everything in a reversed manner.

I don’t think the authors should change their terms—this manner is consistent with other retweet studies. But it would reduce reader confusion if there could be more reminders of which is which so that the substance of the claims is more apparent. ”

We agree that our definition of out- and in-degree could be confusing for readers used to the opposite definition. As the Referee points out, our definition is consistent with other retweet studies and is the natural definition for the study of information spread since, with our definition, the direction of edges corresponds to the direction of information flow.

To clarify this point we added the following remark on page 6:

“ This definition of the direction of the links is well suited for our study as the direction of the links represents the direction of the information flow between Twitter users. ”

We also added the following note in the caption of Fig. 3 to remind the reader of our definition (a similar note was already present in the caption of Tab. 2):

“ The out-degree of a node, i.e. a user, is equal to the number of different users that have retweeted at least one of her/his tweets. Its in-degree represents the number of different users she/he retweeted. ”

As well as the following note in the caption of Fig. 2:

“ The size of the nodes is proportional to their Collective Influence score, CI_{out} , and the shade of the nodes’ color represents their out-degree, i.e. the number of different users that have retweeted

at least one of her/his tweets with a URL directing to a news outlet, from dark (high out-degree) to light (low out-degree). ”

“ This leads to the larger, more substantive point I pointed to in my initial review. Admittedly my articulation was not so clear, but in reading the authors’ response, and the revised ms, I have a better fix on what concerns me.

The problem stems from the meaning of the term “connectivity” and its interpretation within regions of a network of self-selected nodes. Connectivity can mean, technically, the quantity of connections within the network (measured by density or average degree etc.). And it can mean, substantively, the social cohesiveness of the network—its ability to spread information and so forth.

Unfortunately, in this context, the two are not as closely tied together as they normally would be because of the nature of retweet networks as “self-selected.” Specifically, because they exclude isolates – individuals who consume and share this information but do not retweet – these networks are specifically biased in the way they represent connectivity between networks of different sizes.

A network with a small number of active individuals and a large number of isolates is not well connected in a substantive way. And if isolates are measured, such a network will show low connectivity via average degree because of all of the 0 degree scores. Furthermore, in such networks, a community that converts isolates to nodes of degree 1 raises average degree and raises connectivity and cohesiveness. But in a retweet network, where isolates are not included, such conversions penalize the connectivity as scored by average degree. Suddenly there are more 1 or 2 degree nodes, reducing average degree, but they do not replace 0 degree nodes, as they were censored out. So at the margins, retweet networks that inspire participation, generating low degree nodes, appear less “connected” than networks that don’t.

Another way of saying this is that, in a retweet network where isolates are excluded, size itself is an indicator of connectivity. That is, larger networks are better connected in some ways than smaller ones, because they have penetrated a larger portion of the underlying population. This argument does not counter the author’s technical findings, but it means they should be careful with the words used to describe those findings.

Based on this I have a few suggestions. First, instead of using the broad term “connectivity,” which is technically correct but substantively ambiguous in this context, the authors should describe what each measure means for substantive understanding.

Second, the average degree analysis should not be presented in a way that suggests any conclusions can be drawn from it on its own, without consulting the distributions. The authors do draw such conclusions on p. 9 when the state “However the retweet networks... tighter community structures.” It would be better to hold off on this until after the distributions have been described and compared. ”

We thank the Referee for clarifying this issue. We agree that our usage of the term “connectivity” was imprecise and that conclusions should not be drawn without consulting the distributions.

We amended our remark on page 7 to clarify our usage of the term “more connected” and to only refer to average quantities:

“ These results show that users spreading fake and extremely biased news, although in smaller numbers, are not only more active in average, as shown in Tab.1 but also connected (through retweets) to more users in average than users in the traditional news networks. ”

We also clarified the meaning of “more densely connected” on page 20:

“ We analyzed the structure of the information diffusion network of each category of news and found that fake and extremely biased (right) news diffusion networks are more densely connected, i.e. in average users retweet more people and are more retweeted, and have less heterogeneous connectivity distributions than traditional, center and left-leaning, news diffusion networks. ”

“ Thirdly, the big question is whether the larger networks (center, left-leaning) are elaborations of the smaller ones (fake news, extreme right), or whether they entirely lack the structure that the smaller ones have achieved. For example, is there a “fake news-like” subgraph nested within the center left? Is the center-left a fake-news like core plus some very high out degree nodes and a large number of excess, low in degree nodes? If this is the case, the underlying distributions would be different, but the substantive meaning would overlap.

Looking at figure 3, and heeding the author’s analysis of the probability of low out degree nodes (p. 9), this doesn’t seem likely. But if this is more directly answerable it would add substantially to the understanding and limit misinterpretations. ”

We thank the Referee for this interesting remark. Although the degree distributions have similar shapes, our analysis indicates that the larger networks (center, left-leaning) are not just scaled up versions of the smaller ones (fake, extreme bias) as shown by significant differences in the distributions and the size independent (bootstrapped) measurements of the degree heterogeneity.

We added the following remark on page 7 to clarify this point:

“ We measure the heterogeneity of the distribution with a bootstrapping procedure (see Tab. 2) to ensure the independence of the measure on the networks’ size. Our analysis indicates that the larger networks (center, left leaning) differ from the smaller ones not just by their size but also by their structure. ”

“ Also, the following wording on page 9 could be clarified “We also see that fake, extremely biased (right and left) and right networks are characterized by a higher probability of having lower degree nodes ($k_{out} < 100$), indicating that the audience of these news is on average more connected, i.e. retweet more people, than the audience of more traditional news.” I had to read this several times to get it right in my head. What is meant, I think, is that in these extreme networks, a greater portion of the degree (retweets) appear in the lower degree nodes, thus indicating a diversity of attention. At any rate, it is confusing to think about a higher probability of lower degree nodes implying more connectivity without referring to the contrast (that there are fewer high degree nodes).

Signed, Drew Margolin ”

We thank the Referee for pointing out this confusing sentence. Indeed, a higher probability of lower degree nodes does not directly imply a higher average degree. It is really the entire shape of the distribution that determine the average degree. As a matter of fact the fake and extreme bias (right) distribution stay above the center and left leaning distribution up to much larger degree than $k_{out} = 100$. The center and left leaning distributions are more straight (on the log-log plot) and decrease with a steeper slope resulting in a lower average degree.

We have removed this sentence and replaced it with the following in order to clarify this point on page 10:

“ We also see that the degree distributions of the fake, extremely biased (right) and right networks are characterized by less steep slopes on the log-log plots than the other distributions, resulting in a larger average degree, thus indicating a wider diversity of attention from the audience of these news, i.e. they typically retweet more people and are retweeted by more people, than the audience of more traditional news. ”

“ Reviewer #2 (Remarks to Authors):

The authors have done a thorough job responding to my two main issues: the rating of bias and ideology among urls, and the causal analysis. The main remaining issues are mainly ones of presentation: the article continues to do a poor job of foregrounding and streamlining its main results. I discuss a few such matters below, but enjoin the editors to spend particular care in polishing and organizing the exposition, if possible.

Regarding the revisions, the distinction into extreme-bias-left through extreme-bias-right plus fake news is much clearer. While one can always quibble, reducing the total set to a smaller number that can be more directly manually verified is substantial improvement. ”

We are glad that the Referee is satisfied with our answers to his main issues and hope that he will find the foregrounding and streamlining of our main results improved in this new revision.

“ Regarding the causal analysis, while one can always quibble with an approach, the revised methods are much superior to the previous granger causality, even if the overall results remain the same. And the joint analyses and plots in Figures 6 and 7 are a distinct improvement as well. One remaining puzzling aspect for me is how the right ”influencers” are nevertheless caused by, and not causal of, the Trump or other groups. This apparent contrast between the network and temporal analyses seems under-explored in the discussion. In particular, the paragraph added (”The interpretation of...”) and comment made (”In our case...”) on page 17 of the author response are well put, but there still seems to be a tension between the belief that the center/left influencers are truly causing the Clinton supporters because they are journalists, and the implicit suggestion that nevertheless the fake news and right are not actually causing the Trump supporters despite being ”influencers”. Perhaps this is just a confusion due to the term ”influencers,” which emphasizes the temporal activity at least as much as the network structures; this confusion is especially compounded in the final discussion where the (equally confusing) term ”spreaders” seems to appear instead. ”

We thank the Referee for this remark. Indeed the usage of the term “influencers” was confusing since we find that some “influencers” do not “influence” the activity of the supporters.

To clarify this point, we replaced the term “influencers” by the term “top news spreaders” since this is, by definition, how we identify them, i.e. they are the most important sources of news. We replaced the term influencers in the entire manuscript and in Figs. 8, S3, S4 and S5.

We also added the following remark on page 19 in order to better comment on the fact that the top new spreaders on the right do not influence the activity of the supporters:

“ We note that the absence of a causal effect from the top spreaders of the fake, extremely biased and of the different right news categories on the Trump supporters’ activity suggests that the Trump supporters’ activity is not controlled by a small set of users, not even by the users that are the most important sources of news. This dynamics contrasts with the one of the Clinton supporters which seems to be governed by the activity of a small set of center and left leaning influencers. ”

“ I also appreciate the authors responses to the other reviewers, particularly trying to parse out the unique effects of Trump staff and Breitbart, although the claim that removing Breitbart from the analysis doesn’t change the results could be made more explicitly in the main text. ”

We are glad that the Referee appreciate our revisions. We added the following note on page 6 to explicitly claim that removing Breitbart from the analysis does not change the results:

“ Our analysis shows that removing Breitbart from the extreme bias category does not change our results significantly. ”

“ A few other minor points:

- I find that neither figure 2 nor 3 is particularly informative. It seems like Figure 2 would be better with all the groups/networks combined in a single network, rather than a series of nearly-identical artificially separated sub-networks. ”

We thank the Referee for this remark but we prefer to keep figure 2 to highlight how the influencers relate to their groups.

However, following the suggestion of the Referee we made a new figure where the networks are combined and we show it in the main text in Fig. 4.

We added the following note in the main text to refer to this new Figure, on page 12:

“ Figure 4 shows the combined retweet network formed by top 30 news spreaders of all media categories and reveals the separation of the top news spreaders in two main clusters and the relative importance of the top spreaders. ”

We also wish to keep figure 3 as it displays information about the differences of degree distributions of the different media categories supplementary to the average degree and heterogeneity (as pointed out by Referee 1) which is an important results of our investigation.

“ - Figure 5 is potentially a bit misleading, suggesting that the fake news influencers in the bottom panel are the analog to the left-leaning influencers in the top panel. ”

We thank the Referee for pointing out this issue. We changed the colors of the fake and left-leaning top news spreaders time series in order to clearly distinguish them in Fig. 6.

“ - P 14: why a threshold of 0.49? ”

We thank the Referee for asking a justification for this threshold. It corresponds to the value where the correlation values, when sorted decreasingly, experience the largest gap in difference, 0.07, (from 0.49 to 0.42) while the average difference of the sorted correlation values is 0.014. It is therefore a natural value to place a threshold.

We added the following note on page 15 to better explain this point:

“ The value of $r = 0.49$ corresponds to the place of the largest gap between the sorted correlation values. ”

“ - Page 3: The newly added material beginning with ”We find that each website...” is confusingly written. ”

We rewrote this section in order to make it clear. It now reads as (p. 3):

“ Websites we classified in the extremely biased (right) category, respectively extremely biased (left) category, have a ranking between *right* bias and *extreme right* bias, respectively *left* and *extreme left*, on mediabiasfactcheck.com. The bias ranking on www.allsides.com of these same websites is *right*, respectively *left*, (corresponding to the most biased categories of www.allsides.com). The website mediabiasfactcheck.com also reports a level of factual reporting for each websites and we find that all the websites we classified in the extremely bias category have a level of factual reporting which is *mixed* or worse. We also find that all the websites remaining in the fake news category have a bias between *right* and *extreme right* on mediabiasfactcheck.com. ”

“ Reviewer #3 (Remarks to Authors):

First of all, I would like to thank the authors for their effort to improve the manuscript and addressing most of the reviewers’ comments and providing additional evidence. ”

We are glad that the Referee finds that our revisions improved the initial manuscript.

“ In general, I am satisfied with the revised version. Yet, I would urge the authors to formulate some claims more carefully to prevent confusion and reduce the risk of misinterpretation.

In the abstract authors state ”We find that 25% of these tweets disseminate fake or extremely biased news”. Although I raised earlier my concern about classifying single URL based on the domain, this sentence also states a conclusion regarding the content of an article. I think a more strict statement may require the authors to investigate the fraction of fake or extremely biased articles in each domain and their true popularity. Do extremely biased news from Breitbart shared more than some of the factual news from the same site? ”

We thank the Referee for pointing out this imprecision. Although we clearly explain this issue in the main text, the abstract was still imprecise. We modified it to clarify this sentence in the abstract and it now reads as:

“ We find that 25% of these tweets link to websites containing fake or extremely biased news. ”

“ Since the authors collected individual tweets, it would be a useful exercise to evaluate most shared articles in each category to show the categorization of the articles aligns with the label assigned to the domain. ”

We thank the Referee for this interesting suggestion. We added a Supplementary Data file `SuppData_top_urls_per_category.csv` containing the top 10 urls of each media category along with notes about their classification on fact checking websites (when available) and links to the fact checking websites. We observe that the most popular URLs in the fake news category have claims that are either false, exaggerated or unproven. The most popular links in the extreme bias category have unproven claims or articles with a mixture of true and false claims. We have only found one clearly false claim in the traditional news media categories (<https://www.snopes.com/fact-check/newsweek-proves-that-wikileaks-is-leaking-phony-hillary-clinton-emails/>), however, most of the popular URLs in the traditional news categories have not been fact checked as false or true by fact-checking organizations.

We added the following remark on page 4 to clarify this point:

“ Supplementary Data file `SuppData_top_urls_per_category.csv` contains the top 10 URLs of each media category along with notes about their classification on fact checking websites (when available), links to the fact checking websites and additional information. We observe that the classification of the most popular URLs is well aligned with the label assigned to their domains. ”

“ Yet, these are ideas to improve the paper– up to the authors to consider or not. The paper is fir for publication, either way. ”

We thank again the Reviewers for their constructive comments and hope that they will find that these revisions clarified their issues and that the manuscript is fit for publication in Nature Communications.

Yours sincerely,

Alexandre Bovet & Hernán A. Makse

Reviewers' comments:

Reviewer #1 (Remarks to the Author):

I appreciate the authors efforts in addressing my comments. I'm mostly satisfied, but I do think there is some misinterpretation of what I intended regarding the "sub-graphs" question.

As I read the authors' response, their point is that based on the analysis of the network structure, including through the bootstrapping procedure, the larger networks (left-leaning; center) are structurally different from the extreme networks (extreme bias (right), fake news, right). I don't dispute this analysis. But the interpretation is more ambiguous, and in an important way. As it stands now the authors seem to imply that the networks are apples to apples, and so the most skewed distribution and lower average degree in the left-leaning, center networks have means that they lacks some structure or tendency that the extreme networks possess. But this is not necessarily the case.

Specifically, the authors' analysis can be interpreted substantively in two different ways. One is that people on the right build relationships and share information in manner that is different from people on this left. But an alternative interpretation is that there is a roughly equal number of people on the right and the left who share info (retweet) in the manner of the extreme networks, and that they are connected to each other similarly. However, the center/left leaning networks then also have a large number of other individuals "tacked on" to the network based on another feature the authors have observed – the presence of broadcast networks that feed their ideology or information needs.

As Goel, Watts and Goldstein show, "broadcasting" is disruptive to understanding diffusion patterns based on network data. For example, in these data the highest out degree node in the center category is @CNN, an account that reaches millions of people without going through Twitter. Perhaps people watch a story on TV, then go to Twitter to retweet it, an avenue that is not available to @Patriotic_Folks.

As far as I can tell, this would lead to the structures the authors have observed – larger size, lower average degree, heavier skewness etc. in the left and center left networks. And bootstrapping, or any other test that crunched data randomly drawn from each network, would not eliminate this possibility, because the network structures are in fact different. But why this difference exists is unclear.

In particular, if broadcast, or other dynamics relating celebrity to the casual engagement of a large number of users are present, comparing the aggregates only will make it look like the two different (sets of) networks are different in activity and connectivity. But this is only because broadcast (or other celebrity) dynamics select-in more people into the left networks. The activist core of the left might be identical to the right networks. Furthermore, because a retweet network is directional and based on behavior, not attitude or commitment, they may not be in any way influenced by the causal, low engagement broadcast/celebrity retweeter. In this context the networks are different, structurally, but not so different, substantively. The activists do what they do, in the same way, in both sets of networks. Then one set of activists gets broadcast/celebrity retweeters tacked on to them, changing the metrics of their network as a whole. The problem is that we lack observation of the underlying population -- low engagement users. In left leaning and center, they are present, self-selected in by exposure to broadcast diffusion; in extreme, fake news etc., they are absent, because this information doesn't have access to this form of diffusion.

So in this context I will restate my point. Maybe the center and left leaning networks each contain a "sub-network" that has the same structure as the smaller, right networks. Within this sub-network, information dynamics are the same. There is, in other words, no left/right difference. But there is a left/right difference at the broadcast level. The left networks have the imprint of broadcast dynamics – many low in degree nodes, a few enormous out degree nodes – and the

right don't, which is observable from the lists that the authors have provided. Or, alternatively, maybe there really is no (similarly sized) sub-graph within the left networks that can match the right networks in connectivity. This means there is something different about the way that the people in this networks organize and share information.

Either of these explanations for the authors findings is interest, but they are quite different. I'm not sure that the authors can distinguish them with their data – whether there is a sub-region of the center and left leaning networks that are structurally similar to the right networks, or whether it is possible to remove “broadcast dynamics” from the networks. If it's easy that would advance understanding and sharpen the paper, but I don't think it's necessary. But I do think it is necessary to acknowledge these divergent possibilities in the interpretation of the results. The difference is apparent at an aggregate level, but does not account for the fact that one set of networks, being much larger and having access to more diffusion technologies, may contain two very different sub-groups, one of which is actually not substantively different from the comparison set.

Goel, S., Watts, D. J., & Goldstein, D. G. (2012). The structure of online diffusion networks. In Proceedings of the 13th ACM conference on electronic commerce (pp. 623–638). ACM. Retrieved from <http://dl.acm.org/citation.cfm?id=2229058>

Reviewer #2 (Remarks to the Author):

This paper has been impressively improved over these rounds of review; the authors have done a great job. I particularly appreciate the revisions and improvements to the figures, and the addition of Figure 4 in response to the previous round, which makes particularly clear the partisan asymmetry of fake news. While nitpicks can be infinite, I'll keep the rest of mine to myself, and recommend publishing the article forthwith.

Reviewer #3 (Remarks to the Author):

The authors have successfully addressed most of my concerns/requests, improve their manuscript. The paper is fit for publication in its present form,

Response to Referees Letter

Dear Editor and Reviewers:

We have received your reports on our manuscript “Influence of fake news in Twitter during the 2016 US presidential election”(NCOMMS-18-08034). We are glad that most of the issues are resolved and hope to solve the last issues with this revision.

In the following we have colored the Reviewers comments in blue and the text pasted from the revised manuscript in red to facilitate reading.

“ Reviewer #1 (Remarks to authors):

I appreciate the authors efforts in addressing my comments. I’m mostly satisfied, but I do think there is some misinterpretation of what I intended regarding the “sub-graphs” question. ”

We apologize for not fully answering the Referee’s comment about the sub-graphs in our previous revision. We hope that the Referee will find that the present revision fully addresses this point.

“ As I read the authors’ response, their point is that based on the analysis of the network structure, including through the bootstrapping procedure, the larger networks (left-leaning; center) are structurally different from the extreme networks (extreme bias (right), fake news, right). I don’t dispute this analysis. But the interpretation is more ambiguous, and in an important way. As it stands now the authors seem to imply that the networks are apples to apples, and so the most skewed distribution and lower average degree in the left-leaning, center networks have means that they lacks some structure or tendency that the extreme networks possess. But this is not necessarily the case.

Specifically, the authors’ analysis can be interpreted substantively in two different ways. One is that people on the right build relationships and share information in manner that is different from people on this left. But an alternative interpretation is that there is a roughly equal number of people on the right and the left who share info (retweet) in the manner of the extreme networks, and that they are connected to each other similarly. However, the center/left leaning networks then also have a large number of other individuals “tacked on” to the network based on another feature the authors have observed – the presence of broadcast networks that feed their ideology or information needs.

As Goel, Watts and Goldstein show, “broadcasting” is disruptive to understanding diffusion patterns based on network data. For example, in these data the highest out degree node in the center category is @CNN, an account that reaches millions of people without going through Twitter. Perhaps people watch a story on TV, then go to Twitter to retweet it, an avenue that is not available to @Patriotic_Folks.

As far as I can tell, this would lead to the structures the authors have observed – larger size, lower average degree, heavier skewness etc. in the left and center left networks. And bootstrapping, or any other test that crunched data randomly drawn from each network, would not eliminate this possibility, because the network structures are in fact different. But why this difference exists is unclear. In particular, if broadcast, or other dynamics relating celebrity to the casual engagement of a large number of users are present, comparing the aggregates only will make it look like the two different (sets of) networks are different in activity and connectivity. But this is only because broadcast(or other celebrity) dynamics select-in more people into the left networks. The activist core of the left might be identical to the right networks. Furthermore, because a retweet network is directional and based on behavior, not attitude or commitment, they may not be in any way influenced by the causal, low engagement broadcast/celebrity tweeter. In this context the networks are different, structurally, but not so different, substantively. The activists do what they do, in the same way, in both sets of networks. Then one set of activists gets broadcast/celebrity tweeters tacked on to them, changing the metrics of their network as a whole. The problem is that we lack

observation of the underlying population – low engagement users. In left leaning and center, they are present, self-selected in by exposure to broadcast diffusion; in extreme, fake news etc., they are absent, because this information doesn't have access to this form of diffusion.

So in this context I will restate my point. Maybe the center and left leaning networks each contain a “sub-network” that has the same structure as the smaller, right networks. Within this sub-network, information dynamics are the same. There is, in other words, no left/right difference. But there is a left/right difference at the broadcast level. The left networks have the imprint of broadcast dynamics – many low in degree nodes, a few enormous out degree nodes – and the right don't, which is observable from the lists that the authors have provided. Or, alternatively, maybe there really is no (similarly sized) sub-graph within the left networks that can match the right networks in connectivity. This means there is something different about the way that the people in this networks organize and share information.

Either of these explanations for the authors findings is interest, but they are quite different. I'm not sure that the authors can distinguish them with their data – whether there is a sub-region of the center and left leaning networks that are structurally similar to the right networks, or whether it is possible to remove “broadcast dynamics” from the networks. If it's easy that would advance understanding and sharpen the paper, but I don't think it's necessary. But I do think it is necessary to acknowledge these divergent possibilities in the interpretation of the results. The difference is apparent at an aggregate level, but does not account for the fact that one set of networks, being much larger and having access to more diffusion technologies, may contain two very different sub-groups, one of which is actually not substantively different from the comparison set.

Goel, S., Watts, D. J., & Goldstein, D. G. (2012). The structure of online diffusion networks. In Proceedings of the 13th ACM conference on electronic commerce (pp. 623–638). ACM. Retrieved from <http://dl.acm.org/citation.cfm?id=2229058> ”

We thank the Referee for clarifying his point. We fully agree that it is important to acknowledge this possible interpretation of the difference in the structure of the diffusion networks. As the Referee suspects, we do not have a convincing manner to remove the “broadcast” dynamics from the networks. However, we added remarks that clearly acknowledge this plausible interpretation.

The first remark we added is a paragraph in the section “Networks of information flow” on page 10:

“ We note that an important difference between the largest networks, i.e center and left leaning news, and the fake and extremely biased networks is that the former have typically access to more broadcasting technologies. For example, the highest out degree node in the center media category is @CNN, an account that reaches millions of people without going through Twitter. However, broadcasting is disruptive to understanding diffusion patterns based on network data [33]. Indeed, the structural differences we observe may be explained by the fact that there is something different about the way that the people in these networks organize and share information but it may also be the case that there are subgroups of users in the center and left leaning news networks that form diffusion networks with a similar structure as the smaller fake and extremely biased news networks. In this case, the center and left leaning networks then also have a large number of other individuals added to these subgroups based on another feature – the presence of important broadcast networks that feed their ideology or information needs. ”

The second remark is in the section “Discussion” on page 21:

“ The difference is apparent at an aggregate level and does not account for the fact that one set of networks, the center and left leaning networks, being much larger and having access to more broadcasting technologies, may contain two very different subgroups, one of which may be structurally similar to the group of users diffusing fake and extremely biased news, i.e. more densely connected, and the other may be mostly responding to the broadcasting of large news outlets. ”

We also thank the Referee for the pertinent reference to Goel, S., Watts, D. J., & Goldstein, D. G. (2012) that we added to the main manuscript.

[33] Goel, S., Watts, D. J. & Goldstein, D. G. The Structure of Online Diffusion Networks. *Proceedings of the 13th ACM Conference on Electronic Commerce* **1**, 623–638 (2012)

We hope that the Referee is satisfied with this added material and will find the revised manuscript fit for publication.

“ Reviewer #2 (Remarks to Authors):

This paper has been impressively improved over these rounds of review; the authors have done a great job. I particularly appreciate the revisions and improvements to the figures, and the addition of Figure 4 in response to the previous round, which makes particularly clear the partisan asymmetry of fake news. While nitpicks can be infinite, I'll keep the rest of mine to myself, and recommend publishing the article forthwith.
”

“ Reviewer #3 (Remarks to Authors):

The authors have successfully addressed most of my concerns/requests, improve their manuscript. The paper is fit for publication in its present form, ”

We thank again the Reviewers for all of their constructive comments and hope that they will find that the manuscript fit for publication in Nature Communications.

Yours sincerely,

Alexandre Bovet & Hernán A. Makse

References

- [33] Goel, S., Watts, D. J. & Goldstein, D. G. The Structure of Online Diffusion Networks. *Proceedings of the 13th ACM Conference on Electronic Commerce* **1**, 623–638 (2012).

REVIEWERS' COMMENTS:

Reviewer #1 (Remarks to the Author):

I appreciate the authors taking the time to consider my (repeated) comments and am happy to recommend it for publication.